# Dielectrocapillarity for exquisite control of fluids

Anna T. Bui [1,2] & Stephen J. Cox [2] ✉

Spatially varying electric fields are prevalent throughout nature, such as in nanoporous materials and biological membranes, and technology, e.g, patterned electrodes and van der Waals heterostructures. While uniform fields cause free ions to migrate, for polar fluids they simply reorient the constituent molecules. In contrast, electric field gradients (EFGs) induce a dielectrophoretic force, offering fine control of polar fluids even in the absence of free charges. Despite their vast potential for optimizing fluid behavior, EFGs remain largely unexplored at the microscopic level due to the absence of a rigorous first-principles theory of electrostriction. By integrating state-of-the-art advances in liquid state theory and deep learning, we reveal how EFGs modulate fluid structure and capillarity. We demonstrate that dielectrophoretic coupling enables tunable control over the liquid–gas phase transition, capillary condensation, and fluid uptake into porous media. Our findings establish "dielectrocapillarity"–the use of EFGs to manipulate confined fluids–as a powerful mechanism for controlling volumetric capacity in nanopores, holding immense potential for energy storage, selective gas separation, and tunable hysteresis in neuromorphic nanofluidics. Furthermore, by linking nanoscale dielectrocapillarity to macroscopic dielectrowetting, we establish a foundation for field-controlled wetting and adsorption phenomena of polar fluids across length scales.

Nanoporous materials such as metal-organic frameworks[1], carbon-based supercapacitors[2,3], and nanofluidic devices[4–7] rely on their ability to uptake and store fluids, in either the gaseous or liquid state, which directly impacts the performance of energy storage[8], chemical separation[9], and filtration technologies[10]. The physics of capillarity plays a fundamental role in determining fluid uptake in these systems. It is well established–initially at the macroscopic scale through the works of Young, Laplace, and Kelvin[11,12], and later at the microscopic scale[13,14]–that adsorption depends not only on the system's thermodynamic state, but also on the confinement length and the substrate–fluid interaction. These factors are typically intrinsic material properties. As a result, extensive research into enhancing adsorption in porous materials has focused on optimizing these factors, e.g., by tuning porosity or chemical functionalization[15]. However, the

potential for manipulation by external means–using applied fields to control confined fluids–remains relatively unexplored.

Electric fields offer a compelling mechanism to control the structure, phase behavior[16], and interfacial properties[17,18] of fluids. While a uniform field exerts a direct force only on free charges such as ions, non-uniform fields with electric field gradients (EFGs) generate dielectrophoretic forces on neutral polar molecules. Dielectrowetting experiments have demonstrated that EFGs influence macroscopic contact angles, in a manner consistent with a modified Young's equation[18–21]. Whether these effects translate to nanoscale capillarity, however, is unclear. Addressing this issue is important, as porous media and membranes are rarely uniform; surface heterogeneities[3,22], defects[23], and curvature are natural sources of large EFGs. Such inhomogeneous fields can also be engineered using, e.g., patterned

[1]Yusuf Hamied Department of Chemistry, University of Cambridge, Lensfield Road, Cambridge, UK. [2]Department of Chemistry, Durham University, South Road, Durham, UK. ✉e-mail: stephen.j.cox@durham.ac.uk

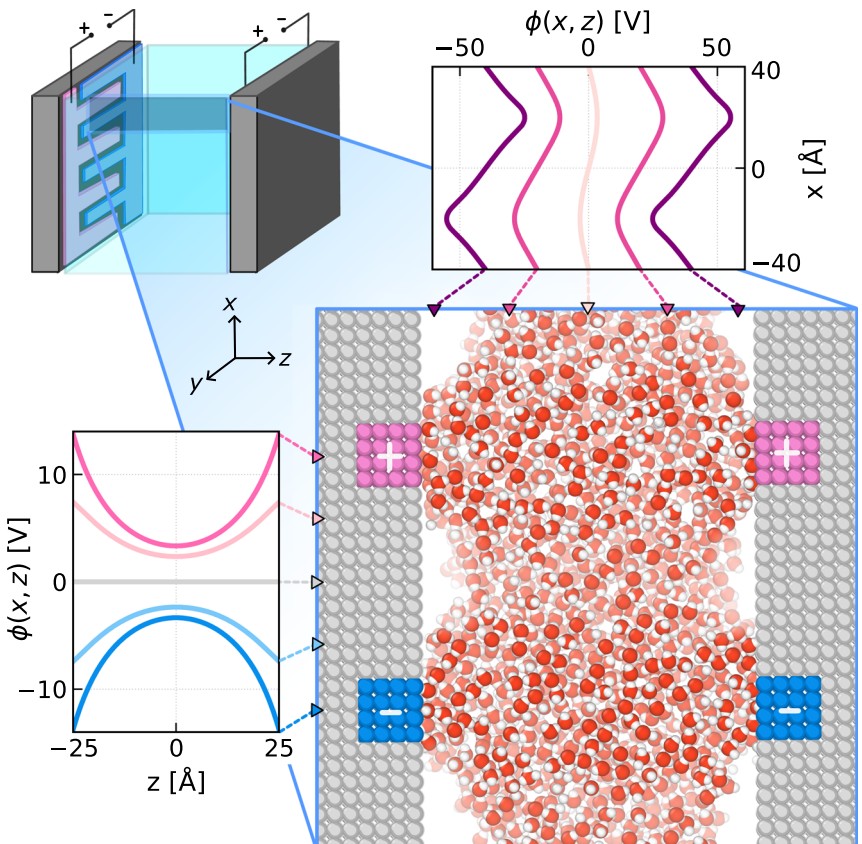

**Fig. 1 | Inhomogeneous electric fields arising from interdigitated electrodes strongly influence water's wetting behavior.** Snapshot of an SPC/E water simulation in a hydrophobic slit with alternating positive and negative electrode patches. Either holding the electrodes at a 10 V potential difference or attributing a fixed charge of $\pm 0.05\,e$/atom causes the fluid to exhibit enhanced wetting at the walls, accompanied by strong lateral density oscillations. Cross-sections of the electrostatic potential in the constant charge setup are shown parallel to the surface (top right) and normal to the surface (bottom left).

electrodes[17,18,24], atomic force microscopes[5], or layered van der Waals heterostructures[4,25].

Such EFGs introduce new length scales that can be comparable to the natural correlation lengths of the confined fluid, posing a severe challenge for a comprehensive theoretical description. With their inherent microscopic resolution, molecular dynamics simulations have provided key insights into the structural response of fluids to electric fields[26] and phase behavior under confinement[27]. However, computational limitations mean that simulation studies typically fix the number of molecules in the system, which introduces mechanical strains in an uncontrolled fashion. As a result, existing approaches—whether experimental, computational, or theoretical—face significant limitations in efficiently resolving both the microscopic restructuring and emergent macroscopic reorganization of fluids in non-uniform electric fields.

Here, we investigate how EFGs on the molecular length scale can be harnessed to manipulate mesoscopic fluid properties and phase behavior. Bringing together the latest advances in liquid state theory, computer simulation, and machine learning, we develop a multiscale framework to study electrostriction in polar fluids—that is, their density response to applied electric fields—within the grand canonical ensemble, representative of real conditions in which fluid molecules can enter and leave a pore. We show that EFGs provide tunable control over the liquid–gas phase transition and directly influence adsorption capability by capillary condensation—we dub this new phenomenon "dielectrocapillarity." Given the critical role of volumetric fluid uptake in nanoporous materials for energy storage[1], gas separation[9], and filtration technologies[10], our findings establish dielectrocapillarity as a promising avenue in fine-tuning and optimization of such processes. Furthermore, the ability to regulate hysteresis introduces a new level of programmability in nanofluidic systems, where EFG-driven control of adsorption and desorption rates could offer external tunability akin to synaptic plasticity in neuromorphic nanofluidic circuits[28,29].

## Results
### Multiscale approach for electrostriction in fluids
Although molecular simulations cannot fully capture the influence of EFGs on dielectric fluids, they offer vivid qualitative insights into the underlying physics and illustrate how EFGs may arise within nanoscale devices, as illustrated in Fig. 1. Here, we present a simulation snapshot of water at 300 K confined between two hydrophobic substrates that have been patterned with alternating stripes of positive and negative charge—a set up that provides a caricature of an interdigitated electrode. As can be clearly seen in Fig. 1, charging this device induces local wetting near the charged stripes, with pronounced density variations along the direction parallel to the surface ($x$).

These features arise from the complex interplay between the fluid's charge and number densities, and their collective response to EFGs, arising from the inhomogeneous electrostatic potential $\phi(x, z)$, where $z$ is along the surface normal. It is instructive to consider the form of $\phi$ far from either surface; here, it will resemble the linear superposition of the asymptotic limit of the two substrates considered independently,

$$\phi_{\text{single}}(x, z) \sim \phi_0 \sin\left(\frac{2\pi x}{L_x}\right) \exp\left(\frac{-2\pi|z - z_s|}{L_x}\right), \quad (1)$$

where $z_s$ indicates the plane of the substrate's outermost atoms. We therefore observe that inhomogeneity of $\phi$ is characterized by a

sinusoidal oscillation of period $L_x$ along $x$, and an exponential decay from the surface along $z$. This asymptotic analysis captures the essential behavior of $\phi(x, z)$ computed explicitly from the potential energy of a test charge, as seen in Fig. 1.

The pronounced wetting behavior observed with simulation in Fig. 1 strongly suggests that such EFGs will influence capillarity of the polar fluid; that is, the amount of fluid adsorbed at constant chemical potential. Addressing this issue, however, demands an accurate and efficient framework for determining structure, thermodynamics, and phase behavior in an open system—this lies beyond the practical limits of present day molecular simulations. Instead, we turn to classical density functional theory (cDFT), an exact statistical mechanical framework for inhomogeneous fluids.

Within cDFT, the equilibrium structure and thermodynamics of a fluid can be determined from first principles by its excess intrinsic Helmholtz free energy functional $\mathcal{F}_{\text{intr}}^{\text{ex}}([\rho], T)$, where $\rho(\mathbf{r})$ is the average inhomogeneous density of the fluid and $T$ is the temperature. This central result, established in ref. 30, places cDFT as the liquid-state generalization of its celebrated electronic structure counterpart[31]—as a modern theory for inhomogeneous fluids. cDFT is naturally formulated in the grand canonical ensemble, where the chemical potential acts as the control variable governing particle exchange in confined systems. The chemical potential, $\mu$, maps directly onto the relative humidity or vapor pressure for gases, and chemical activity for liquids.

In practice, the exact form of $\mathcal{F}_{\text{intr}}^{\text{ex}}([\rho], T)$ is generally unknown. Instead of relying on traditional approximations, we leverage state-of-the-art data-driven methodologies to supervise-learn functional mappings directly from quasi-exact reference data from grand canonical Monte Carlo simulations[32–34]. This machine-learned cDFT framework has already been successfully applied to liquid–gas coexistence[33], liquid–liquid phase separation[35], and the electric double layer[34]. Going beyond established deep-learning approaches to cDFT, we capture electrostriction arising from the coupling between mass and charge density of the fluid by explicitly learning the "hyperfunctional" $\mathcal{F}_{\text{intr}}^{\text{ex}}([\rho], T)$ where $\phi(\mathbf{r})$ is the inhomogeneous electrostatic potential and $\beta = 1/(k_B T)$ with $k_B$ the Boltzmann constant, as recently introduced in ref. 36. A practical limitation of this cDFT approach is that, at present, only inhomogeneities with planar symmetry can be investigated directly. As a result, the neural-network representation of this functional was trained on simulation data generated under random planar electrostatic potentials $\phi(z)$. Nonetheless, we demonstrate below that calculations in which $\phi$ only varies along a single cartesian direction provide general insight into the influence of EFGs on wetting and capillarity. Details of the practical implementation of the theory are given in the Methods section.

The resulting cDFT framework is not only efficiently computable on standard hardware but also unparalleled in its ability to simultaneously capture microscopic fluid structure and mesoscopic phase behavior under arbitrary non-uniform electric fields. Unlike atomistic simulations, which are computationally prohibitive for such a broad exploration, our method achieves orders-of-magnitude speedup, completing each calculation in less than a minute without sacrificing quantitative accuracy. Importantly, this computational efficiency is realized after a one-time training stage: once the functional has been learned, it can be used to perform thousands of free-energy calculations at negligible cost. This amortized advantage is what enables us to map complete phase diagrams, adsorption isotherms, and metastable fluid branches—tasks that would require enhanced sampling and thermodynamic integration if carried out by molecular simulation alone. This efficiency enables an unprecedented, highly accurate mapping of a polar fluid's response to EFGs of varying strengths and wavelengths across different thermodynamic conditions, from supercritical to subcritical regimes, spanning both bulk and confined environments. With this powerful tool in hand, we now uncover emergent electrostrictive phenomena that arise from the complex interplay of thermodynamics, confinement, and response to EFGs.

To systematically investigate how EFGs influence fluids, we primarily consider a minimal molecular model that incorporates soft-core repulsion, van der Waals attraction, and long-range dipolar interactions. The advantage of using such a simple molecular model is that we can exhaustively explore a broad range of thermodynamic conditions, while potentially uncovering common behaviors among polar fluids, from molecular liquids to colloidal systems. In addition, we also investigate a commonly used simple point charge model for water (SPC/E[37]) that explicitly incorporates hydrogen-bonding, under thermodynamic conditions close to its critical point. Where comparison with ionic fluids is made, we will also show results for a prototypical model comprising oppositely charged hard spheres[34]; in this case, we use a straightforward generalization of cDFT to multicomponent systems.

## Dielectrophoretic coupling with EFGs under non-uniform electric fields

While EFGs will be most pronounced near the surfaces that generate them, Fig. 1 demonstrates that they may persist relatively far from the interface. In the specific case considered in Fig. 1, midway between the substrates we observe a sinusoidal electrostatic potential along the $x$ direction. This motivates us to understand the direct influence of such sinusoidal potentials on the fluid, without explicitly considering interfaces. In fact, for $L_x$ larger than a few molecular diameters, we can define bulk response if we consider averages over a thin slice of thickness $\Delta z \ll L_x$ (Fig. S16). Although such a clean separation of bulk and interfacial response becomes challenging as $L_x$ approaches molecular length scales, the full potential (as opposed to its asymptotic form) comprises modes of decreasing wavelength—it is therefore instructive to understand bulk-like response across a broad range of wavelengths.

While the behavior of fluids under uniform electric fields has been extensively studied[38–41], non-uniform electric fields remain comparatively less explored. Consequently, the effects of EFGs on fluids are less well understood, even in bulk. As a starting point, we characterize the bulk response of water, the simple polar fluid, and the electrolyte, all under supercritical conditions, when subjected to a sinusoidal electric field, $E^*(z) = E_{\text{max}}^* \sin(2\pi z/\lambda)$, as shown in Fig. 2a. Quantities labeled with an asterisk are expressed in reduced units, defined in the Methods section.

In the case of water, Fig. 2b, we see that the applied electric field induces a significant structural reorganization along the field direction; its average density profile $\rho^*(z)$ is locally depleted in regions of weaker field strength, while molecular reorientation leads to an inhomogeneous average charge density distribution $n^*(z)$. As water is overall neutral and therefore experiences no net electrophoretic force, the observed local reorganization arises instead from dielectrophoretic forces[42,43], $f_{\text{DEP}} \sim \nabla E^2$, which push the fluid towards regions of higher electric field strength—an effect termed "dielectrophoretic rise." This dielectrophoretic force is the same that drives dielectrophoresis, which is widely exploited to manipulate biological cells[44] and colloids[45], but its role in molecular fluids has received little attention. While water provides an important example of a polar fluid, the observed effects are by no means specific to aqueous systems. As seen in Fig. 2c, the overall picture is the same for the simple polar fluid, aside from the fact that it exhibits a symmetric response, whereas for water, depletion is stronger in regions where $\nabla E < 0$ than where $\nabla E > 0$ due to the inherent charge asymmetry of the water molecule.

In contrast, applying a sinusoidal field to an electrolyte induces "electrophoretic rise" in which the fluid migrates toward regions of lower electric field strength (Fig. 2d). This is a result of the electrophoretic force $f_{\text{EP}} = qE$, where $E$ is the local field strength and $q$ is the ionic charge, causing the anions and cations to reorganize, with peaks

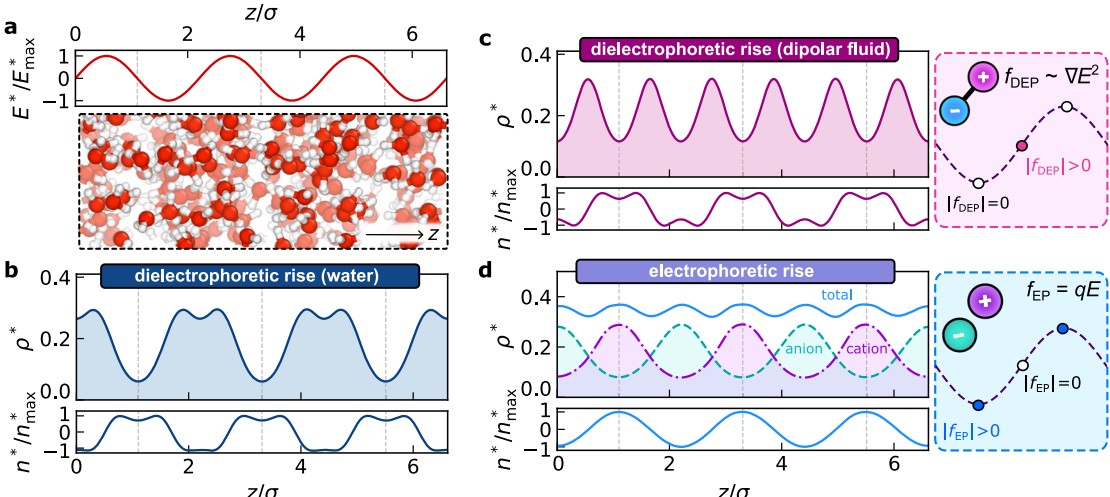

**Fig. 2 | Reorganization of fluids under non-uniform electric fields.** An applied electric field, $E^*(z) = E^*_{max} \sin(2\pi z/\lambda)$, shown in (**a**), induces pronounced density variations in bulk supercritical water, as can clearly be seen from a snapshot of a molecular dynamics simulation. **b** Results from cDFT for the number ($\rho^*(z)$, top) and charge ($n^*(z)$, bottom) densities capture this behavior. It can clearly be seen that number density is locally depleted where $|\nabla E|$ is large, and locally enhanced where $|\nabla E|$ is small. The same qualitative behavior is seen in (**c**) for a supercritical dipolar fluid, except that its response is symmetric, in contrast to water where local depletion depends upon the sign of $\nabla E$. **d** In contrast to both water and the dipolar fluid, an electrolyte is locally depleted in regions of low field strength due to electrophoretic forces (dashed purple and green lines show cation and anion density, respectively, while the solid blue line shows the total density). Reduced units are described in the Methods section.

in their density profiles out of phase due to their opposing charges. In this way, polar fluids and ionic fluids display electromechanical responses that are fundamentally distinct from each other.

Crucially, dielectrophoretic coupling depends on the EFGs as well as the absolute field strength, and therefore offers greater control over the fluid's response. To illustrate this, we present in Fig. 3a how dielectrophoretic rise can be amplified by controlling the applied field. Owing to their qualitatively similar behavior, in the remainder of the article we focus on the dipolar fluid, with results for water given in the SI, Figs. S13 and S14. As the wavelength of the sinusoidal field decreases, local mass accumulation becomes more pronounced, reflecting the system's response to larger local EFGs, even as the maximum field amplitude remains unchanged. As a function of field strength, dielectrophoretic rise exhibits strong non-linearity, as evident from the change in maximum local density $\Delta\rho^*_{max} = \rho^*_{max} - \rho^*_{max, 0}$ from zero field shown in Fig. 3b, which highlights the complexity and collectiveness of electrostrictive response in fluids.

Dielectrophoretic response is not solely determined by the magnitude of the applied EFGs but also by the fluid's intermolecular interactions. This distinction is of practical importance when considering colloidal systems in which effective interactions can be tuned. For example, in electrolyte solutions, zwitterionic Janus particles have diameters that far exceed the electrostatic screening length, making their dipolar interactions inherently short-ranged[46]. To explore the effect on colloidal fluids, we consider a nearly identical model fluid whose dipolar interactions are screened, decaying on a length scale comparable to the molecular diameter $\sigma$. When the wavelength, $\lambda$, of the electric field is comparable to the particle size, $\lambda \approx \sigma$, both fluids exhibit identical dielectrophoretic response, shown by the dashed blue line in Fig. 3b. This result reflects that, for wavelengths on the molecular scale, response is dominated by local reordering of individual particles. However, for $\lambda \gg \sigma$, behaviors differ significantly; for systems with long-ranged interactions, molecular dipoles collectively reorient to screen the applied field, weakening its effect over extended distances. In contrast, the short-ranged colloidal system lacks this screening, leading to a much stronger response (dashed purple lines, Fig. 3b and inset). Such an equilibrium effect could be leveraged for programmable directed self-assembly[47], where EFGs in combination with tunable Janus particle surfaces[48] may provide a powerful tool for

tailoring the assembly of extended structures with dielectrophoretic forces, with additional tunability arising from the solvent and ionic strength.

## Fine tuning liquid-vapor coexistence with EFGs

The results so far demonstrate that EFGs cause local reorganization of a supercritical dielectric fluid into regions of low and high density. A natural question then arises: how do EFGs influence the phase behavior of such a single-component fluid? Understanding this fundamental issue will be of central importance to the optimal design of devices with switchable functionality, in a similar spirit to the study of electric-field-induced phase transformations in solid-state materials[49,50].

In Fig. 3c, we show the density profiles of the fluid along the direction of an external sinusoidal electric field with $\lambda/\sigma = 1.7$, at $T^* < T^*_{c,0}$, where $T^*_{c,0}$ is the critical temperature in the absence of an external field. Results are shown for different chemical potentials. At zero field, the stable solutions separate into homogeneous vapor and liquid states at low and high chemical potential, respectively. As the field strength increases, stronger EFGs (in absolute terms) not only give rise to dielectrophoretic rise but also destabilize liquid–vapor phase separation. Consequently, within the coexistence region, the vapor phase becomes denser, or contracts, while the liquid phase expands. For sufficiently large EFGs ($E^*_{max} = 8$), the fluid undergoes a transition to a single-phase supercritical fluid. By locating vapor and liquid solutions with equal grand potentials, in Fig. 3d we map out the liquid–vapor binodal curve for the dipolar fluid, both without an external field and under this sinusoidal field with $E^*_{max} = 4$. Strikingly, we find that the critical temperature $T_c$ shifts downward under the non-uniform field. To our knowledge, this marks the first report of a shift in $T_c$ for a single-component fluid induced by an electric field that varies on the microscopic length scale. Notably, unlike for uniform fields[51–53], while the binodal line still represents liquid–vapor equilibrium, the density within each phase is no longer spatially uniform due to dielectrophoretic rise. In this case, the bulk density $\bar\rho$ represents the fluid's density averaged over a volume large compared to $\sigma^3$. For water, we observe similar behavior, with a downward shift in $T_c$ of $\approx 50$ K at $E_{max} = 0.4$ V Å$^{-1}$ (Fig. S14). These observations highlight the exquisite level of control that one can exert over dielectric fluids with EFGs.

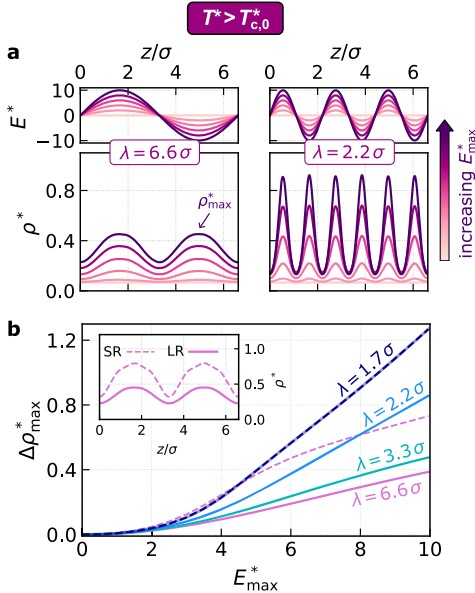

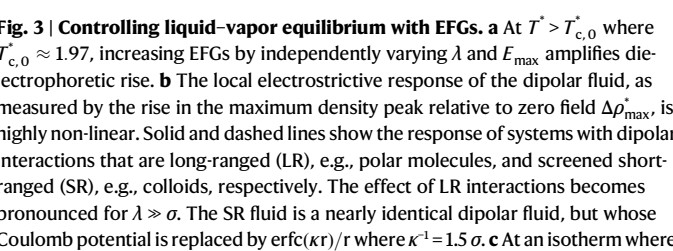

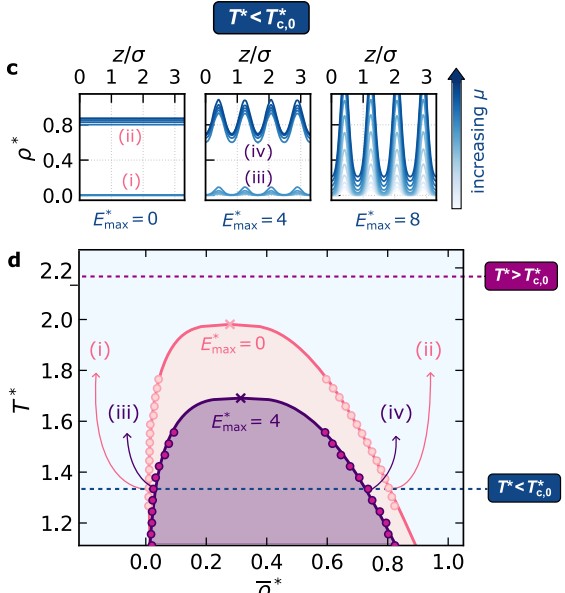

**Fig. 3 | Controlling liquid–vapor equilibrium with EFGs. a** At $T^* > T^*_{c,0}$ where $T^*_{c,0} \approx 1.97$, increasing EFGs by independently varying $\lambda$ and $E_{max}$ amplifies dielectrophoretic rise. **b** The local electrostrictive response of the dipolar fluid, as measured by the rise in the maximum density peak relative to zero field $\Delta \rho^*_{max}$, is highly non-linear. Solid and dashed lines show the response of systems with dipolar interactions that are long-ranged (LR), e.g., polar molecules, and screened short-ranged (SR), e.g., colloids, respectively. The effect of LR interactions becomes pronounced for $\lambda \gg \sigma$. The SR fluid is a nearly identical dipolar fluid, but whose Coulomb potential is replaced by $\text{erfc}(\kappa r)/r$ where $\kappa^{-1} = 1.5\,\sigma$. **c** At an isotherm where $T^* < T^*_{c,0}$, stable solutions for the density $\rho^*(z)$ under a sinusoidal electric field with $\lambda/\sigma = 1.7$, are shown for different values of the chemical potential. These results are used to investigate liquid–vapor coexistence. **d** Results in light pink show the binodal of the dipolar fluid in the absence of an electric field. At $E^*_{max} = 4$, $T^*_c$ shifts to a lower temperature, as seen in the binodal in dark purple. Solid symbols show results obtained from the multiscale cDFT approach, while crosses indicate estimates of $T^*_c$ using the law of rectilinear diameters and critical exponents[14]. Solid lines serve as a guide to the eye.

Here, we have tuned the EFGs by varying $E^*_{max}$ for fixed $\lambda$. We could also have varied $\lambda$ for fixed $E^*_{max}$, which provides additional control over the phase behavior (Fig. S11); importantly, these results demonstrate that the effects we have reported remain at larger wavelengths. Since such phase transitions are fully reversible, they will be particularly relevant for functional nanofluidic and nano-electromechanical devices where dynamic and reconfigurable phase control is advantageous[4,5]. The act of confinement by itself already leads to fluid behavior that can differ substantially from bulk. EFGs represent an additional powerful tool for tailoring the properties of fluids for the purposes of device design.

## Controlling adsorption into porous media through dielectrocapillary phenomena

In a slit pore, uptake of a fluid can be monitored by isothermal adsorption. For fluids below their critical point, of which liquid water at room temperature is an important example, adsorption is governed by capillary effects, such that condensation can occur at chemical potentials below saturation. This phenomenon, known as capillary condensation, underlies the filling of nanochannels[4], in which fluid uptake is controlled by adjusting the relative humidity of the environment. Such an experimental setup describes an equilibrium between the nanochannel and a reservoir (Fig. 4a), and maps directly on to our theoretical framework, which is formulated in the grand canonical ensemble. In the absence of an applied external field, it is well-established that capillarity is controlled by the chemical potential of the reservoir, the length scale of confinement, the substrate–fluid interaction and temperature[13]. With EFGs, we introduce an additional experimental handle by which to control fluid adsorption behavior; we call this new phenomenon "dielectrocapillarity."

We investigate a liquid at $T^*/T^*_{c,0} = 0.68$ confined to a solvophobic slit comprising two repulsive walls separated by a distance $H = 6.6\,\sigma$. In Fig. 4b, we show the fluid uptake into the slit as a function

of the chemical potential, referenced to its coexistence value at zero field, $\Delta \mu = \mu - \mu_{co,0}$, typical of an adsorption/desorption isotherm measurement. In the absence of an electric field, the transition is discontinuous, exhibiting a hysteresis loop—this is a well-established hallmark of capillary condensation in nanopores and mesopores, observed across experiments, simulations, and theory[13,54–56]. Such hysteresis arises from the metastability of the vapor during condensation and liquid during evaporation.

To gain insight into the influence of EFGs, we perform a "computational experiment" in which we apply a sinusoidal electric field across the slit. While such a set up does not correspond to an EFG established by the substrate walls themselves (see, e.g., Fig. 1), it does allow us to assess the effects of a particular mode. The impact is twofold: (1) condensation shifts to more negative $\Delta \mu$, i.e., the slit can be filled at even lower humidity; and (2) hysteresis is reduced. Keeping $\lambda/\sigma = 1.7$ fixed, for sufficiently high field strengths, hysteresis disappears entirely. In other words, we have changed the nature of the transition from first-order to continuous. This behavior directly results from the dielectrophoretic coupling that drives the fluid toward its supercritical state (Fig. 3). Important for our fundamental understanding of fluids under confinement, this result demonstrates that EFGs not only shift the bulk critical temperature, but also influence the capillary critical temperature.

Non-uniform electric fields clearly offer an additional lever for controlling capillary filling. As illustrated in Fig. 4c, adsorption/desorption can be actively controlled by switching the field on or off, offering a precise, tunable means of regulating fluid uptake. Such an ability to tailor hysteresis introduces a new level of programmability in nanofluidic systems, where EFGs can potentially serve as an external control parameter for dynamically altering adsorption and desorption rates. Notably, tunable hysteresis in capillary condensation can also serve as a memory mechanism in neuromorphic nanofluidic circuits[28,29], where phase transitions encode state-dependent responses akin to synaptic plasticity.

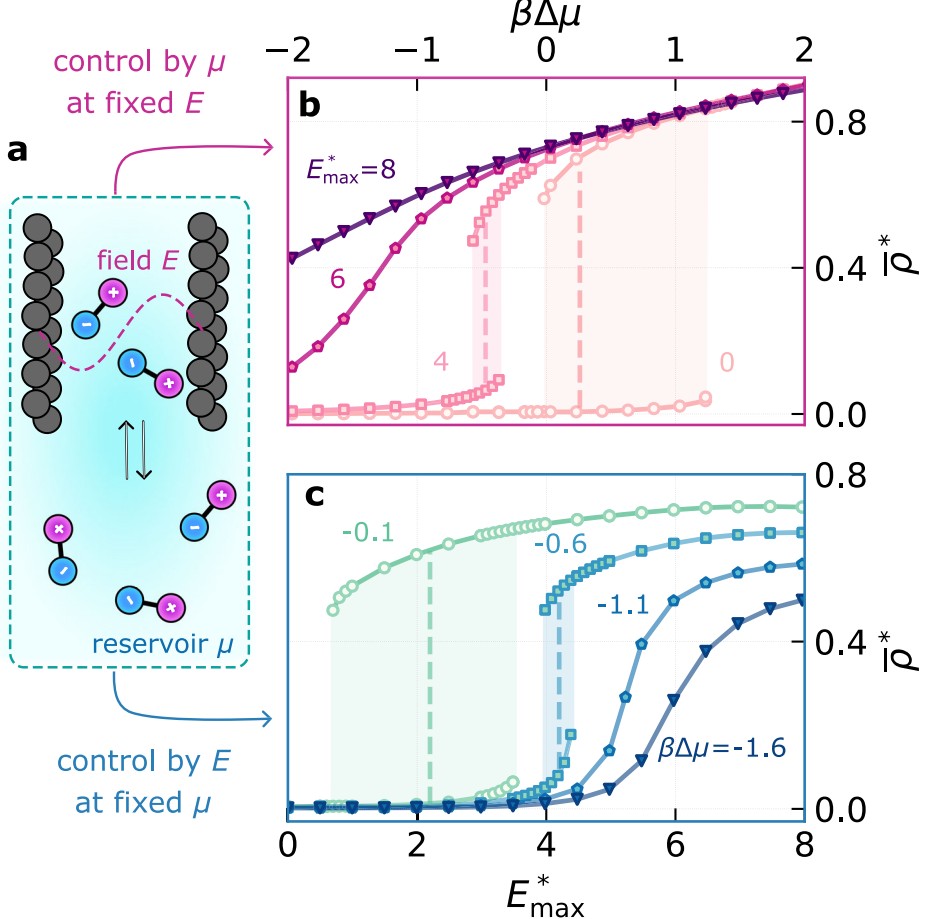

**Fig. 4 | Control of fluid uptake by dielectrocapillarity. a** Schematic of a fluid in a slit pore, with $H/\sigma = 6.6$, in equilibrium with a reservoir at chemical potential $\mu = \mu_{co,0} + \Delta\mu$. **b** Adsorption/desorption isotherms at $T^*/T^*_{c,0} = 0.68$ obtained by varying $\mu$ for different $E_{max}$ at fixed $\lambda/\sigma = 1.7$. Larger $E_{max}$ promotes adsorption, while simultaneously decreasing hysteresis. For large enough $E^*_{max}$, the transition becomes continuous. **c** Adsorption/desorption isotherms at $T^*/T^*_{c,0} = 0.68$ obtained by varying $E_{max}$ at fixed $\lambda/\sigma = 1.7$ for different $\Delta\mu$. Changing $E_{max}$ can switch the pore between filled and empty states. The vertical dashed lines indicate the equilibrium transition, i.e., where both adsorbed "liquid" and "gas" states are stable.

## Connecting to dielectrowetting experiments

We have introduced dielectrocapillarity as an additional mechanism of controlling capillary phenomena, complementing electrocapillarity[57] where electrolytes respond to applied potentials. On the macroscopic scale, these effects manifest in electrowetting[19] and dielectrowetting[18,20], where the contact angle of a droplet can be tuned with applied potentials. Our nanoscale simulations in Fig. 1 show that a similar phenomenology emerges under confinement: wetting is strongly enhanced directly over the electrode patches, while the electrostatic potential generated by the interdigitated electrodes decays into the slit. Following previous experimental work[18,20], this motivates us to employ a simple planar electrostatic potential, $\phi(z) = \phi_0 \exp(-2\pi z/L_x)$ to probe wetting at the nanoscale. Similar to our arguments above, results obtained from such a potential can be considered to report on average behavior in a slice of thickness $\Delta x \ll L_x$ (Fig. S16).

Within a macroscopic model, the resulting contact angle can be described by a modified Young's law[18,20],

$$\cos\theta(\phi_0) = \cos\theta_0 + \frac{\alpha}{\gamma_{lv}}\phi_0^2, \qquad (2)$$

where $\theta_0$ is the Young's contact angle at zero field, $\gamma_{lv}$ the liquid–vapor surface tension, and $\alpha$ a material parameter related to the dielectric permittivity of the liquid. This relationship assumes that the electrostatic energy stored in the liquid droplet is well-described by dielectric continuum theory, and that $L_x$ is sufficiently small so that any changes in energy due to the electric field are effectively localized to the solid-liquid interface. Under these assumptions, the electrostatic free energy per unit area obeys the simple quadratic scaling $\gamma_{elec} = -\alpha\phi_0^2$.

Our multiscale framework provides a microscopic perspective on this phenomenology, and allows us to test whether, at the nanoscale, EFGs indeed enhance the wetting of dielectric liquids and the extent to which the scaling prescribed by Eq. (2) holds. To this end, we applied $\phi(z)$ with $L_x \approx 7\sigma$ to the confined dipolar model, symmetrically from both walls of a solvophobic slit. While not trained on such electrostatic potentials, as can be seen in Fig. 5b, the neural functional extrapolates well, yielding physically plausible results, aside from some oscillations at high wetting that are likely minor artifacts. As $\phi_0$ increases, so too does the contact density at the wall, verifying that enhanced wetting occurs. We quantify this effect by computing $\gamma_{elec} = \frac{1}{2}\int_{-\infty}^{\infty} dz\,\phi(z)n(z)$, which, as can be seen in Fig. 5c, decreases in an approximately quadratic fashion as $\phi_0$ increases. By identifying $\alpha = -\partial\gamma_{elec}/\partial(\phi_0^2)$, we reconstruct the potential-dependent contact angle, as shown in Fig. 5c. While the results of our microscopic theory are broadly in line the macroscopic model (Eq. (2)) some subtle differences are observed, especially at small $\phi_0$. These appear to be correlated with significant changes in the local compressibility near the interface, $\chi_\mu(z) = \partial\rho(z)/\partial\mu$, as $\phi_0$ increases, which indicates a suppression of density fluctuations. Such microscopic details are lacking in the dielectric continuum model that underpins Eq. (2).

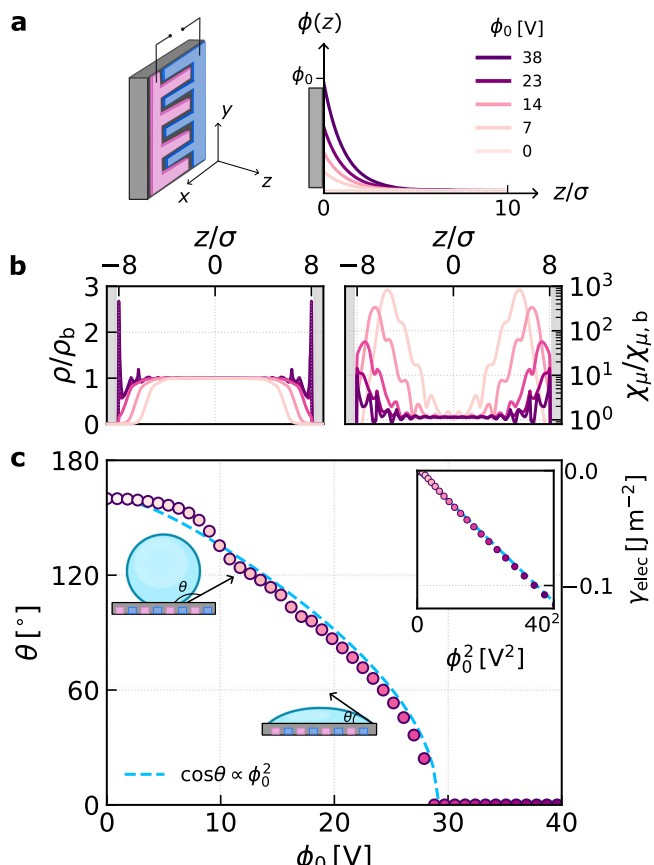

**Fig. 5 | Connecting to dielectrowetting experiments.** The electrostatic potential from interdigitated electrodes, shown schematically in (**a**), decays exponentially. Applying this potential symmetrically from both confining walls in a slit geometry with $H \approx 16\,\sigma$ enhances wetting of the solid-liquid interface, as can be seen in the density profiles (left) and changes in local compressibility (right) in (**b**) (both quantities are normalized by their bulk values). **c** Electrostatic free energy per unit area from cDFT (inset) exhibits a quadratic dependence on $\phi_0$, enabling reconstruction of the contact angle via Eq. (2) (assuming $\theta_0 = 160°$ and using $\gamma_{lv} = 0.025\,\mathrm{J\,m^{-2}}$ computed from direct coexistence simulation), in direct analogy with dielectrowetting experiments[18,20].

## Discussion

Our findings establish EFGs as a powerful and versatile tool for manipulating fluids. We have revealed their ability to structure fluids, modulate phase transitions, and control capillary effects. Crucially, we demonstrate that EFGs not only influence a fluid's behavior in bulk, but also give rise to dielectrocapillarity, a new phenomenon in which capillary condensation and criticality under confinement can be finely tuned. By placing this nanoscale physics in direct correspondence with macroscopic dielectrowetting experiments, our work provides a microscopic foundation for the design of EFG-controlled wetting and adsorption phenomena. These discoveries are made possible by our development of a multiscale approach that provides a first-principles description of electromechanics[36].

The effects uncovered in this work concern the equilibrium behavior of dielectric liquids, and omit potentially important nonequilibrium effects such as pore entry and exit[58–60], electrokinetic phenomena[61–63], and controlled wetting dynamics such as rate-dependent droplet spreading[20]. Nonetheless, the implications of our results for nonequilibrium behavior are potentially far-reaching. A natural possible progression from this work is to augment our first-principles framework for electromechanics with dynamical extensions of cDFT[64], opening a promising route toward a microscopic

understanding of how EFGs impact non-equilibrium processes. Moreover, the current framework naturally accommodates mixtures of dielectric liquids – in such cases, the excess intrinsic free energy, $\mathcal{F}_{\mathrm{intr}}^{\mathrm{ex}}([\{\rho_\nu\}, \beta\phi], T)$ acquires a functional dependence on the density fields of all species present, with $\nu$ indexing each component. This generalization opens the door to investigating more intricate phase behavior and interfacial phenomena—including liquid–liquid phase separation driven by EFGs[16,65].

The ability to reversibly control phase behavior and adsorption with electric fields unlocks new avenues for manipulating fluids across multiple length scales, from adaptive nanofluidic devices, to tunable sorption in porous materials, to colloidal assembly. At the nanoscale, nuclear magnetic resonance techniques[3] can directly validate these effects. With strong EFGs experimentally accessible via atomic force microscope tips, optical tweezers, and patterned electrode configurations, our results lay the foundation for future experimental exploration, paving the way for new strategies in developing energy storage, selective separation, and responsive fluidic technologies.

## Methods
### Reduced units
When reported, reduced units are defined by $T^* = k_B T/\varepsilon$, $\rho^* = \rho\sigma^3$, and $E^* = E\,\sigma^{3/2}\varepsilon^{-1/2}$, where $\sigma$ sets the molecular length scale and $\varepsilon$ the energy scale. Specifically for each fluid: (i) SPC/E water, $\sigma = 3.166$ Å, $\varepsilon = 0.65\,\mathrm{kJ\,mol^{-1}}$; the dipolar fluid, $\sigma = 3.024$ Å, $\varepsilon = 1.87\,\mathrm{kJ\,mol^{-1}}$; and (iii) the electrolyte, $\sigma = 2.76$ Å, $\varepsilon = e^2/\sigma$, where $e$ is the elementary charge.

### Hyperdensity functional theory
We employed the recently developed first-principles theory for electromechanics[36] based on hyperdensity functional theory[66], which provides an exact variational framework for the coupled electromechanical response of fluids. For a single-component fluid, at specified $\mu$, $T$, planar external non-electrostatic potential $V_{\mathrm{ext}}(z)$, and electrostatic potential $\phi(z)$, the grand potential functional is

$$\Omega([\rho, \beta\phi], T) = \mathcal{F}_{\mathrm{intr}}^{\mathrm{id}}([\rho], T) + \mathcal{F}_{\mathrm{intr}}^{\mathrm{ex}}([\rho, \beta\phi], T) + \int \mathrm{d}z\, \rho(z)[V_{\mathrm{ext}}(z) - \mu],$$

where the ideal intrinsic Helmholtz free energy is $\mathcal{F}_{\mathrm{intr}}^{\mathrm{id}}([\rho], T) = k_B T \int \mathrm{d}z\, \rho(z)[\ln \zeta^{-1}\Lambda^3 \rho(z) - 1]$, with $\Lambda$ denoting the thermal de Broglie wavelength and $\zeta$ denotes a partition function accounting for intramolecular molecular degrees of freedom of each fluid particle.

The equilibrium number and charge density profiles are obtained by solving the Euler–Lagrange (EL) equation

$$\rho_{\mathrm{eq}}(z) = \frac{\zeta}{\Lambda^3} \exp[-\beta V_{\mathrm{ext}}(z) + \beta\mu + c^{(1)}(z; [\rho_{\mathrm{eq}}, \beta\phi], T)],$$

and evaluating

$$n_{\mathrm{eq}}(z) = n^{(1)}(z; [\rho_{\mathrm{eq}}, \beta\phi], T),$$

with the one-body direct correlation functional $c^{(1)}$ and charge density functional $n^{(1)}$ defined as first functional derivatives of the excess intrinsic Helmholtz free energy functional

$$c^{(1)}(z; [\rho, \beta\phi], T) = -\frac{\delta\beta\mathcal{F}_{\mathrm{intr}}^{\mathrm{ex}}([\rho, \beta\phi], T)}{\delta\rho(z)},$$

$$n^{(1)}(z; [\rho, \beta\phi], T) = \frac{\delta\mathcal{F}_{\mathrm{intr}}^{\mathrm{ex}}([\rho, \beta\phi], T)}{\delta\beta\phi(z)}.$$

For fluids whose intermolecular interactions are long-ranged, such as ionic and dielectric fluids, the functional dependence of $c^{(1)}$ and

$n^{(1)}$ on $\rho$ and $\phi$ is, in general, non-local. Note that, as we only discuss these functionals evaluated at equilibrium in this work, we drop the "eq" subscript for notational convenience.

In order to learn these functional mappings, we applied the techniques introduced in refs. 34,36 where the local functionals of short-ranged mimic fluids, $c_R^{(1)}(z;[\rho,\beta\phi],T)$ and $n_R^{(1)}(z;[\rho,\beta\phi],T)$, were learned from molecular simulation data using the neural functional method[32]. A SR mimic fluid is defined as a system whose Coulomb interactions are replaced with

$$\frac{1}{r} \to \frac{\text{erfc}(\kappa r)}{r}.$$

For SPC/E water and the dipolar fluid, we used $\kappa^{-1} = 4.5$ Å. For the electrolyte, we used the functionals trained in ref. 34, with $\kappa^{-1} = 5.0$ Å. The effects of long-ranged interactions are accounted for in a well-controlled mean field fashion using local molecular field theory (LMFT). Implementation details and full derivations are provided in ref. 36 and Section S1 of the SI. We also validated the theory against computer simulations in Section S3 of the SI.

### Generation of training data

For the electrolyte, we employed the functional reported in ref. 34 based upon the restricted primitive model. For SPC/E water and the dipolar fluid, we generated training data using a combination of grand canonical Monte Carlo (GCMC) and molecular dynamics (MD) simulations to construct local reference functionals $c_R^{(1)}(z;[\rho,\beta\phi],T)$ and $n_R^{(1)}(z;[\rho,\beta\phi],T)$. For each fluid, we sampled ~ 2000 randomized external conditions spanning both subcritical and supercritical regimes. Random inhomogeneous electrostatic potentials of the form $\phi(z) = \phi_0 \cos(2\pi k z)$ were applied. In some cases, planar walls of the form of a 9-3 Lennard–Jones potential were also included. For each condition, GCMC (performed with our own code at https://github.com/annatbui/gcmc) determined the mean particle number, which was then used to initialize MD simulations in the NVT ensemble with LAMMPS[67], from which the number and charge density profiles are sampled. For water, we defined the molecular center for sampling $\rho(z)$ at the oxygen site, and for the dipolar fluid at the midpoint between the two opposite charges. The charge density was computed as $n(z) = \sum_i \sum_\alpha q_\alpha \langle \delta(z - z_{\alpha,i}) \rangle$, where the outer sum is over all molecules, and the inner sum is over all sites belonging to molecule $i$, with $z_{\alpha,i}$ and $q_\alpha$ indicating the $z$ coordinate and charge, respectively, of site $\alpha$. This hybrid scheme reproduces the accuracy of pure GCMC sampling while significantly accelerating convergence of inhomogeneous structures. The total computation time for the generation of the entire dataset is on the order of ~ $10^5$ CPU hours.

### Training neural functionals

For both SPC/E water and the dipolar fluid, we train two neural networks to represent $c_R^{(1)}([\rho,\beta\phi],T)$ and $n_R^{(1)}([\rho,\beta\phi],T)$ following the local learning strategy[32]. The machine learning routine was implemented in Keras/Tensorflow[68]. Inputs consisted of local density and electrostatic potential in a sliding spatial window of size 10 Å from the center of the position of interest. To effectively learn spatial variations, the model internally computes the gradient of $\beta\phi(z)$ using a central difference scheme. In addition to these spatially varying inputs, a separate input node encodes $T$ as a scalar. The full architecture details are provided in Figs. S5 and S6. The dataset was split roughly in a 3:1:1 ratio for training, validation, and test sets. Models were trained for 200 epochs with a batch size of 256, using an exponentially decaying learning rate starting at 0.001, achieving errors comparable to the estimated simulation noise. The training of the neural networks was done on a GPU (`NVIDIA GeForce RTX 3060`) in a few hours. We also verified that the trained functional can recover standard simulation results including bulk

equation of state (Fig. S7) the binodal at zero applied field (Fig. S8) as well as number density and charge density response (Fig. S9).

### Using the neural functionals

Evaluating the trained neural functionals is fast (~ milliseconds) and can be performed on a CPU or GPU. Combining with LMFT give us $c^{(1)}([\rho,\beta\phi],T)$ and $n^{(1)}([\rho,\beta\phi],T)$. The EL equation is solved self-consistently with a mixed Picard iteration scheme, which typically converges within minutes. To determine the liquid–vapor coexistence line at zero electric field, we calculate isotherms of the chemical potential as a function of the bulk density $\rho_b$ from

$$\beta\mu = \ln(\Lambda^3 \zeta^{-1} \rho_b) - c^{(1)}([\rho_b, \beta\phi = 0], T),$$

and perform a Maxwell construction to find the coexisting liquid and vapor densities at the binodal. For inhomogeneous systems, i.e, as a result of an applied non-uniform electric field, we set the chemical potential and temperature and find the inhomogeneous solutions by solving the EL equation. The isotherms of the chemical potential, now as a function of the mean density, $\bar{\rho} = L^{-1} \int_0^L dz\, \rho(z)$ where $L$ is the total length of the domain over which $\rho(z)$ is defined, now have distinct jumps when the system undergoes phase separation, giving liquid $\rho_l$ and vapor $\rho_v$ solutions. To distinguish stable solutions from metastable solutions, we also calculate the grand potential $\Omega_\phi = \Omega([\rho,\beta\phi],T)$ at a fixed $\beta\phi$ where the excess free energy term can be evaluated via functional line integration[32]. Performing this procedure for $\beta\phi = 0$ gives results consistent with those obtained by Maxwell construction. Note that the pressure of the bulk fluid at zero field is given by $-PV = \beta\Omega_0$. To investigate capillary condensation in slit pores, adsorption isotherms are obtained from the mean density $\bar{\rho} = H^{-1} \int_0^H dz\, \rho(z)$ where $H$ is the slit height. Hysteresis loops are mapped by seeding different initial guesses when solving the EL equation.

In the dielectrowetting calculations, the electrostatic free energy per contact area is given as

$$\gamma_{\text{elec}} = \frac{1}{2} \int_{-\infty}^{+\infty} dz\, \phi(z) n(z),$$

where the factor of a half comes from there being two symmetric confining walls. To construct the dependence of $\theta$ on $\phi_0$, we obtain $\alpha = -\partial\gamma_{\text{elec}}/\partial(\phi_0^2)$ and calculate $\gamma_{\text{lv}}$ from a direct coexistence molecular dynamics simulation. We estimate $\theta_0 = 160°$ based on the maximum value of the local compressibility at $\phi_0 = 0$ and comparing to ref. 69; while more detailed calculations can provide a more accurate estimate, this value is consistent with the solvophobic nature of the slit, and is sufficient for the purposes of demonstrating the increased wetting with $\phi_0$ shown in Fig. 4.

## Data availability

The data supporting the findings of this study and source data of the paper is openly available on Zenodo[70].

## Code availability

The code used to train the models in this study is available on Github (https://github.com/annatbui/dielectrocapillarity-cdft). Training data were generated using our in-house code for GCMC simulations available on Github (https://github.com/annatbui/gcmc) or Zenodo[71] and the LAMMPS code for MD simulations[67].

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

## Acknowledgements

Via membership of the UK's HEC Materials Chemistry Consortium funded by EPSRC (EP/X035859), this work used the ARCHER2 UK National Supercomputing Service. A.T.B. acknowledges funding from the Oppenheimer Fund and Peterhouse College, University of Cambridge. S.J.C. is a Royal Society University Research Fellow (Grant No. URF \R1\211144) at Durham University.

## Author contributions

A.T.B.: Conceptualization (equal); Investigation (equal); Writing—original draft (equal); Writing—review & editing (equal). S.J.C.: Conceptualization (equal); Investigation (equal); Writing—original draft (equal); Writing—review & editing (equal).

## Competing interests

The authors declare no competing interests.
