## [Transparent Peer Review file · Nature Communications]

Dielectrocapillarity for exquisite control of fluids

Corresponding Author: Dr Stephen Cox

Version 0:

Reviewer comments:

Reviewer #1

(Remarks to the Author)

Overall Evaluation

This manuscript introduces the novel concept of "dielectrocapillarity," demonstrating how electric field gradients (EFGs) can regulate phase behavior, capillary condensation, and adsorption of confined fluids. By integrating classical density functional theory (cDFT) with deep learning, the authors develop a multiscale framework that offers new strategies for applications in nanoporous energy storage, gas separation, and neuromorphic devices. The study addresses a critical gap in understanding electrostrictive effects in fluids, with innovative methodology and clear logical progression. While the theoretical framework and simulations provide robust support for key conclusions, the work could be strengthened by deeper experimental validation, comparative analysis with competing techniques, and expanded discussion of its limitations. The research holds significant promise for both fundamental science and applied technology, though further refinements are recommended to enhance its broader impact.

Detailed Comments

Limited Experimental Validation: The study relies predominantly on theoretical simulations and model derivations without direct experimental data (e.g., observations of dielectrocapillary effects in actual nanofluidic devices). The authors should supplement the manuscript with comparisons to existing experimental techniques (e.g., atomic force microscopy, dielectrophoresis experiments) or provide proof-of-concept experimental results to reinforce the credibility of the findings.

Incomplete Comparison with Traditional Electric Field Methods: While the manuscript distinguishes between uniform electric fields and EFGs, it lacks a systematic comparison of dielectrocapillarity with conventional techniques like electrowetting or dielectrophoresis in terms of control precision, energy consumption, and applicability. A clear articulation of the method's unique advantages and limitations is necessary.

Generalizability to Multicomponent Fluids: The current focus on single-component polar fluids limits relevance to real-world applications involving complex mixtures (e.g., electrolyte solutions, multicomponent gases). Simulations or discussions on extending the model to multicomponent systems would enhance practical utility.

Clarity of Graphical Information: Some curve annotations in Figures 1 and 2 (e.g., differences between SR/LR interactions) and definitions of non-dimensional units (e.g., reduced units in SI) are unclear. Additional explanations in figure captions or the main text would improve accessibility.

Underdeveloped Discussion of Application Scenarios: While applications in supercapacitors and neuromorphic devices are mentioned, linking the findings to specific case studies (e.g., performance improvements in existing nanoporous materials) would more vividly demonstrate translational potential.

Applicability of Theoretical Assumptions: The model's reliance on simplified assumptions (hard-sphere repulsion, van der Waals attraction) raises questions about its validity for systems with strong hydrogen bonding (e.g., water) or complex interfacial chemistry. A sensitivity analysis or discussion of these limitations is recommended.

Conclusion

This work establishes a groundbreaking framework for fluid manipulation, with the concept of dielectrocapillarity offering significant promise for nanofluidics and energy technologies. Despite gaps in experimental validation and application-level analysis, the originality of the concept and methodological advancements are compelling. Addressing the above comments—particularly by incorporating experimental data, enhancing comparative analyses, and refining graphical clarity—would strengthen the manuscript's scientific rigor and broader appeal. With these improvements, the study would solidly position itself as a high-impact contribution to Nature Communications.

Reviewer #2

(Remarks to the Author)

This manuscript presents a numerical investigation of the properties of polar fluids under electric gradients. The starting point is to use dielectrophoresis, a phenomenon under which the polar fluids undergoes local forces in an electric gradient. The central claim is that such forces could be used to compact or dilate the fluids. The manuscript explores the possibility of using these compaction/dilatation forces to tune the liquid-vapor phase diagram of the fluid and then to use that to change wetting properties inside confined systems. In essence, the idea is to explore how dielectrophoresis might modify capillary effects in a similar way as electrophoresis modifies capillary forces.

The idea itself is interesting and creative. Dielectrophoresis being quite subtle, I have no doubt that peculiar phenomena might arise in specific systems, so dielectrocapillary effects are indeed a possibility – that has not been studied so far. The manuscript also includes an investigation of screening of the dielectrophoretic interactions which is captivating too.

At this stage, however, the model makes 2 major assumptions which are not well justified and, therefore, make it hard to validate whether or not the effect might be real. First, an equilibrium AI model is used to study nonequilibrium phenomena. Second, capillarity related to properties near confining surfaces however the response of the surfaces to the electrical gradients are not accounted for and may strongly tune the response – as is the case for electrocapillarity. Another main point is that it is not clear that the orders of magnitude of changes investigated on a minimal polar fluid model might indeed be significant and hence accessible in experiments. This is not made easier by the current style of the manuscript which sometimes makes unjustified claims. At this stage I rather see the model as a remarkable and creative hypothesis, but its potential applicability in realistic systems is not discussed at all, making it hard to see if it could be real.

I have several detailed comments, that I put below.

(1) What goes inside the model and what data the AI is trained on is quite opaque, at least from the main text. Is it trained on the different fluids mentioned or on other fluids? How is the functional learned exactly, aka what is the training process? (related to one of my main question) What makes us sure that a model that was designed and trained at equilibrium and is potentially suitable to investigate driving by uniform fields and then suitable to investigate drivings by non uniform fields? Are some standard results in the field recovered? At this stage it is still not clear to me how the AI training enables one to study the system faster/better so it would be nice to discuss potential limitations/advantages a bit more.

(2) The electrophoretic rise part could be described and explained more carefully. At this stage, I still don't understand it. I believe this is mainly caused by an absence of details. First, in which direction is the field E imposed? Along the z direction? To me it appears that regions of higher field actually have a locally neutral ion concentration and so there should be no force on the fluid in these regions, yet this is apparently not what is observed. I can not make sense of the schematic drawing because it is not clear where the periodic unit is taken along the axis and what the line represents. Eventually I find it would be worthwhile to discuss a little bit how, already at this stage, working in the μVT ensemble instead of NVT leads to interesting differences compared to more traditional simulations setups where the total density varies spatially a lot. This is still surprising to me since I expect fluid to be more or less incompressible and so homogeneous over such lengthscales. Actually, the relationship between the densities ρ^* and n^* is not clear from the images or the caption. Maybe one should consider putting in that information maybe twice. Usually n is used for numbers and so would be used instead of ρ , so that convention was puzzling.

(3) It is equally hard to understand the dielectrophoretic rise for similar reasons. What is n^* in the case of the polar fluid? The fluid should be locally neutral.

(4) I do not understand how the phase diagram is constructed – this is information that is completely in the SI but some short explanation would be especially useful. Would it be possible to give some actual illustration of what the separated phases would look like? Do they really have the wiggly interfaces that are shown on the sketches? Because then this creates a surface tension in the fluid that is much bigger than that for a flat interface; so there's a price to pay for this compacted state... which makes the story a lot more complicated.

(5) In the confinement case there is a problem of dynamic entrance effects. There may be a question of how fast you are indeed able to recruit more particles within the small opening to contract the fluid etc.

(6) Often electrical gradients are constructed by using specific/man made surface properties, e.g. patterning electrodes (as e.g. in <https://pubs.acs.org/doi/full/10.1021/acs.jpcclett.4c00278>) or others. So there is a deep question there of how the electric gradient is made and how that couples to the fluid properties which was somewhat discarded.

Potential improvement of clarity/wording/typos:

(12) : of a fluid or of a polar fluid?

(general) Electrostrictive is not such a common word so I would suggest defining it.

(introduction) I do not see how phenomena of gas uptake and separation can be so easily put under the same hat as liquid uptake since the electrokinetic properties in such dilute phases may be vastly different.

(68) "grand canonical ensemble, which is most representative of real conditions in which fluid molecules can dynamically enter and leave a pore" It is and it is not. As far as I can tell, there is no great way of simplifying the entrance/exits from a pore, because the grand canonical ensemble will remove correlations between particles exiting and coming back, which are correlations that exist. Also, all these things are nonequilibrium so really we don't know. So the grand canonical ensemble is

a way of representing things, that may have some advantages compared to others, but it is likely not the best simplification of the system.

(94) "such minimal models uncover universal behaviors that emerge across different scales, from molecular liquids to colloidal systems." -> this is a bold claim that is not justified.

(112) twice the word "directly" and "supervise-learn" is an unusual concatenation

(169) "dielectrophoretic coupling depends on the EFG rather than the absolute field strength". They depend both on the pattern on the field strength.

In Fig. 1d, the color code is not given.

Literature:

In terms of fluid uptake and filling of pores, there is a body of literature related to tailoring wettability properties that could be discussed (see below). Diffusiophoresis has already been used as an externally tunable force to manipulate filling or unfilling of nanopores with imposed chemical gradients (so indeed there already exists some external manipulation – although not as easy as electric fields) to manipulate particles.

On the filling of nanopores:

- Giacomello, A., Chinappi, M., Meloni, S., & Casciola, C. M. (2012). Metastable Wetting on Superhydrophobic Surfaces: Continuum and Atomistic Views of the Cassie-Baxter–Wenzel Transition. *Physical review letters*, 109(22), 226102.

- Picard, C., Gérard, V., Michel, L., Cattoën, X., & Charlaix, E. (2021). Dynamics of heterogeneous wetting in periodic hybrid nanopores. *The Journal of Chemical Physics*, 154(16).

Manipulating concentration gradients to fill/unfill dead-end pores

- Migacz, R. E., Castleberry, M., & Ault, J. T. (2024). Enhanced diffusiophoresis in dead-end pores with time-dependent boundary solute concentration. *Physical Review Fluids*, 9(4), 044203.

- Marbach, S., & Bocquet, L. (2019). Osmosis, from molecular insights to large-scale applications. *Chemical Society Reviews*, 48(11), 3102-3144.

- Shin, S., Um, E., Sabass, B., Ault, J. T., Rahimi, M., Warren, P. B., & Stone, H. A. (2016). Size-dependent control of colloid transport via solute gradients in dead-end channels. *Proceedings of the National Academy of Sciences*, 113(2), 257-261.

- Shin, S., Ault, J. T., Warren, P. B., & Stone, H. A. (2017). Accumulation of colloidal particles in flow junctions induced by fluid flow and diffusiophoresis. *Physical Review X*, 7(4), 041038.

Version 1:

Reviewer comments:

Reviewer #1

(Remarks to the Author)

The revision is thorough and comprehensive, so my recommendation is to accept this manuscript for publication in *Nature Communications* in its present form.

Reviewer #2

(Remarks to the Author)

I thank the authors for carefully addressing my comments. I think adding in water and contact angles were good ideas that make the results a bit easier to put in experimental perspective. The authors have also made significant efforts in making their manuscript more pedagogical.

I would still like to raise a minor comment: In my view, "out-of-equilibrium" means that a source of energy is put in the system, in this case, it is the spatially varying electric field. So I was quite worried that the AI had been trained on data that was never under applied external fields. Indeed, that was not the case, as the authors explain in the methods. To me that is something important that enhances the validity (a priori) of the trained model and I would therefore encourage the authors to say that explicitly in the main text.

I would also like to congratulate the authors on so diligently (and also sensibly) answering both reviewers.

Apart from my minor comment above, I am happy to recommend this manuscript for publication in *Nature Comm*.

Reviewer #3

(Remarks to the Author)

The manuscript deals with the topic of how electrical field gradients induce novel phenomena relating to phase behaviour, capillarity and adsorption in liquids. Focus of the work is in confined conditions and could have implications on porous materials and their use for energy storage applications etc.

The manuscript is well structured and provides a well thought of theoretical study. The authors have already addressed most open questions, and the manuscript stands on its own.

My concerns are in regards to not being able to come up or understand an experiment to verify this theory (falsifiability).

The example given in Fig. 4 is too simple and does not provide any sort of evidence that the theory is valid except showing that it fits a specific boundary condition. Due to the use of cDFT combined with machine learning methods, the approach seems like a black box and it is hard to understand its core assumptions (e.g. the choice of potentials at the start) and their validity. I would recommend publishing in a more specialised journal, and focusing future work to include geometries and conditions where it could be experimentally tested. I would recommend considering the impacts of the theory in also other geometries, e.g. nanopores and their use for sensing and filtration.

Reviewer #4

(Remarks to the Author)

In this article, the authors use ML-based numerical methods to study a novel phenomenon, dielectrocapillarity, where a spatially-varying electric field is used to modulate the density of a dipolar fluid. More precisely, they use a neuron network to learn thermodynamic functionals from MD simulations, and then use these functionals to (1) estimate the density profile of a bulk liquid under a electric field gradient, notably with consequences on the liquid-gas phase transition, (2) fluid entry into a small pore and (3) model electrowetting.

The effects reported here are definitely novel and promising, both in terms of technique and in relevance with experiments. My main concerns, as detailed below, regard (1) the clarity of certain aspects of the work (notably figures) and (2) the highly idealized settings considered here, which contrast with the bold claims throughout the article. Overall, though, I support publication of this work in Nature Communications.

Major comments:

1) I find it hard to understand the general procedure behind this work, in particular with what relates to geometry, which is only detailed in SI file (note a typo p8 of SI: " $L_x = L_z = 80 \text{ \AA}$ " should, I imagine, read " $L_x = L_y = 80 \text{ \AA}$ "). If I understand correctly, the authors generate MD trajectories with a slab geometry ($L_z \ll L_x, L_y$), from which they compute thermodynamic functionals, which are then learnt by a neuron network. How generalizable are the results to other geometries? The electrowetting proof-of-concept shows that changing the shape of the gradient is OK; but could, for example, be the learnt functionals be used to cases where the electric field gradient depends on more than one variable? Or non-slab geometries?

2) Related to the previous point, the authors state that their method greatly reduces computational time, by comparing a timescale to evaluate the NN to an MD simulation time. This is trivial, since what matters here should instead be the total computational time (ie including training and testing), unless the neural functional can be used directly to cases with, for example, completely different geometries. Can the authors comment on that?

3) I am relatively unconvinced by the type of electric field gradients that are considered here. While the numerical results seem fine and interesting, I doubt they really correspond to something technically achievable. They use molecular-scale gradients (wavelength comparable to the molecular size σ) while comparing to micron-scale experiments (local or interdigitated electrodes, for example). That of course does not mean that considering short wavelengths is wrong, but I doubt the effect survives under molecular confinement as a surface provides interactions with liquid other than through a static electric field (adsorption, chemical defects, roughness, dielectric confinement, polarizability, etc.). In addition, under such confinement, I would expect the gradient to be along the surface (because of natural/engineered defects, for example), rather than normal to it; I do not see any way of replicating experimentally the situation of Fig 1 and 2.

4) Similarly, for Fig 3-4 I would expect that chemical effects (eg acid-base reactions with the wall) play a much more important role at the considered lengthscales, and that modelling the walls with a static external electric field is plain wrong. Another thing that comes to my mind is that for this kind of questions (wettability and fluid filling of a small pore), the MD simulations used as training data are typically quite bad - it is for example very hard to correctly estimate a contact angle from simulations of water+LJ walls, and in nanometric slit, slightly changing the LJ parameters of the wall can result in either full wetting or dewetting, at odds with experimental results eg on carbon slits. Here, the authors choose one particular value of the walls LJ coefficients, so I would be very cautious about the results, which could be of second-order magnitude compared to changing interaction parameters.

5) Some of the results discussed here are in the "large length scale regime", eg Janus colloids. I am afraid this limit is quite a dangerous one to take: colloids under an applied field are basically never at thermal equilibrium (because of electrophoresis, for example), so the current framework would not apply, even if it was extended to include dynamical effects (one would need to add hydrodynamics, ions, etc.). While the thought experiment presented in lines 208-226 is conceptually interesting, it is important to signal such limitations.

Minor/presentation remarks

6) Line 28: "Fundamentally, fluid uptake in these systems is driven by capillary forces"
Not really. Separation and filtration processes are typically out of equilibrium, with (electro/diffusio)osmotic effects often playing a major role.

7) Line 40: "While a uniform field exerts a force only on free charges such as ions"
This is misleading; fundamentally this force is cancelled and fully transmitted to the solvent because of viscous dissipation.

In almost all nanofluidic experiments, applying a uniform electric field results in a net flow of water (because of electro-osmosis and related phenomena). In typical slit-like geometries like the one considered here, the effect is often dominant compared to other transport phenomena. See for example the 2021 review by Kavokine, Netz and Bocquet.

8) I find Fig 2 difficult to follow due to the high number of curves, colors, line styles, etc. In particular, I do not fully understand Fig 2d. What is represented as inset? What are the two blue and red rectangles on the right?

9) For all figures, some of the 'pastel' colors are difficult to see after printing.

10) Line 322: "That is, EFGs impact criticality of polar fluids in both the three-dimensional and confined two-dimensional Ising universality classes" I do not understand this sentence, I would suggest explaining more. I guess that the authors are referring to the fact that the liquid-gas phase transition corresponds to the 3D Ising class; but I am less familiar with the capillary transition and I would imagine that its does not belong to a single universality class but rather would depend on eg. the pore's geometry. In practice, you would likely need to have a wedge/1D/0D geometry to have a localized electric field in the confined system, so I do not think it is realistic to have the exact 2D regime.

I also do not understand what this discussion is bringing here; if the observed effect modifies the system's chemical potential, it is obvious that it will affect both phase transitions, which are defined from it... It is not particularly surprising that modifying the system's parameters shifts the critical points; what would be more relevant would be to discuss orders of magnitude, like the 50K shift in the case of the liquid-gas transition of water. In addition to this, the authors could discuss what orders of magnitude can be reached in terms of electric field (gradient) experimentally. They list a value of $E = 0.4 \text{ V/\AA}$. To get this kind of field from say a point charge of charge Zq - say surface defects - in water over 1 nm (which corresponds roughly to the setting considered here), you would need $Z = 10$, which is a bit wild.

11) Related to the previous point, it seems to me that the only two relevant regimes for how to create an electric field gradient within a confined pore are (1) electric field gradient occurring on a molecular lengthscale close to the walls or (2) a gradient with a lengthscale equal to the system size. The authors do not really discuss the impact of the system size in the settings corresponding to Fig 3.

Version 2:

Reviewer comments:

Reviewer #2

(Remarks to the Author)

The authors have addressed all my comments and I recommend publication in this form.

Reviewer #3

(Remarks to the Author)

I am satisfied with the answer provided by the authors. It addresses my concerns sufficiently to allow publishing in Nature Comms.

Reviewer #4

(Remarks to the Author)

The authors addressed all my concerns. I support publication in the present form.

Response to reviewer comments for NCOMMS-25-37255-T “Dielectrocapillarity for exquisite control of fluids”

Anna T. Bui^{1,2} and Stephen J. Cox^{2, a)}

¹⁾Yusuf Hamied Department of Chemistry, University of Cambridge, Lensfield Road, Cambridge, CB2 1EW, United Kingdom

²⁾Department of Chemistry, Durham University, South Road, Durham, DH1 3LE, United Kingdom

(Dated: September 26, 2025)

We would like to thank the editors and reviewers for their time in reading our manuscript. While both reviewers are generally positive, finding our work *interesting*, *innovative* and *addresses a critical gap*, they also raised important points regarding: (i) assumptions underlying our theoretical models, (ii) the generality of our approach, and (iii) connections to experiments. In response, we have carried out additional analyses and revisions to fully address these concerns.

Below we detail our responses to all comments from the reviewers point-by-point. The reviewers' comments are **reproduced in turquoise**. Changes are **highlighted in magenta**. The reference list from the main paper is replicated in full at the end the document; this is simply so citations in text quoted from the manuscript appear correctly.

^{a)}Electronic mail: stephen.j.cox@durham.ac.uk

Report of Reviewer 1

Overall Evaluation

This manuscript introduces the novel concept of “dielectrocapillarity,” demonstrating how electric field gradients (EFGs) can regulate phase behavior, capillary condensation, and adsorption of confined fluids. By integrating classical density functional theory (cDFT) with deep learning, the authors develop a multiscale framework that offers new strategies for applications in nanoporous energy storage, gas separation, and neuromorphic devices. The study addresses a critical gap in understanding electrostrictive effects in fluids, with innovative methodology and clear logical progression. While the theoretical framework and simulations provide robust support for key conclusions, the work could be strengthened by deeper experimental validation, comparative analysis with competing techniques, and expanded discussion of its limitations. The research holds significant promise for both fundamental science and applied technology, though further refinements are recommended to enhance its broader impact.

We thank the reviewer for this positive and thoughtful evaluation. We appreciate their recognition of the novelty and potential impact of our work, and we agree with their constructive suggestions. Below, we provide a detailed response to the specific points raised.

Detailed Comments

Limited Experimental Validation: The study relies predominantly on theoretical simulations and model derivations without direct experimental data (e.g., observations of dielectrocapillary effects in actual nanofluidic devices). The authors should supplement the manuscript with comparisons to existing experimental techniques (e.g., atomic force microscopy, dielectrophoresis experiments) or provide proof-of-concept experimental results to reinforce the credibility of the findings.

We are sympathetic to the reviewer’s criticism that our work is primarily computational and theoretical, focusing on nanoscale effects where direct experimental implementation remains challenging. We are actively collaborating with experimental colleagues using surface force apparatus, however, such nanoscale experiments are extremely challenging and lie beyond the scope of the current study. We can, however, link directly to macroscopic experiments which not only give confidence on the faithfulness of our microscopic approach, but also provide indirect support for our findings. In particular, dielectrowetting studies have demonstrated that the contact angle of dielectric liquids can be tuned by applying well-controlled inhomogeneous electric fields

- G. McHale, C. V. Brown, M. I. Newton, G. G. Wells, and N. Sampara, Dielectrowetting driven spreading of droplets, *Phys. Rev. Lett.* 107, 186101 (2011)
- G. McHale, C. V. Brown, and N. Sampara, Voltage-induced spreading and superspreading of liquids, *Nat. Commun* 4, 1605 (2013)

These experiments demonstrate that dielectrophoretic forces can drive droplet spreading and the contact angle observed obeys certain phenomenological scaling laws. To further establish the connection between our theoretical findings and experimental reality, we performed additional analyses using exponentially decaying fields analogous to those generated by interdigitated electrodes in the above experiments. This not only allows us to see whether we observe the same behavior between our microscopic approach and the macroscopic experiments, but also

the extent to which macroscopic theory can be expected to hold at the nanoscale. Consistent with experiment and macroscopic theory, our results show enhanced wetting as the voltage is increased, with minor differences observed at low voltages – this we can rationalize in terms of suppressed density fluctuations at the interface (an effect omitted in macroscopic approaches). Importantly, these observations are consistent with a downward shift of T_c under the application of EFGs, which is also the basis of dielectrocapillarity.

We have made substantial changes to the manuscript in light of this comment – we have added a new section, **Connection to dielectrowetting experiments**, and a new **Figure 4**, which demonstrate how our models capture these effects from first principles and provide a direct microscopic foundation for the phenomenological laws observed in experiments:

Figure 4: Connecting to dielectrowetting experiments. The electrostatic potential from interdigitated electrodes, shown schematically in (a), decays exponentially. Applying this potential symmetrically from both confining walls in a slit geometry with $H \approx 16\sigma$ enhances wetting of the solid-liquid interface, as can be seen in the density profiles (left) and changes in local compressibility (right) in (b) (both quantities are normalized by their bulk values). (c) Electrostatic free energy per unit area from cDFT (inset) exhibits a quadratic dependence on ϕ_0 , enabling reconstruction of the contact angle via Eq. 1 (assuming $\theta_0 = 160^\circ$ and using $\gamma_{lv} = 0.025 \text{ J m}^{-2}$ computed from direct coexistence simulation), in direct analogy with dielectrowetting experiments^{14,50}.

(Lines 339–387)

Connecting to dielectrowetting experiments

Thus far we have introduced dielectrocapillarity as an additional mechanism of controlling capillary phenomena, complementing electrocapillarity⁵¹ where electrolytes respond

to applied potentials. On the macroscopic scale, these effects manifest in electrowetting⁵² and dielectrowetting^{14,50}, where the contact angle of a droplet can be tuned with applied potentials. In the case of dielectrowetting, a typical experimental setup employs interdigitated electrodes, as shown schematically in Fig. 4a, which generate an exponentially decaying electrostatic potential $\phi(z) = \phi_0 \exp(-z/d)$ that drives a droplet of dielectric fluid to spread. The resulting contact angle can be described by a modified Young's law^{14,50},

$$\cos \theta(\phi_0) = \cos \theta_0 + \frac{\alpha}{\gamma_{lv}} \phi_0^2, \quad (1)$$

where θ_0 is the Young's contact angle at zero field, γ_{lv} the liquid–vapor surface tension, and α a material parameter related to the dielectric response of the liquid. This relationship assumes that the electrostatic energy stored in the liquid droplet is well-described by dielectric continuum theory, and that d is sufficiently small so that any changes in energy due to the electric field are effectively localized to the solid-liquid interface. Under these assumptions, the electrostatic free energy per unit area obeys the simple quadratic scaling $\gamma_{elec} = -\alpha \phi_0^2$.

Our multiscale framework provides a microscopic perspective on this phenomenology, and allows us to test whether, at the nanoscale, EFGs indeed enhance the wetting of dielectric liquids and the extent to which the scaling prescribed by Eq. 1 holds. To this end, we applied an exponentially decaying electrostatic potential with $d = 1.1 \sigma$ to the confined dipolar model, symmetrically from both walls of a solvophobic slit. While not trained on such electrostatic potentials, as can be seen in Fig. 4b, the neural functional extrapolates well, yielding physically plausible results, aside from some oscillations at high wetting that are likely minor artifacts. As ϕ_0 increases, so too does the contact density at the wall, verifying that enhanced wetting occurs. We quantify this effect by computing $\gamma_{elec} = \frac{1}{2} \int_{-\infty}^{\infty} dz \phi(z) n(z)$, which, as can be seen in Fig. 4c, decreases in an approximately quadratic fashion as ϕ_0 increases. By identifying $\alpha = -\partial \gamma_{elec} / \partial (\phi_0^2)$, we reconstruct the potential-dependent contact angle, as shown in Fig. 4c. While the results of our microscopic theory are broadly in line with the macroscopic model (Eq. 1) some subtle differences are observed, especially at small ϕ_0 . These appear to be correlated with significant changes in the local compressibility near the interface, $\chi_\mu(z) = \partial \rho(z) / \partial \mu$, as ϕ_0 increases, which indicates a suppression of density fluctuations. Such microscopic details are lacking in the dielectric continuum model that underpins Eq. 1.

Incomplete Comparison with Traditional Electric Field Methods: While the manuscript distinguishes between uniform electric fields and EFGs, it lacks a systematic comparison of dielectrocapillarity with conventional techniques like electrowetting or dielectrophoresis in terms of control precision, energy consumption, and applicability. A clear articulation of the methods unique advantages and limitations is necessary.

We thank the reviewer for highlighting the need to compare the phenomenon we propose with traditional electric-field-based fluid manipulation methods. Dielectrocapillarity can be viewed as a parallel to electrocapillarity: for dielectric (polar) fluids, it arises from coupling with electric field gradients, while for conducting (ionic) fluids, electrocapillarity arises from coupling with the field itself. Thus, while they are both capillary phenomena, they apply to distinctly different systems (dielectrics and conductors) – beyond this fact, a systematic comparison in terms of control precision, energy consumption and applicability challenging.

Similar reasoning applies when trying to contrast electrowetting vs dielectrowetting, and further, dielectrowetting vs dielectrocapillarity. Indeed, as we make clear in our response to the

previous comment, dielectrocapillarity and dielectrowetting should be seen as complementary phenomena.

Regarding dielectrophoresis, this is a nonequilibrium effect that, although governed by the same dielectrophoretic forces, we cannot yet tackle with our framework.

In light of this comment, we have changed the manuscript to convey that dielectrocapillarity is a complementary technique to the more common approaches mentioned by the reviewer:

(Lines 340–344) ...Thus far we have introduced dielectrocapillarity as an additional mechanism of controlling capillary phenomena, complementing electrocapillarity⁵¹ where electrolytes respond to applied potentials. On the macroscopic scale, these effects manifest in electrowetting⁵² and dielectrowetting^{14,50}...

We also highlight the limitations regarding, and possible extensions to, non-equilibrium phenomena in the Conclusions:

(Lines 402–413) The effects uncovered in this work concern the equilibrium behavior of dielectric liquids, and omit potentially important nonequilibrium effects such as pore entry and exit^{53–55}, electrokinetic phenomena^{56–58}, and controlled wetting dynamics such as rate-dependent droplet spreading⁵⁰. Nonetheless, the implications of our results for nonequilibrium behavior are potentially far-reaching. A natural possible progression from this work is to augment our first-principles framework for electromechanics with dynamical extensions of cDFT⁵⁹, opening a promising route toward a microscopic understanding of how EFGs impact non-equilibrium processes....

Generalizability to Multicomponent Fluids: The current focus on single-component polar fluids limits relevance to real-world applications involving complex mixtures (e.g., electrolyte solutions, multicomponent gases). Simulations or discussions on extending the model to multicomponent systems would enhance practical utility.

We thank the reviewer for bringing up this important point.

Indeed, experiments have demonstrated liquid–liquid phase separation in binary mixtures under strong electric field gradients:

- Y. Tsori, F. Tournilhac, and L. Leibler, Demixing in simple fluids induced by electric field gradients, *Nature* 430, 544 (2004)
- Y. Tsori and L. Leibler, Phase-separation in ion-containing mixtures in electric fields, *Proc. Natl. Acad. Sci. U.S.A* 104, 7348 (2007)

highlighting the relevance of multicomponent systems in this context. While our study focuses on the liquid–vapor transition of single-component polar fluids, the theoretical framework is general and naturally extends to multicomponent mixtures (indeed we already included results for a simple electrolyte solution with implicit solvation).

We now make it clearer when describing the theory that it is readily generalizable to mixtures (and that we use it in the present study).

(Lines 107–111) ...Where comparison with ionic fluids is made, we will also show results for a prototypical model comprising oppositely charged hard spheres²⁴; in this case we use a straightforward generalization of cDFT to multicomponent systems.

In the conclusions, we also include a more detailed discussion on the potential to apply our approach to mixtures of dielectric fluids, citing the above papers for context.

(Lines 413–420) ...Moreover, the current framework naturally accommodates mixtures of dielectric liquids – in such cases, the excess intrinsic free energy, $\mathcal{F}_{\text{intr}}^{\text{ex}}(\{\rho_\nu\}, \beta\phi, T)$ acquires a functional dependence on the density fields of all species present, with ν indexing each component. This generalization opens the door to investigating more intricate phase behavior and interfacial phenomena—including liquid–liquid phase separation driven by EFGs^{12,60}.

Clarity of Graphical Information: Some curve annotations in Figures 1 and 2 (e.g., differences between SR/LR interactions) and definitions of non-dimensional units (e.g., reduced units in SI) are unclear. Additional explanations in figure captions or the main text would improve accessibility.

To improve accessibility, we have added detailed information in both the figure captions and the main text:

(Caption to Figure 1) Reorganization of fluids under non-uniform electric fields. An applied electric field, $E^*(z) = E_{\text{max}}^* \sin(2\pi z/\lambda)$...Reduced units are described in the Methods section.

(Lines 151–153) ...Quantities labeled with an asterisk are expressed in reduced units, defined in the Methods section.

(Lines 462–469) Reduced units. When reported, reduced units are defined by $T^* = k_{\text{B}}T/\varepsilon$, $\rho^* = \rho\sigma^3$, and $E^* = E\sigma^{3/2}\varepsilon^{-1/2}$, where σ sets the molecular length scale and ε the energy scale. Specifically for each fluid: (i) SPC/E water, $\sigma = 3.166 \text{ \AA}$, $\varepsilon = 0.65 \text{ kJ mol}^{-1}$; the dipolar fluid, $\sigma = 3.024 \text{ \AA}$, $\varepsilon = 1.87 \text{ kJ mol}^{-1}$; and (iii) the electrolyte, $\sigma = 2.76 \text{ \AA}$, $\varepsilon = e^2/\sigma$, where e is the elementary charge.

(Caption to Figure 2) Controlling liquid–vapor equilibrium with EFGs...Solid and dashed lines show the response of systems with dipolar interactions that are long-ranged (LR), e.g., polar molecules, and screened short-ranged (SR), e.g., colloids, respectively. The effect of LR interactions becomes pronounced for $\lambda \gg \sigma$. **The SR fluid is a nearly identical dipolar fluid, but whose Coulomb potential is replaced by $\text{erfc}(\kappa r)/r$ where $\kappa^{-1} = 1.5 \sigma$...**

Underdeveloped Discussion of Application Scenarios: While applications in supercapacitors and neuromorphic devices are mentioned, linking the findings to specific case studies (e.g., performance improvements in existing nanoporous materials) would more vividly demonstrate translational potential.

We thank the reviewer for this comment, but wish to reiterate that our focus in this work is on establishing a fundamental framework for understanding how electric field gradients manipulate fluids. To the best of our knowledge, dielectrocapillarity is a new, unreported phenomena – it is too early to give a detailed specific case study on device performance. However, we note that

we now directly link our results to dielectrowetting experiments, which helps to link to potential advances in device design based on this physics. We therefore expand our discussion in the revised manuscript to connect with potential applications in this area. In the Conclusions:

(Lines 388–398) Our findings establish EFGs as a powerful and versatile tool for manipulating fluids. We have revealed their ability to structure fluids, modulate phase transitions, and control capillary effects. Crucially, we demonstrate that EFGs not only influence a fluid's behavior in bulk, but also give rise to dielectrocapillarity, a new phenomenon in which capillary condensation and criticality under confinement can be finely tuned. By placing this nanoscale physics in direct correspondence with macroscopic dielectrowetting experiments, our work provides a microscopic foundation for the design of EFG-controlled wetting and adsorption phenomena.

(Lines 421–432) The ability to reversibly control phase behavior and adsorption with electric fields unlocks new avenues for manipulating fluids across multiple length scales, from adaptive nanofluidic devices, to tunable sorption in porous materials, to colloidal assembly. At the nanoscale, nuclear magnetic resonance techniques³ can directly validate these effects. With strong EFGs experimentally accessible via atomic force microscope tips, optical tweezers, and patterned electrode configurations, our results lay the foundation for future experimental exploration, paving the way for new strategies in developing energy storage, selective separation, and responsive fluidic technologies.

Applicability of Theoretical Assumptions: The models reliance on simplified assumptions (hard-sphere repulsion, van der Waals attraction) raises questions about its validity for systems with strong hydrogen bonding (e.g., water) or complex interfacial chemistry. A sensitivity analysis or discussion of these limitations is recommended.

We thank the reviewer for raising this important point that, which is also raised by Reviewer 2. To assess the robustness of our conclusions beyond simplified dipolar models, we have repeated our analysis for SPC/E water. The results confirm our main findings: dielectrophoretic rise is observed at supercritical temperature, and the critical temperature shifts downward with increasing EFG strength. Importantly, water also reveals additional features due to its intrinsic charge asymmetry, which we now highlight and discuss. These new results are incorporated throughout the revised manuscript and provided in detail in the SI.

(Lines 84–94) To investigate how EFGs influence polar fluids, we primarily consider a minimal molecular model that incorporates soft-core repulsion, van der Waals attraction, and long-range dipolar interactions. The advantage of using such a simple molecular model is that we can exhaustively explore a broad range of thermodynamic conditions, while potentially uncovering common behaviors among polar fluids, from molecular liquids to colloidal systems. In addition, we also investigate a commonly used simple point charge model for water (SPC/E²¹) that explicitly incorporates hydrogen-bonding, under thermodynamic conditions close to its critical point.

(Lines 148–151) ...As a starting point, we characterize the bulk response of water, the simple polar fluid, and the electrolyte, all under supercritical conditions, when subjected to a sinusoidal electric field, $E^*(z) = E_{\max}^* \sin(2\pi z/\lambda)$, as shown in Fig. 1a.

Figure 1: Reorganization of fluids under non-uniform electric fields. An applied electric field, $E^*(z) = E_{\max}^* \sin(2\pi z/\lambda)$, shown in (a), induces pronounced density variations in bulk supercritical water, as can clearly be seen from a snapshot of a molecular dynamics simulation. (b) Results from cDFT for the number ($\rho^*(z)$, top) and charge ($n^*(z)$, bottom) densities capture this behavior. It can clearly be seen that number density is locally depleted where the $|\nabla E|$ is large, and locally enhanced where $|\nabla E|$ is small. The same qualitative behavior is seen in (c) for a supercritical dipolar fluid, except that its response is symmetric, in contrast to water where local depletion depends upon the sign of ∇E . (d) In contrast to both water and the dipolar fluid, an electrolyte is locally depleted in regions of low field strength due to electrophoretic forces (purple and green lines show cation and anion density, respectively, while the blue line shows the total density). Reduced units are described in the Methods section.

Figure S13: Dielectrophoretic rise in water under supercritical condition. The left panel shows the density response of the simple dipolar fluid (500 K, same data as in main manuscript) while the the right panel shows that for SPC/E water (700 K).

(Lines 154–174) In the case of water, Fig. 1b, we see that the applied electric field induces a significant structural reorganization along the field direction; its average density profile $\rho^*(z)$ is locally depleted in regions of weaker field strength, while molecular reori-

Figure S14: Water liquid–vapor equilibrium under EFGs. The intensity of the field E_{\max} is labeled. The bottom panel shows the density response as the chemical potential is varied continuously.

entation leads to an inhomogeneous average charge density distribution $n^*(z)$. As water is overall neutral and therefore experiences no net electrophoretic force, the observed local reorganization arises instead from dielectrophoretic forces, $f_{\text{DEP}} \sim \nabla E^2$ ^{33,34}, which push the fluid towards regions of higher electric field strength—an effect termed “dielectrophoretic rise.” This dielectrophoretic force is the same that drives dielectrophoresis, which is widely exploited to manipulate biological cells³⁵ and colloids³⁶, but its role in molecular fluids has received little attention. While water provides an important example of a polar fluid, the observed effects are by no means specific to aqueous systems. As seen in Fig. 1c, the overall picture is the same for the simple polar fluid, aside from the fact that it exhibits a symmetric response, whereas for water, depletion is stronger in regions where $\nabla E < 0$ than where $\nabla E > 0$ due to the inherent charge asymmetry of the water molecule.

(Lines 186–191) To illustrate this, we present in Fig. 2a how dielectrophoretic rise can be amplified by controlling the applied field. Owing to their qualitatively similar behavior, in the remainder of the article we focus on the dipolar fluid, with results for water given in the SI, Figs. S13 & S14.

(Lines 263–267) For water, we observe similar behavior, with a downward shift in T_c of approx. 50 K at $E_{\max} = 0.4 \text{ V } \text{Å}^{-1}$ (Fig. S14). These observations highlight the exquisite level of control that one can exert over dielectric fluids with EFGs.

Conclusion

This work establishes a groundbreaking framework for fluid manipulation, with the concept of dielectrocapillarity offering significant promise for nanofluidics and energy technologies. Despite gaps in experimental validation and application-level analysis, the originality of the concept and methodological advancements are compelling. Addressing the above comments particularly by

incorporating experimental data, enhancing comparative analyses, and refining graphical clarity would strengthen the manuscripts scientific rigor and broader appeal. With these improvements, the study would solidly position itself as a high-impact contribution to Nature Communications.

We thank the reviewer for their constructive evaluation and thoughtful suggestions, which have significantly improved the manuscript. In response, we have:

1. incorporated new analyses connecting our framework directly to dielectrowetting experiments, including a new section and Figure 4;
2. added comparative discussion with established electric-field-based methods such as electrowetting, electrocapillarity, and dielectrophoresis;
3. clarified the generalizability of our framework to multicomponent fluids and highlighted its implications for future studies;
4. refined figure captions and the definition of reduced units to improve accessibility; and
5. repeated our analysis for SPC/E water to test robustness beyond simplified dipolar models, with new results presented in the main text and SI (Figs. S13 and S14).

Report of Reviewer 2

This manuscript presents a numerical investigation of the properties of polar fluids under electric gradients. The starting point is to use dielectrophoresis, a phenomenon under which the polar fluids undergoes local forces in an electric gradient. The central claim is that such forces could be used to compact or dilate the fluids. The manuscript explores the possibility of using these compaction/dilatation forces to tune the liquid-vapor phase diagram of the fluid and then to use that to change wetting properties inside confined systems. In essence, the idea is to explore how dielectrophoresis might modify capillary effects in a similar way as electrophoresis modifies capillary forces.

The idea itself is interesting and creative. Dielectrophoresis being quite subtle, I have no doubt that peculiar phenomena might arise in specific systems, so dielectrocapillary effects are indeed a possibility that has not been studied so far. The manuscript also includes an investigation of screening of the dielectrophoretic interactions which is captivating too.

We thank the reviewer for their thoughtful evaluation and for highlighting the novelty and interest of our work.

At this stage, however, the model makes 2 major assumptions which are not well justified and, therefore, make it hard to validate whether or not the effect might be real. First, an equilibrium AI model is used to study nonequilibrium phenomena. Second, capillarity related to properties near confining surfaces however the response of the surfaces to the electrical gradients are not accounted for and may strongly tune the response as is the case for electrocapillarity.

Regarding the first point, we emphasize that throughout the manuscript we investigate the equilibrium response of fluids. The theoretical basis is laid out in recently formulated liquid state theory based on hyperdensity functional theory (an extension of classical density functional theory) that is capable of describing electromechanics in fluids as an emergent phenomenon:

- A. T. Bui and S. J. Cox, A first-principles approach to electromechanics in liquids, *J. Phys.: Condens. Matter* 37, 285101 (2025).

We have now summarize the key results used in our study in the Methods section.

We identified several places where the use of the word “dynamic” or “dynamical” may have given an incorrect impression that we were investigating nonequilibrium effects. We apologize if this was misleading, and we have removed references to “dynamic” or “dynamical.” We also now include a clear statement in the conclusions that we only investigate equilibrium effects, and acknowledge that we omit potentially interesting nonequilibrium effects. We also give an outlook regarding how our work may be viewed as a platform for investigating such phenomena.

Regarding the second point, we agree that this is an important consideration. In our framework, the confining walls are modeled as fixed external potentials, and thus their back-response to the applied electric gradients is not explicitly captured. In macroscale dielectrowetting experiments, this effect is generally expected to be small, and indeed, experimental observations support this assumption. To address this limitation, we have included a discussion of field gradients generated by patterned electrodes in dielectrowetting phenomena, which provides insight into how surface responses may influence the fluid behavior. This is further detailed in point 6.

(Lines 470–508) Hyperdensity functional theory. We employed the recently developed first-principles theory for electromechanics²⁷ based on hyperdensity functional theory⁶², which provides an exact variational framework for the coupled electromechanical response of fluids. For a single-component fluid, at specified μ , T , planar external non-electrostatic potential $V_{\text{ext}}(z)$, and electrostatic potential $\phi(z)$, the grand potential functional is

$$\Omega([\rho, \beta\phi], T) = \mathcal{F}_{\text{intr}}^{\text{id}}([\rho], T) + \mathcal{F}_{\text{intr}}^{\text{ex}}([\rho, \beta\phi], T) + \int dz \rho(z)[V_{\text{ext}}(z) - \mu],$$

where the ideal intrinsic Helmholtz free energy is $\mathcal{F}_{\text{intr}}^{\text{id}}([\rho], T) = k_{\text{B}}T \int dz \rho(z)[\ln \zeta^{-1} \Lambda^3 \rho(z) - 1]$, with Λ denoting the thermal de Broglie wavelength and ζ denotes a partition function accounting for intramolecular molecular degrees of freedom of each fluid particle.

The equilibrium number and charge density profiles are obtained by solving the Euler-Lagrange (EL) equation

$$\rho_{\text{eq}}(z) = \frac{\zeta}{\Lambda^3} \exp[-\beta V_{\text{ext}}(z) + \beta\mu + c^{(1)}(z; [\rho_{\text{eq}}, \beta\phi], T)],$$

and evaluating

$$n_{\text{eq}}(z) = n^{(1)}(z; [\rho_{\text{eq}}, \beta\phi], T),$$

with the one-body direct correlation functional $c^{(1)}$ and charge density functional $n^{(1)}$ defined as first functional derivatives of the excess intrinsic Helmholtz free energy functional

$$c^{(1)}(z; [\rho, \beta\phi], T) = -\frac{\delta \beta \mathcal{F}_{\text{intr}}^{\text{ex}}([\rho, \beta\phi], T)}{\delta \rho(z)},$$

$$n^{(1)}(z; [\rho, \beta\phi], T) = \frac{\delta \mathcal{F}_{\text{intr}}^{\text{ex}}([\rho, \beta\phi], T)}{\delta \beta\phi(z)}.$$

For fluids whose intermolecular interactions are long-ranged, such as ionic and dielectric fluids, the functional dependence of $c^{(1)}$ and $n^{(1)}$ on ρ and ϕ is, in general, non-local. Note that, as we only discuss these functionals evaluated at equilibrium in this work, we drop the “eq” subscript for notational convenience.

In order to learn these functional mappings, we applied the techniques introduced in Refs.^{24,27} where the local functionals of short-ranged mimic fluids, $c_{\text{R}}^{(1)}(z; [\rho, \beta\phi], T)$ and $n_{\text{R}}^{(1)}(z; [\rho, \beta\phi], T)$, were learned from molecular simulation data using the neural functional method²⁵. A SR mimic fluid is defined as a system whose Coulomb interactions are replaced with

$$\frac{1}{r} \rightarrow \frac{\text{erfc}(\kappa r)}{r}.$$

For SPC/E water and the dipolar fluid, we used $\kappa^{-1} = 4.5 \text{ \AA}$. For the electrolyte, we used the functionals trained in Ref. 24, with $\kappa^{-1} = 5.0 \text{ \AA}$. The effects of long-ranged interactions are accounted for in a well-controlled mean field fashion using local molecular field theory (LMFT). Implementation details and full derivations are provided in Ref.²⁷ and Section S1 of the SI. We also validated the theory against computer simulations in Section S3 of the SI.

(Lines 67–69) ...within the grand canonical ensemble, which is most representative of real conditions in which fluid molecules can **dynamically enter** and leave a pore.

(Lines 77–79) Furthermore, the ability to **dynamically regulate** hysteresis introduces a new level of programmability in nanofluidic systems...

(Lines 393–395) ...dielectrocapillarity, a new phenomenon in which capillary condensation and criticality under confinement can be **dynamically finely** tuned.

(Lines 402–413) The effects uncovered in this work concern the equilibrium behavior of dielectric liquids, and omit potentially important nonequilibrium effects such as pore entry and exit^{53–55}, electrokinetic phenomena^{56–58}, and controlled wetting dynamics such as rate-dependent droplet spreading⁵⁰. Nonetheless, the implications of our results for nonequilibrium behavior are potentially far-reaching. A natural possible progression from this work is to augment our first-principles framework for electromechanics with dynamical extensions of cDFT⁵⁹, opening a promising route toward a microscopic understanding of how EFGs impact non-equilibrium processes.

Another main point is that it is not clear that the orders of magnitude of changes investigated on a minimal polar fluid model might indeed be significant and hence accessible in experiments. This is not made easier by the current style of the manuscript which sometimes makes unjustified claims. At this stage I rather see the model as a remarkable and creative hypothesis, but its potential applicability in realistic systems is not discussed at all, making it hard to see if it could be real.

We thank the reviewer for raising this important point that, which is also raised by Reviewer 1. As we discuss in response to Reviewer 1, we now include results for model of water with explicit hydrogen bonds and electrostatics, thus demonstrating that the results in this work are applicable to a realistic fluid. We reproduce our response to reviewer 1 below for ease of reference:

To assess the robustness of our conclusions beyond simplified dipolar models, we have repeated our analysis for SPC/E water. The results confirm our main findings: dielectrophoretic rise is observed at supercritical temperature, and the critical temperature shifts downward with increasing EFG strength. Importantly, water also reveals additional features due to its intrinsic charge asymmetry, which we now highlight and discuss. These new results are incorporated throughout the revised manuscript and provided in detail in the SI.

(Lines 84–94) To investigate how EFGs influence polar fluids, we **primarily** consider a minimal molecular model that incorporates **soft-core** repulsion, van der Waals attraction, and long-range dipolar interactions. **The advantage of using such a simple molecular model is that we can exhaustively explore a broad range of thermodynamic conditions, while potentially uncovering common behaviors among polar fluids, from molecular liquids to colloidal systems.** In addition, we also investigate a commonly used simple point charge model for water (SPC/E²¹) that explicitly incorporates hydrogen-bonding, under thermodynamic conditions close to its critical point.

(Lines 148–151) ...As a starting point, we characterize the bulk response of **water, the simple polar fluid, and the electrolyte, all under supercritical conditions**, when subjected to a sinusoidal electric field, $E^*(z) = E_{\max}^* \sin(2\pi z/\lambda)$, as shown in Fig. 1a.

Figure 1: Reorganization of fluids under non-uniform electric fields. An applied electric field, $E^*(z) = E_{\max}^* \sin(2\pi z/\lambda)$, shown in (a), induces pronounced density variations in bulk supercritical water, as can clearly be seen from a snapshot of a molecular dynamics simulation. (b) Results from cDFT for the number ($\rho^*(z)$, top) and charge ($n^*(z)$, bottom) densities capture this behavior. It can clearly be seen that number density is locally depleted where the $|\nabla E|$ is large, and locally enhanced where $|\nabla E|$ is small. The same qualitative behavior is seen in (c) for a supercritical dipolar fluid, except that its response is symmetric, in contrast to water where local depletion depends upon the sign of ∇E . (d) In contrast to both water and the dipolar fluid, an electrolyte is locally depleted in regions of low field strength due to electrophoretic forces (purple and green lines show cation and anion density, respectively, while the blue line shows the total density). Reduced units are described in the Methods section.

Figure S13: Dielectrophoretic rise in water under supercritical condition. The left panel shows the density response of the simple dipolar fluid (500 K, same data as in main manuscript) while the the right panel shows that for SPC/E water (700 K).

(Lines 154–174) In the case of water, Fig. 1b, we see that the applied electric field induces a significant structural reorganization along the field direction; its average density profile $\rho^*(z)$ is locally depleted in regions of weaker field strength, while molecular reori-

Figure S14: Water liquid–vapor equilibrium under EFGs. The intensity of the field E_{\max} is labeled. The bottom panel shows the density response as the chemical potential is varied continuously.

entation leads to an inhomogeneous average charge density distribution $n^*(z)$. As water is overall neutral and therefore experiences no net electrophoretic force, the observed local reorganization arises instead from dielectrophoretic forces, $f_{\text{DEP}} \sim \nabla E^{23,34}$, which push the fluid towards regions of higher electric field strength—an effect termed “dielectrophoretic rise.” This dielectrophoretic force is the same that drives dielectrophoresis, which is widely exploited to manipulate biological cells³⁵ and colloids³⁶, but its role in molecular fluids has received little attention. While water provides an important example of a polar fluid, the observed effects are by no means specific to aqueous systems. As seen in Fig. 1c, the overall picture is the same for the simple polar fluid, aside from the fact that it exhibits a symmetric response, whereas for water, depletion is stronger in regions where $\nabla E < 0$ than where $\nabla E > 0$ due to the inherent charge asymmetry of the water molecule.

(Lines 186–191) To illustrate this, we present in Fig. 2a how dielectrophoretic rise can be amplified by controlling the applied field. Owing to their qualitatively similar behavior, in the remainder of the article we focus on the dipolar fluid, with results for water given in the SI, Figs. S13 & S14.

(Lines 263–267) For water, we observe similar behavior, with a downward shift in T_c of approx. 50 K at $E_{\max} = 0.4 \text{ V \AA}^{-1}$ (Fig. S14). These observations highlight the exquisite level of control that one can exert over dielectric fluids with EFGs.

I have several detailed comments, that I put below.

(1) What goes inside the model and what data the AI is trained on is quite opaque, at least from the main text. Is it trained on the different fluids mentioned or on other fluids? How is the functional learned exactly, aka what is the training process? (related to one of my main question) What makes us sure that a model that was designed and trained at equilibrium and

is potentially suitable to investigate driving by uniform fields and then suitable to investigate drivings by non uniform fields? Are some standard results in the field recovered? At this stage it is still not clear to me how the AI training enables one to study the system faster/better so it would be nice to discuss potential limitations/advantages a bit more.

We thank the reviewer for raising these important points regarding the construction and training of our model. In the revised manuscript, we have clarified that for each fluid considered (SPC/E water, the dipolar fluid, and the restricted primitive model of an electrolyte) we construct a neural functional trained on simulation data specific to that system. The phenomena explored in the manuscript are purely equilibrium in nature. We have verified that our approach reproduces standard simulation results, including the bulk equation of state, the zero-field binodal, and the number and charge density profiles. Furthermore, we now include results demonstrating that the theory can recover results from dielectrowetting experiments, as discussed in point 6.

To address this point, we have moved some content from the supporting information to the main manuscript. Specifically, we have added the construction, training, and validation of the AI model in the Methods:

(Lines 509–534) Generation of training data. For the electrolyte, we employed the functional reported in Ref.²⁴ based upon the restricted primitive model. For SPC/E water and the dipolar fluid, we generated training data using a combination of grand canonical Monte Carlo (GCMC) and molecular dynamics (MD) simulations to construct local reference functionals $c_R^{(1)}(z; [\rho, \beta\phi], T)$ and $n_R^{(1)}(z; [\rho, \beta\phi], T)$. For each fluid, we sampled ~ 2000 randomized external conditions spanning both subcritical and supercritical regimes. Random inhomogeneous electrostatic potentials of the form $\phi(z) = \phi_0 \cos(2\pi kz)$ were applied. In some cases, planar walls of the form of a 9-3 Lennard–Jones potential were also included. For each condition, GCMC (performed with our own code⁶³) determined the mean particle number, which was then used to initialize MD simulations in the NVT ensemble with LAMMPS⁶⁴, from which the number and charge density profiles are sampled. For water, we defined the molecular center for sampling $\rho(z)$ at the oxygen site, and for the dipolar fluid at the midpoint between the two opposite charges. The charge density was computed as $n(z) = \sum_i \sum'_\alpha q_\alpha \langle \delta(z - z_\alpha) \rangle$, where the outer sum is over all molecules, and the inner primed sum is over all sites belonging to molecule i , with z_α and q_α indicating the z coordinate and charge, respectively, of site α . This hybrid scheme reproduces the accuracy of pure GCMC sampling while significantly accelerating convergence of inhomogeneous structures. The total computation time for the generation of the entire dataset is in the order of $\sim 10^5$ CPU hours.

(Lines 535–555) Training neural functionals. For both SPC/E water and the dipolar fluid, we train two neural networks to represent $c_R^{(1)}([\rho, \beta\phi], T)$ and $n_R^{(1)}([\rho, \beta\phi], T)$ following the local learning strategy²⁵. The machine learning routine was implemented in Keras/Tensorflow⁶⁵. Inputs consisted of local density and electrostatic potential in a sliding spatial window of size 10 \AA from the center of the position of interest. To effectively learn spatial variations, the model internally computes the gradient of $\beta\phi(z)$ using a central difference scheme. In addition to these spatially varying inputs, a separate input node encodes T as a scalar. The full architecture details are provided in Figs. S5 and S6. The dataset was split roughly in a 3:1:1 ratio for training, validation, and test sets. Models were trained for 200 epochs with a batch size of 256, using an exponentially decaying learning rate starting at 0.001, achieving errors comparable to the estimated simulation

noise. The training of the neural networks was done on a GPU (NVIDIA GeForce RTX 3060) in a few hours. We also verified that the trained functional can recover standard simulation results including bulk equation of state (Fig. S7) the binodal at zero applied field (Fig. S8) as well as number density and charge density response (Fig. S9).

(2) The electrophoretic rise part could be described and explained more carefully. At this stage, I still don't understand it. I believe this is mainly caused by an absence of details. First, in which direction is the field E imposed? Along the z direction? To me it appears that regions of higher field actually have a locally neutral ion concentration and so there should be no force on the fluid in these regions, yet this is apparently not what is observed. I can not make sense of the schematic drawing because it is not clear where the periodic unit is taken along the axis and what the line represents. Eventually I find it would be worthwhile to discuss a little bit how, already at this stage, working in the μ VT ensemble instead of NVT leads to interesting differences compared to more traditional simulations setups where the total density varies spatially a lot. This is still surprising to me since I expect fluid to be more or less incompressible and so homogeneous over such lengthscales. Actually, the relationship between the densities ρ^* and n^* is not clear from the images or the caption. Maybe one should consider putting in that information maybe twice. Usually n is used for numbers and so would be used instead of ρ , so that convention was puzzling.

(3) It is equally hard to understand the dielectrophoretic rise for similar reasons. What is n^* in the case of the polar fluid? The fluid should be locally neutral.

We address points 2 and 3 together regarding the electrophoretic and dielectrophoretic rises.

The applied electric field is oriented along the z direction, and the structural reorganization of the fluid occurs along this axis.

To clarify the mechanisms, we have added a molecular depiction through a simulation snapshot of bulk water undergoing dielectrophoretic rise (Fig. 1a).

For efficient sampling of the condensed phase, training data is acquired using a combination of grand canonical Monte Carlo (GCMC) and molecular dynamics (MD) simulations, with randomly generated inhomogeneous potential energy landscapes in a planar geometry. We first use GCMC to determine the average total number of particles N at a specified $\{\mu; V; T; V_{\text{ext}}(z); \phi(z)\}$, and then perform canonical MD simulations with $\{N; V; T; f_{\text{ext}}(z); E(z)\}$. We have validated that this approach gives results indistinguishable from when we only performed GCMC simulations, yet significantly helps to converge density profiles

(Lines 503–528) Generation of training data. ... For SPC/E water and the dipolar fluid, we generated training data using a combination of grand canonical Monte Carlo (GCMC) and molecular dynamics (MD) simulations to construct local reference functionals $c_{\text{R}}^{(1)}(z; [\rho, \beta\phi], T)$ and $n_{\text{R}}^{(1)}(z; [\rho, \beta\phi], T)$. For each fluid, we sampled ~ 2000 randomized external conditions spanning both subcritical and supercritical regimes. Random inhomogeneous electrostatic potentials of the form $\phi(z) = \phi_0 \cos(2\pi kz)$ were applied. In some cases, planar walls of the form of a 9-3 Lennard–Jones potential were also included. For each condition, GCMC (performed with our own code⁶³) determined the mean particle number, which was then used to initialize MD simulations in the NVT ensemble with LAMMPS⁶⁴, from which the number and charge density profiles are sampled. This hybrid

scheme reproduces the accuracy of pure GCMC sampling while significantly accelerating convergence of inhomogeneous structures...

Regarding the relationship between number and charge densities: The number density $\rho(z)$ is defined at a molecular center: for water, the oxygen site; for the dipolar fluid, the midpoint between the two point charges; for the electrolyte, at the center of each ion. The charge density is defined as $n(z) = \sum_i \sum'_\alpha q_\alpha \langle \delta(z - z_\alpha) \rangle$, where the outer sum is over all molecules, and the inner primed sum is over all sites belonging to molecule i , with z_α and q_α indicating the z coordinate and charge, respectively, of site α . For water and the dipolar fluid, there is no simple analytical relation between $n(z)$ and the molecular center number density; instead, $n(z)$ is evaluated through the learned functional $n^{(1)}(z; [\rho, \beta\phi], T)$. For electrolytes, the relationship is straightforward: $n(z) = q_+ \rho_+(z) + q_- \rho_-(z)$. We note that the use of ρ for number density and n for charge density follows conventions in classical DFT and microscopic simulations, which seems to be the opposite of the typical notation in continuum methods.

This information is now in the Methods

(Lines 475–482) The equilibrium number and charge density profiles are obtained by solving the Euler–Lagrange (EL) equation

$$\rho_{\text{eq}}(z) = \frac{\zeta}{\Lambda^3} \exp \left[-\beta V_{\text{ext}}(z) + \beta \mu + c^{(1)}(z; [\rho_{\text{eq}}, \beta\phi], T) \right],$$

and evaluating

$$n_{\text{eq}}(z) = n^{(1)}(z; [\rho_{\text{eq}}, \beta\phi], T),$$

with the one-body direct correlation functional $c^{(1)}$ and charge density functional $n^{(1)}$ defined as first functional derivatives of the excess intrinsic Helmholtz free energy functional

$$c^{(1)}(z; [\rho, \beta\phi], T) = -\frac{\delta \beta \mathcal{F}_{\text{intr}}^{\text{ex}}([\rho, \beta\phi], T)}{\delta \rho(z)},$$

$$n^{(1)}(z; [\rho, \beta\phi], T) = \frac{\delta \mathcal{F}_{\text{intr}}^{\text{ex}}([\rho, \beta\phi], T)}{\delta \beta \phi(z)}.$$

For fluids whose intermolecular interactions are long-ranged, such as ionic and dielectric fluids, the functional dependence of $c^{(1)}$ and $n^{(1)}$ on ρ and ϕ is, in general, non-local. Note that, as we only discuss these functionals evaluated at equilibrium in this work, we drop the “eq” subscript for notational convenience.

We also rewrote the description of dielectrophoretic and electrophoretic rise in the main text and the caption of Figure 1.

(Caption to Figure 1) Reorganization of fluids under non-uniform electric fields.

An applied electric field, $E^*(z) = E_{\text{max}}^* \sin(2\pi z/\lambda)$, shown in **(a)**, induces pronounced density variations in bulk supercritical water, as can clearly be seen from a snapshot of a molecular dynamics simulation. **(b)** Results from cDFT for the number ($\rho^*(z)$, top) and charge ($n^*(z)$, bottom) densities capture this behavior. It can clearly be seen that number density is locally depleted where the $|\nabla E|$ is large, and locally enhanced where $|\nabla E|$ is small. The same qualitative behavior is seen in **(c)** for a supercritical dipolar fluid, except that its response is symmetric, in contrast to water where local depletion depends upon the sign of ∇E . **(d)** In contrast to both water and the dipolar fluid, an

electrolyte is locally depleted in regions of low field strength due to electrophoretic forces (purple and green lines show cation and anion density, respectively, while the blue line shows the total density). Reduced units are described in the Methods section.

(Lines 154–174) In the case of water, Fig. 1b, we see that the applied electric field induces a significant structural reorganization along the field direction; its average density profile $\rho^*(z)$ is locally depleted in regions of weaker field strength, while molecular reorientation leads to an inhomogeneous average charge density distribution $n^*(z)$. As water is overall neutral and therefore experiences no net electrophoretic force, the observed local reorganization arises instead from dielectrophoretic forces, $f_{\text{DEP}} \sim \nabla E^2$ ^{33,34}, which push the fluid towards regions of higher electric field strength—an effect termed “dielectrophoretic rise.” This dielectrophoretic force is the same that drives dielectrophoresis, which is widely exploited to manipulate biological cells³⁵ and colloids³⁶, but its role in molecular fluids has received little attention. While water provides an important example of a polar fluid, the observed effects are by no means specific to aqueous systems. As seen in Fig. 1c, the overall picture is the same for the simple polar fluid, aside from the fact that it exhibits a symmetric response, whereas for water, depletion is stronger in regions where $\nabla E < 0$ than where $\nabla E > 0$ due to the inherent charge asymmetry of the water molecule.

(Lines 175–183) In contrast, applying a sinusoidal field to an electrolyte induces “electrophoretic rise” in which the fluid migrates toward regions of lower electric field strength (Fig. 1d). This is a result of the electrophoretic force $f_{\text{EP}} = qE$, where E is the local field strength and q is the ionic charge, causing the anions and cations to reorganize, with peaks in their density profiles out of phase due to their opposing charges. In this way, polar fluids and ionic fluids display electromechanical responses that are fundamentally distinct from each other.

(4) I do not understand how the phase diagram is constructed this is information that is completely in the SI but some short explanation would be especially useful. Would it be possible to give some actual illustration of what the separated phases would look like? Do they really have the wiggly interfaces that are shown on the sketches? Because then this creates a surface tension in the fluid that is much bigger than that for a flat interface; so theres a price to pay for this compacted state which makes the story a lot more complicated.

To determine the liquid–vapor coexistence line at zero field, we compute isotherms of the chemical potential as a function of bulk density and apply Maxwell construction to identify the coexisting liquid and vapor densities at the binodal. That is, we explicitly feed in a homogeneous bulk density into the neural network. Under non-uniform electric fields, due to electromechanical coupling, we do not know *a priori* the exact form of the average microscopic one-body density. In this case, we instead fix μ, T and obtain inhomogeneous solutions by solving the Euler–Lagrange equation. The resulting chemical potential isotherms, expressed in terms of the mean density, exhibit discontinuities that mark phase separation. Stable branches are distinguished from metastable ones by comparing grand potentials. The binodal is thus constructed by locating liquid and vapor solutions with equal grand potentials, rather than by solving for solution with a liquid–vapor interface explicitly.

The original sketches in Fig. 2d were schematic illustrations of density modulation along z in bulk, not literal depictions of curved interfaces – we appreciate that this may have been misleading. In addition to the changes made in Fig. 1, which hopefully clarify that we are applying the sinusoidal field to a bulk system, in Fig. 2c we now explicitly plot both liquid and vapor solutions across the chemical potential range, which provides a direct illustration of the separated phases.

Figure 2: Controlling liquid–vapor equilibrium with EFGs. (a) At $T^* > T_{c,0}^*$ where $T_{c,0}^* \approx 1.97$, increasing EFGs by independently varying λ and E_{\max} amplifies dielectrophoretic rise. (b) The local electrostrictive response of the dipolar fluid, as measured by the rise in the maximum density peak relative to zero field $\Delta\rho_{\max}^*$, is highly non-linear. Solid and dashed lines show the response of systems with dipolar interactions that are long-ranged (LR), e.g., polar molecules, and screened short-ranged (SR), e.g., colloids, respectively. The effect of LR interactions becomes pronounced for $\lambda \gg \sigma$. The SR fluid is a nearly identical dipolar fluid, but whose Coulomb potential is replaced by $\text{erfc}(\kappa r)/r$ where $\kappa^{-1} = 1.5\sigma$. (c) At an isotherm where $T^* < T_{c,0}^*$, stable solutions for the density $\rho^*(z)$ under a sinusoidal electric field with $\lambda/\sigma = 1.7$, are shown for different values of the chemical potential. These results are used to investigate liquid–vapor coexistence. (d) Results in light pink show the binodal of the dipolar fluid in the absence of an electric field. At $E_{\max}^* = 4$, T_c^* shifts to a lower temperature, as seen in the binodal in dark purple. Solid symbols show results obtained from the multiscale cDFT approach, while crosses indicate estimates of T_c^* using the law of rectilinear diameters and critical exponents²⁸. Solid lines serve as a guide to the eye.

In the main text we added:

(Lines 237–254) In Fig. 2c, we show the density profiles of the fluid along the direction of an external sinusoidal electric field with $\lambda/\sigma = 1.7$, at $T^* < T_{c,0}^*$, where $T_{c,0}^*$ is the critical temperature in the absence of an external field. Results are shown for different chemical potentials. At zero field, the stable solutions separate into homogeneous vapor and liquid states at low and high chemical potential, respectively. As the field strength increases, stronger EFGs (in absolute terms) not only give rise to dielectrophoretic rise but also destabilize liquid–vapor phase separation. Consequently, within the coexistence region, the vapor phase becomes denser, or contracts, while the liquid phase expands.

For sufficiently large EFGs ($E_{\max}^* = 8$), the fluid undergoes a transition to a single-phase supercritical fluid. By locating vapor and liquid solutions with equal grand potentials, in Fig. 2d we map out the liquid–vapor binodal curve for the dipolar fluid, both without an external field and under this sinusoidal field with $E_{\max}^* = 4$.

and more extensively in the Methods

(Lines 556–595) Using the neural functionals. Evaluating the trained neural functionals is fast (\sim milliseconds) and can be performed on a CPU or GPU. Combining with LMFT give us $c^{(1)}([\rho, \beta\phi], T)$ and $n^{(1)}([\rho, \beta\phi], T)$. The EL equation is solved self-consistently with a mixed Picard iteration scheme, which typically converges within minutes. To determine the liquid–vapor coexistence line at zero electric field, we calculate isotherms of the chemical potential as a function of the bulk density ρ_b from

$$\beta\mu = \ln(\Lambda^3 \rho_b) - c^{(1)}([\rho_b, \beta\phi = 0], T),$$

and perform a Maxwell construction to find the coexisting liquid and vapor densities at the binodal. For inhomogeneous systems, i.e., as a result of an applied non-uniform electric field, we set the chemical potential and temperature and find the inhomogeneous solutions by solving the EL equation. The isotherms of the chemical potential, now as a function of the mean density, $\bar{\rho} = L^{-1} \int_0^L dz \rho(z)$ where L is the total length of the domain over which $\rho(z)$ is defined, now have distinct jumps when the system undergoes phase separation, giving liquid ρ_l and vapor ρ_v solutions. To distinguish stable solutions from metastable solutions, we also calculate the grand potential $\Omega_\phi = \Omega([\rho, \beta\phi], T)$ at a fixed $\beta\phi$ where the excess free energy term can be evaluated via functional line integration²⁵. Performing this procedure for $\beta\phi = 0$ gives results consistent with those obtained by Maxwell construction. Note that the pressure of the bulk fluid at zero field is given by $-PV = \beta\Omega_0$. To investigate capillary condensation in slit pores, adsorption isotherms are obtained from the mean density $\bar{\rho} = H^{-1} \int_0^H dz \rho(z)$ where H is the slit height. Hysteresis loops are mapped by seeding different initial guesses when solving the EL equation. In the dielectrowetting calculations, the electrostatic free energy per contact area is given as

$$\gamma_{\text{elec}} = \frac{1}{2} \int_{-\infty}^{+\infty} dz \phi(z) n(z),$$

where the factor of a half comes from there being two symmetric confining walls. To construct the dependence of θ on ϕ_0 , we obtain $\alpha = -\partial\gamma_{\text{elec}}/\partial(\phi_0^2)$ and calculate γ_{lv} from a direct coexistence molecular dynamics simulation. We estimate $\theta_0 = 160^\circ$ based on the maximum value of the local compressibility at $\phi_0 = 0$ and comparing to Ref.⁶⁶; while more detailed calculations can provide a more accurate estimate, this value is consistent with the solvophobic nature of the slit, and is sufficient for the purposes of demonstrating the increased wetting with ϕ_0 shown in Fig. 4.

(5) In the confinement case there is a problem of dynamic entrance effects. There may be a question of how fast you are indeed able to recruit more particles within the small opening to contract the fluid etc.

As stated in our earlier responses, our framework is limited to equilibrium properties and does not consider dynamical entrance effects. While such effects are potentially important, they lie beyond the scope of the present study. We now acknowledge this limitation explicitly in the conclusions, and provide an outlook regarding nonequilibrium effects:

While the effects uncovered here are equilibrium phenomena, their non-equilibrium implications are far-reaching. Our first-principles framework for electromechanics can be augmented with dynamical extensions cDFT⁵⁹, opening a promising route toward a microscopic understanding of non-equilibrium processes under EFGs. Such advances would enable the study of dynamical phenomena including pore entry and exit^{53–55}, electrokinetic transport like diffusiophoresis and electroosmosis^{56–58} and controlled wetting dynamics such as rate-dependent droplet spreading⁵⁰....

(6) Often electrical gradients are constructed by using specific/man made surface properties, e.g. patterning electrodes (as e.g. in <https://pubs.acs.org/doi/full/10.1021/acs.jpcllett.4c00278>) or others. So there is a deep question there of how the electric gradient is made and how that couples to the fluid properties which was somewhat discarded.

We thank the reviewer for raising this important point. Indeed, sinusoidal fields are not realistic representations of fields generated in experiments. However, as a theoretical tool, they allow us to build our understanding by probing specific normal modes of the fluid response. In practice, as the reviewer points out, patterned electrodes are often used generate very specific forms of field gradients. As already discussed in response to Reviewer 1, the dielectrowetting experiments

- G. McHale, C. V. Brown, M. I. Newton, G. G. Wells, and N. Sampara, Dielectrowetting driven spreading of droplets, *Phys. Rev. Lett.* 107, 186101 (2011)
- G. McHale, C. V. Brown, and N. Sampara, Voltage-induced spreading and superspreading of liquids, *Nat. Commun* 4, 1605 (2013)

are particularly relevant to our work. Here, interdigitated electrodes are used to show how dielectrophoretic forces change the contact angle of dielectric droplets on surfaces. The resulting field decays exponentially away from the surface, $\phi(z) = \phi_0 \exp(-z/d)$. To connect our theoretical framework to this realistic field form, we performed additional analyses with exponentially decaying fields analogous to those generated by interdigitated electrodes. This enables us to describe dielectrowetting at the nanoscale within the same modeling approach. Our results show enhanced wetting and suppressed density fluctuations in the fluid, consistent with a downward shift of T_c under the application of EFGs, which is also the basis of dielectrocapillarity.

Following the reviewer' comment, we have added a new section, **Connection to dielectrowetting experiments**, and a new **Figure 4**:

(Lines 339–387)

Connecting to dielectrowetting experiments

Thus far we have introduced dielectrocapillarity as an additional mechanism of controlling capillary phenomena, complementing electrocapillarity⁵¹ where electrolytes respond to applied potentials. On the macroscopic scale, these effects manifest in electrowetting⁵² and dielectrowetting^{14,50}, where the contact angle of a droplet can be tuned with applied potentials. In the case of dielectrowetting, a typical experimental setup employs interdigitated electrodes, as shown schematically in Fig. 4a, which generate an exponentially decaying electrostatic potential $\phi(z) = \phi_0 \exp(-z/d)$ that drives a droplet of dielectric fluid to spread. The resulting contact angle can be described by a modified Young's law^{14,50},

$$\cos \theta(\phi_0) = \cos \theta_0 + \frac{\alpha}{\gamma_{lv}} \phi_0^2, \quad (2)$$

Figure 4: Connection to dielectrowetting. (a) Schematic of interdigitated electrodes (left) and the resulting exponentially decaying potential profile (right), which is applied symmetrically from both confining walls in our slit geometry. (b) Density profiles (left) and local compressibility (right), both normalized by their bulk values, show enhanced wall contact density and suppressed density fluctuations as ϕ_0 increases. (c) Electrostatic free energy per unit contact from cDFT (inset) exhibits a quadratic dependence on ϕ_0 , enabling reconstruction of the contact angle via Eq. 1 (assuming $\theta_0 = 160^\circ$ and using $\gamma_{lv} = 0.025 \text{ J m}^{-2}$ computed from direct coexistence simulation), in direct analogy with dielectrowetting experiments^{14,50}.

where θ_0 is the Young’s contact angle at zero field, γ_{lv} the liquid–vapor surface tension, and α a material parameter related to the dielectric response of the liquid. This relationship assumes that the electrostatic energy stored in the liquid droplet is well-described by dielectric continuum theory, and that d is sufficiently small so that any changes in energy due to the electric field are effectively localized to the solid-liquid interface. Under these assumptions, the electrostatic free energy per unit area obeys the simple quadratic scaling $\gamma_{elec} = -\alpha\phi_0^2$.

Our multiscale framework provides a microscopic perspective on this phenomenology, and allows us to test whether, at the nanoscale, EFGs indeed enhance the wetting of dielectric liquids and the extent to which the scaling prescribed by Eq. 1 holds. To this end, we applied an exponentially decaying electrostatic potential with $d = 1.1\sigma$ to the confined dipolar model, symmetrically from both walls of a solvophobic slit. While not trained on such electrostatic potentials, as can be seen in Fig. 4b, the neural functional extrapolates well, yielding physically plausible results, aside from some oscillations at high wetting that are likely minor artifacts. As ϕ_0 increases, so too does the contact density at the wall, verifying that enhanced wetting occurs. We quantify this effect by computing $\gamma_{elec} = \frac{1}{2} \int_{-\infty}^{\infty} dz \phi(z)n(z)$, which, as can be seen in Fig. 4c, decreases in

an approximately quadratic fashion as ϕ_0 increases. By identifying $\alpha = -\partial\gamma_{\text{elec}}/\partial(\phi_0^2)$, we reconstruct the potential-dependent contact angle, as shown in Fig. 4c. While the results of our microscopic theory are broadly in line the macroscopic model (Eq. 1) some subtle differences are observed, especially at small ϕ_0 . These appear to be correlated with significant changes in the local compressibility near the interface, $\chi_\mu(z) = \partial\rho(z)/\partial\mu$, as ϕ_0 increases, which indicates a suppression of density fluctuations. Such microscopic details are lacking in the dielectric continuum model that underpins Eq. 1.

Potential improvement of clarity/wording/typos:

(12) : of a fluid or of a polar fluid?

We clarified the wording to specify polar fluid as intended

In contrast, electric field gradients (EFGs) induce a dielectrophoretic force, offering exquisite electrokinetic control of **polar fluids** even in the absence of free charges.

(general) Electrostrictive is not such a common word so I would suggest defining it.

We have defined electrostriction at first use in the introduction

(Lines) Bringing together the latest advances in liquid state theory, computer simulation, and machine learning, we develop a multiscale framework to study electrostriction—the density response of a dielectric to applied electric fields—in fluids within the grand canonical ensemble,

(introduction) I do not see how phenomena of gas uptake and separation can be so easily put under the same hat as liquid uptake since the electrokinetic properties in such dilute phases may be vastly different.

As we have clarified in our previous responses, our focus is on equilibrium capillary phenomena, which provide a unifying framework for both gas and liquid uptake. We therefore did not make specific changes to the introduction, but we now explicitly acknowledge electrokinetic effects in the Conclusions in our outlook on nonequilibrium phenomena.

(Lines) While the effects uncovered here are equilibrium phenomena, their non-equilibrium implications are far-reaching. Our first-principles framework for electromechanics can be augmented with dynamical extensions cDFT⁵⁹, opening a promising route toward a microscopic understanding of non-equilibrium processes under EFGs. Such advances would enable the study of dynamical phenomena including pore entry and exit^{53–55}, electrokinetic transport like diffusiophoresis and electroosmosis^{56–58} and controlled wetting dynamics such as rate-dependent droplet spreading⁵⁰....

(68) “grand canonical ensemble, which is most representative of real conditions in which fluid molecules can dynamically enter and leave a pore” It is and it is not. As far as I can tell, there is no great way of simplifying the entrance/exits from a pore, because the grand canonical ensemble will remove correlations between particles exiting and coming back, which are correlations that exist. Also, all these things are non-equilibrium so really we dont know. So the grand canonical ensemble is a way of representing things, that may have some advantages compared to others, but it is likely not the best simplification of the system.

Yes, this is a fair point, though it concerns nonequilibrium effects beyond the scope of this work. Use of the word “dynamically” is also potentially misleading in this context. What we hoped to emphasize is that with the grand canonical framework, we can directly quantify the amount of fluid in the system for a given chemical potential (i.e., a pore in equilibrium with a reservoir). We have therefore removed “dynamically,” in addition to discussing nonequilibrium effects in the Conclusions:

(Lines 402–413) The effects uncovered in this work concern the equilibrium behavior of dielectric liquids, and omit potentially important nonequilibrium effects such as pore entry and exit^{53–55}, electrokinetic phenomena^{56–58}, and controlled wetting dynamics such as rate-dependent droplet spreading⁵⁰. Nonetheless, the implications of our results for nonequilibrium behavior are potentially far-reaching. A natural possible progression from this work is to augment our first-principles framework for electromechanics with dynamical extensions of cDFT⁵⁹, opening a promising route toward a microscopic understanding of how EFGs impact non-equilibrium processes.

The introduction now reads:

(Lines 65–69) ...we develop a multiscale framework to study electrostriction in polar fluids—that is, their density response to applied electric fields—within the grand canonical ensemble, which is most representative of real conditions in which fluid molecules can enter and leave a pore.

(94) “such minimal models uncover universal behaviors that emerge across different scales, from molecular liquids to colloidal systems.” → this is a bold claim that is not justified.

We have now support this claim with additional analyses using a water model (SPC/E), as discussed in a previous point. Specifically, the revised text now reads:

(Lines 84–94) To investigate how EFGs influence polar fluids, we primarily consider a minimal molecular model that incorporates soft-core repulsion, van der Waals attraction, and long-range dipolar interactions. The advantage of using such a simple molecular model is that we can exhaustively explore a broad range of thermodynamic conditions, while potentially uncovering common behaviors among polar fluids, from molecular liquids to colloidal systems. In addition, we also investigate a commonly used simple point charge model for water (SPC/E²¹) that explicitly incorporates hydrogen-bonding, under thermodynamic conditions close to its critical point.

(112) twice the word directly and supervise-learn is an unusual concatenation

We corrected this typo and streamlined the wording. The revised text now reads:

(Lines 113–117) Instead of relying on traditional approximations, we leverage state-of-the-art data-driven methodologies to supervise-learn functional mappings directly from quasi-exact reference data from grand canonical Monte Carlo simulations

(169) “dielectrophoretic coupling depends on the EFG rather than the absolute field strengt”. They depend both on the pattern on the field strength.

We have revised the sentence accordingly:

(Lines 184–186) Crucially, dielectrophoretic coupling depends on the EFGs as well as the absolute field strength, and therefore offers greater control over the fluid's response.

In Fig. 1d, the color code is not given.

This panel is now Figure 2a. We have added a color bar to clarify the color code indicated E_{\max}^* .

Literature: In terms of fluid uptake and filling of pores, there is a body of literature related to tailoring wettability properties that could be discussed (see below). Diffusiophoresis has already been used as an externally tunable force to manipulate filling or unfilling of nanopores with imposed chemical gradients (so indeed there already exists some external manipulation although not as easy as electric fields) to manipulate particles.

On the filling of nanopores:

- Giacomello, A., Chinappi, M., Meloni, S., & Casciola, C. M. (2012). Metastable Wetting on Superhydrophobic Surfaces: Continuum and Atomistic Views of the Cassie-BaxterWenzel Transition. *Physical review letters*, 109(22), 226102.

- Picard, C., Gard, V., Michel, L., Cattoën, X., & Charlaix, E. (2021). Dynamics of heterogeneous wetting in periodic hybrid nanopores. *The Journal of Chemical Physics*, 154(16).

Manipulating concentration gradients to fill/unfill dead-end pores:

- Migacz, R. E., Castleberry, M., & Ault, J. T. (2024). Enhanced diffusiophoresis in dead-end pores with time-dependent boundary solute concentration. *Physical Review Fluids*, 9(4), 044203.

- Marbach, S., & Bocquet, L. (2019). Osmosis, from molecular insights to large-scale applications. *Chemical Society Reviews*, 48(11), 3102-3144.

- Shin, S., Um, E., Sabass, B., Ault, J. T., Rahimi, M., Warren, P. B., & Stone, H. A. (2016). Size-dependent control of colloid transport via solute gradients in dead-end channels. *Proceedings of the National Academy of Sciences*, 113(2), 257-261.

- Shin, S., Ault, J. T., Warren, P. B., & Stone, H. A. (2017). Accumulation of colloidal particles in flow junctions induced by fluid flow and diffusiophoresis. *Physical Review X*, 7(4), 041038.

We thank the reviewer for pointing out this important body of work. We have expanded the Conclusion section to place our results in the broader context of externally controlled wetting and transport phenomena. The above papers have also been cited. The revised text reads:

(Lines 402–413) The effects uncovered in this work concern the equilibrium behavior of dielectric liquids, and omit potentially important nonequilibrium effects such as pore entry and exit^{53–55}, electrokinetic phenomena^{56–58}, and controlled wetting dynamics such as rate-dependent droplet spreading⁵⁰. Nonetheless, the implications of our results for nonequilibrium behavior are potentially far-reaching. A natural possible progression from this work is to augment our first-principles framework for electromechanics with dynamical extensions of cDFT⁵⁹, opening a promising route toward a microscopic understanding of how EFGs impact non-equilibrium processes.

We thank the reviewer for their constructive evaluation and thoughtful suggestions, which have significantly improved the manuscript. In response, we have:

1. added further details on our methods to the main text, including theory, training procedures, and clarification that our focus is on equilibrium capillary processes;

2. repeated our analysis for SPC/E water to test the robustness of our conclusions beyond simplified dipolar models, with new results included in the main text and SI (Figs. S13 and S14);
3. introduced molecular schematics and clarified definitions (e.g., number vs. charge density) to better illustrate field direction and distinguish dielectrophoretic from electrophoretic rise;
4. incorporated new results on dielectrowetting under exponentially decaying fields generated by interdigitated electrodes, directly linking our framework to experimental setups.
5. discussed possible extensions of our framework to non-equilibrium phenomena

REFERENCES

- ¹Z. Chen, P. Li, R. Anderson, X. Wang, X. Zhang, L. Robison, L. R. Redfern, S. Moribe, T. Islamoglu, D. A. Gómez-Gualdrón, T. Yildirim, J. F. Stoddart, and O. K. Farha, Balancing volumetric and gravimetric uptake in highly porous materials for clean energy, *Science* **368**, 297 (2020).
- ²J. Chmiola, G. Yushin, Y. Gogotsi, C. Portet, P. Simon, and P. L. Taberna, Anomalous increase in carbon capacitance at pore sizes less than 1 nanometer, *Science* **313**, 1760 (2006).
- ³X. Liu, D. Lyu, C. Merlet, M. J. A. Leesmith, X. Hua, Z. Xu, C. P. Grey, and A. C. Forse, Structural disorder determines capacitance in nanoporous carbons, *Science* **384**, 321 (2024).
- ⁴Q. Yang, P. Z. Sun, L. Fumagalli, Y. V. Stebunov, S. J. Haigh, Z. W. Zhou, I. V. Grigorieva, F. C. Wang, and A. K. Geim, Capillary condensation under atomic-scale confinement, *Nature* **588**, 250 (2020).
- ⁵L. Fumagalli, A. Esfandiari, R. Fabregas, S. Hu, P. Ares, A. Janardanan, Q. Yang, B. Radha, T. Taniguchi, K. Watanabe, G. Gomila, K. S. Novoselov, and A. K. Geim, Anomalous low dielectric constant of confined water, *Science* **360**, 1339 (2018).
- ⁶L. Bocquet, Nanofluidics coming of age, *Nat. Mater.* **19**, 254 (2020).
- ⁷A. Noy and S. B. Darling, Nanofluidic computing makes a splash, *Science* **379**, 143 (2023).
- ⁸M. Salanne, B. Rotenberg, K. Naoi, K. Kaneko, P.-L. Taberna, C. P. Grey, B. Dunn, and P. Simon, Efficient storage mechanisms for building better supercapacitors, *Nat. Energy* **1**, 16070 (2016).
- ⁹D. S. Sholl and R. P. Lively, Seven chemical separations to change the world, *Nature* **532**, 435 (2016).
- ¹⁰D. L. Gin and R. D. Noble, Designing the next generation of chemical separation membranes, *Science* **332**, 674 (2011).
- ¹¹C. Gu, N. Hosono, J.-J. Zheng, Y. Sato, S. Kusaka, S. Sakaki, and S. Kitagawa, Design and control of gas diffusion process in a nanoporous soft crystal, *Science* **363**, 387 (2019).
- ¹²Y. Tsoi, F. Tournilhac, and L. Leibler, Demixing in simple fluids induced by electric field gradients, *Nature* **430**, 544 (2004).
- ¹³R. A. Hayes and B. J. Feenstra, Video-speed electronic paper based on electrowetting, *Nature* **425**, 383 (2003).
- ¹⁴G. McHale, C. V. Brown, M. I. Newton, G. G. Wells, and N. Sampara, Dielectrowetting driven spreading of droplets, *Phys. Rev. Lett.* **107**, 186101 (2011).
- ¹⁵K. Y. C. Lee, J. F. Klingler, and H. M. McConnell, Electric field-induced concentration gradients in lipid monolayers, *Science* **263**, 655 (1994).
- ¹⁶R. Dupuis, P.-L. Valdenaire, R. J.-M. Pellenq, and K. Ioannidou, How chemical defects influence the charging of nanoporous carbon supercapacitors, *Proc. Natl. Acad. Sci. U.S.A* **119**, e2121945119 (2022).
- ¹⁷C. Merlet, B. Rotenberg, P. A. Madden, P.-L. Taberna, P. Simon, Y. Gogotsi, and M. Salanne, On the molecular origin of supercapitance in nanoporous carbon electrodes, *Nat. Mater.* **11**, 306 (2012).
- ¹⁸V. Kapil, C. Schran, A. Zen, J. Chen, C. J. Pickard, and A. Michaelides, The first-principles phase diagram of monolayer nanoconfined water, *Nature* **609**, 512 (2022).

- ¹⁹T. Xiong, C. Li, X. He, B. Xie, J. Zong, Y. Jiang, W. Ma, F. Wu, J. Fei, P. Yu, and L. Mao, Neuromorphic functions with a polyelectrolyte-confined fluidic memristor, *Science* **379**, 156 (2023).
- ²⁰P. Robin, N. Kavokine, and L. Bocquet, Modeling of emergent memory and voltage spiking in ionic transport through angstrom-scale slits, *Science* **373**, 687 (2021).
- ²¹H. J. C. Berendsen, J. R. Grigera, and T. P. Straatsma, The missing term in effective pair potentials, *J. Phys. Chem.* **91**, 6269 (1987).
- ²²R. Evans, The nature of the liquid-vapour interface and other topics in the statistical mechanics of non-uniform, classical fluids, *Adv. Phys.* **28**, 143 (1979).
- ²³W. Kohn, Nobel lecture: Electronic structure of matter—wave functions and density functionals, *Rev. Mod. Phys.* **71**, 1253 (1999).
- ²⁴A. T. Bui and S. J. Cox, Learning classical density functionals for ionic fluids, *Phys. Rev. Lett.* **134**, 148001 (2025).
- ²⁵F. Sammüller, S. Hermann, D. de las Heras, and M. Schmidt, Neural functional theory for inhomogeneous fluids: Fundamentals and applications, *Proc. Natl. Acad. Sci. U.S.A.* **120**, e2312484120 (2023).
- ²⁶F. Sammüller, M. Schmidt, and R. Evans, Neural density functional theory of liquid-gas phase coexistence, *Phys. Rev. X* **15**, 011013 (2025).
- ²⁷A. T. Bui and S. J. Cox, A first-principles approach to electromechanics in liquids, *J. Phys.: Condens. Matter* **37**, 285101 (2025).
- ²⁸J. Rowlinson and B. Widom, *Molecular Theory of Capillarity*, Dover books on chemistry (Dover Publications, 2002).
- ²⁹P. Debye and K. Kleboth, Electrical field effect on the critical opalescence, *J. Chem. Phys.* **42**, 3155 (1965).
- ³⁰M. J. Stevens and G. S. Grest, Phase coexistence of a Stockmayer fluid in an applied field, *Phys. Rev. E* **51**, 5976 (1995).
- ³¹C. Zhang and M. Sprik, Electromechanics of the liquid water vapour interface, *Phys. Chem. Chem. Phys.* **22**, 10676 (2020).
- ³²G. Cassone and F. Martelli, Electrofreezing of liquid water at ambient conditions, *Nat. Commun* **15**, 1856 (2024).
- ³³H. A. Pohl, *Dielectrophoresis : the behavior of neutral matter in nonuniform electric fields*, Cambridge monographs on physics (Cambridge University Press, Cambridge, 1978) includes bibliographical references and index.
- ³⁴L. Landau and E. Lifshitz, *Electrodynamics of Continuous Media* (Pergamon Press, 1984).
- ³⁵H. A. Pohl and I. Hawk, Separation of living and dead cells by dielectrophoresis, *Science* **152**, 647 (1966).
- ³⁶S. Gangwal, O. J. Cayre, and O. D. Velev, Dielectrophoretic assembly of metallodielectric Janus particles in AC electric fields, *Langmuir* **24**, 13312 (2008).
- ³⁷L. Hong, A. Cacciuto, E. Luijten, and S. Granick, Clusters of charged Janus spheres, *Nano Lett* **6**, 2510 (2006).
- ³⁸A. McMullen, M. Muñoz Basagoiti, Z. Zeravcic, and J. Brujic, Self-assembly of emulsion droplets through programmable folding, *Nature* **610**, 502 (2022).
- ³⁹A. Walther and A. H. E. Müller, Janus particles: Synthesis, self-assembly, physical properties, and applications, *Chem. Rev* **113**, 5194 (2013).
- ⁴⁰N. Lu, P. Zhang, Q. Zhang, R. Qiao, Q. He, H.-B. Li, Y. Wang, J. Guo, D. Zhang, Z. Duan, Z. Li, M. Wang, S. Yang, M. Yan, E. Arenholz, S. Zhou, W. Yang, L. Gu, C.-W. Nan, J. Wu, Y. Tokura, and P. Yu, Electric-field control of tri-state phase transformation with a selective dual-ion switch, *Nature* **546**, 124 (2017).
- ⁴¹F. Zhang, H. Zhang, S. Krylyuk, C. A. Milligan, Y. Zhu, D. Y. Zemlyanov, L. A. Bendersky, B. P. Burton, A. V. Davydov, and J. Appenzeller, Electric-field induced structural transition in vertical MoTe_2 - and $\text{Mo}_{1-x}\text{W}_x\text{Te}_2$ -based resistive memories, *Nat. Mater.* **18**, 55 (2019).
- ⁴²J. Hegseth and K. Amara, Critical temperature shift in pure fluid SF_6 caused by an electric field, *Phys. Rev. Lett.* **93**, 057402 (2004).
- ⁴³S. G. Moore, M. J. Stevens, and G. S. Grest, Liquid-vapor interface of the Stockmayer fluid in a uniform

- external field, *Phys. Rev. E* **91**, 022309 (2015).
- ⁴⁴K. A. Maerzke and J. I. Siepmann, Effects of an applied electric field on the vapor-liquid equilibria of water, methanol, and dimethyl ether, *J. Phys. Chem. B* **114**, 4261 (2010).
- ⁴⁵R. Evans, Fluids adsorbed in narrow pores: phase equilibria and structure, *J. Phys. Condens. Matter* **2**, 8989 (1990).
- ⁴⁶A. Z. Panagiotopoulos, Adsorption and capillary condensation of fluids in cylindrical pores by Monte Carlo simulation in the Gibbs ensemble, *Mol. Phys.* **62**, 701 (1987).
- ⁴⁷R. Valiullin, S. Naumov, P. Galvosas, J. Kärger, H.-J. Woo, F. Porcheron, and P. A. Monson, Exploration of molecular dynamics during transient sorption of fluids in mesoporous materials, *Nature* **443**, 965 (2006).
- ⁴⁸T. Horikawa, D. D. Do, and D. Nicholson, Capillary condensation of adsorbates in porous materials, *Adv. Colloid Interface Sci* **169**, 40 (2011).
- ⁴⁹M. E. Fisher and H. Nakanishi, Scaling theory for the criticality of fluids between plates, *J. Chem. Phys* **75**, 5857 (1981).
- ⁵⁰G. McHale, C. V. Brown, and N. Sampara, Voltage-induced spreading and superspreading of liquids, *Nat. Commun* **4**, 1605 (2013).
- ⁵¹D. C. Grahame, The electrical double layer and the theory of electrocapillarity., *Chem. Rev* **41**, 441 (1947).
- ⁵²F. Mugele and J.-C. Baret, Electrowetting: from basics to applications, *J. Phys.: Condens. Matter* **17**, R705 (2005).
- ⁵³C. Picard, V. Gérard, L. Michel, X. Cattoën, and E. Charlaix, Dynamics of heterogeneous wetting in periodic hybrid nanopores, *J. Chem. Phys* **154**, 164710 (2021).
- ⁵⁴R. E. Migacz, M. Castleberry, and J. T. Ault, Enhanced diffusiophoresis in dead-end pores with time-dependent boundary solute concentration, *Phys. Rev. Fluids* **9**, 044203 (2024).
- ⁵⁵A. Giacomello, M. Chinappi, S. Meloni, and C. M. Casciola, Metastable wetting on superhydrophobic surfaces: Continuum and atomistic views of the cassie-baxter–wenzel transition, *Phys. Rev. Lett* **109**, 226102 (2012).
- ⁵⁶S. Shin, E. Um, B. Sabass, J. T. Ault, M. Rahimi, P. B. Warren, and H. A. Stone, Size-dependent control of colloid transport via solute gradients in dead-end channels, *Proc. Natl. Acad. Sci. U.S.A* **113**, 257 (2016).
- ⁵⁷S. Shin, J. T. Ault, P. B. Warren, and H. A. Stone, Accumulation of colloidal particles in flow junctions induced by fluid flow and diffusiophoresis, *Phys. Rev. X* **7**, 041038 (2017).
- ⁵⁸S. Marbach and L. Bocquet, Osmosis, from molecular insights to large-scale applications, *Chem. Soc. Rev.* **48**, 3102 (2019).
- ⁵⁹T. Zimmermann, F. Sammüller, S. Hermann, M. Schmidt, and D. de las Heras, Neural force functional for non-equilibrium many-body colloidal systems, *Mach. Learn.: Sci. Technol.* **5**, 035062 (2024).
- ⁶⁰Y. Tsoni and L. Leibler, Phase-separation in ion-containing mixtures in electric fields, *Proc. Natl. Acad. Sci. U.S.A* **104**, 7348 (2007).
- ⁶¹A. T. Bui and S. J. Cox, Research data supporting “Dielectrocapillarity for exquisite control of fluid”. Zenodo., <https://doi.org/XX.XXXX/zenodo.XXXXX> (2025).
- ⁶²F. Sammüller, S. Robitschko, S. Hermann, and M. Schmidt, Hyperdensity functional theory of soft matter, *Phys. Rev. Lett.* **133**, 098201 (2024).
- ⁶³A. T. Bui, GCMC with Gaussian truncated potentials, <https://github.com/annatbui/GCMC> (2024).
- ⁶⁴A. P. Thompson, H. M. Aktulga, R. Berger, D. S. Bolintineanu, W. M. Brown, P. S. Crozier, P. J. in 't Veld, A. Kohlmeyer, S. G. Moore, T. D. Nguyen, R. Shan, M. J. Stevens, J. Tranchida, C. Trott, and S. J. Plimpton, LAMMPS - a flexible simulation tool for particle-based materials modeling at the atomic, meso, and continuum scales, *Comput. Phys. Commun* **271**, 108171 (2022).
- ⁶⁵F. Chollet, *Deep Learning with Python* (Manning Publications, 2017).
- ⁶⁶R. Evans and N. B. Wilding, Quantifying density fluctuations in water at a hydrophobic surface: Evidence for critical drying, *Phys. Rev. Lett.* **115**, 016103 (2015).

Response to reviewer comments for NCOMMS-25-37255-T “Dielectrocapillarity for exquisite control of fluids”

Anna T. Bui^{1,2} and Stephen J. Cox^{2, a)}

¹⁾Yusuf Hamied Department of Chemistry, University of Cambridge, Lensfield Road, Cambridge, CB2 1EW, United Kingdom

²⁾Department of Chemistry, Durham University, South Road, Durham, DH1 3LE, United Kingdom

(Dated: December 20, 2025)

We would like to thank all four reviewers for their time in reading our manuscript. In the last round of reviews, we addressed comments from reviewers 1 and 2 only. Aside from a minor suggestion from reviewer 2 (which we address below), we are delighted that both of the original reviewers recommend publication in Nature Communications. We are also grateful to both reviewers for acknowledging our efforts to thoroughly address their comments.

Our revised manuscript has also been sent to a further two peer reviewers. Overall, these reviews are positive, describing our work as *definitely novel and interesting*, echoing sentiments from the previous round where it was also made clear our work *addresses a critical gap*. Reviewers 3 and 4 did, however, raise extra concerns regarding the underlying assumptions of our theoretical models and the generality of our approach.

In response to these concerns, we have made further significant changes to our manuscript. This includes a new figure that clearly depicts an example of how the inhomogeneous fields that we consider can be realized, and how the cDFT provides insight.

Below we detail our responses to all comments from the reviewers point-by-point. The reviewers' comments are **reproduced in turquoise**. Changes are **highlighted in magenta**. The reference list from the main paper is replicated in full at the end the document; this is simply so citations in text quoted from the manuscript appear correctly.

^{a)}Electronic mail: stephen.j.cox@durham.ac.uk

Report of Reviewer 1

The revision is thorough and comprehensive, so my recommendation is to accept this manuscript for publication in Nature Communications in its present form.

We thank the reviewer for this positive feedback. As reviewer 1 recommends publication as is, we have not made any changes in response to their comment.

Report of Reviewer 2

I thank the authors for carefully addressing my comments. I think adding in water and contact angles were good ideas that make the results a bit easier to put in experimental perspective. The authors have also made significant efforts in making their manuscript more pedagogical. I would still like to raise a minor comment: In my view, "out-of-equilibrium" means that a source of energy is put in the system, in this case, it is the spatially varying electric field. So I was quite worried that the AI had been trained on data that was never under applied external fields. Indeed, that was not the case, as the authors explain in the methods. To me that is something important that enhances the validity (a priori) of the trained model and I would therefore encourage the authors to say that explicitly in the main text.

We thank the reviewer for this positive feedback and their helpful suggestion. In response, we now explicitly state in the main text that the ML model was trained on data generated under applied external electrostatic potentials:

(Lines 153–168) Going beyond established deep-learning approaches to cDFT, we capture electrostriction arising from the coupling between mass and charge density of the fluid by explicitly learning the "hyperfunctional" $\mathcal{F}_{\text{intr}}^{\text{ex}}([\rho, \beta\phi], T)$ where $\phi(\mathbf{r})$ is the inhomogeneous electrostatic potential and $\beta = 1/(k_{\text{B}}T)$ with k_{B} the Boltzmann constant, as recently introduced in Ref.³⁶. A practical limitation of this cDFT approach is that, at present, only inhomogeneities with planar symmetry can be investigated directly. As a result, the neural-network representation of this functional was trained on simulation data generated under random planar electrostatic potentials $\phi(z)$. Nonetheless, we demonstrate below that calculations in which ϕ only varies along a single cartesian direction provide general insight into the influence of EFGs on wetting and capillarity. Details of the practical implementation of the theory are given in the Methods section.

I would also like to congratulate the authors on so diligently (and also sensibly) answering both reviewers.

Apart from my minor comment above, I am happy to recommend this manuscript for publication in Nature Comm.

We would like to thank the reviewer for their careful assessment of our work and constructive criticisms. We are happy that they recommend our work for publication in Nature Communications.

Report of Reviewer 3

The manuscript deals with the topic of how electrical field gradients induce novel phenomena relating to phase behaviour, capillarity and adsorption in liquids. Focus of the work is in confined conditions and could have implications on porous materials and their use for energy storage applications etc.

The manuscript is well structured and provides a well thought of theoretical study. The authors have already addressed most open questions, and the manuscript stands on its own.

We thank the reviewer for the positive assessment and for recognizing the structure, clarity, and coherence of our study. We are encouraged that they believe we have addressed most open questions.

My concerns are in regards to not being able to come up or or understand an experiment to verify this theory (falsifiability). The example given in Fig. 4 is too simple and does not provide any sort of evidence that the theory is valid except showing that it fits a specific boundary condition. Due to the use of cDFT combined with machine learning methods, the approach seems like a black box and it is hard to understand its core assumptions (e.g. the choice of potentials at the start) and their validity. I would recommend publishing in a more specialised journal, and focusing future work to include geometries and conditions where it could be experimentally tested. I would recommend considering the impacts of the theory in also other geometries, e.g. nanopores and their use for sensing and filtration.

The reviewer essentially raises two issues in their comment: (i) experimental falsifiability; and (ii) the “black box” nature of using cDFT combined with ML.

To support point (i), they state that the example of dielectrowetting provided by Fig. 4 (now Fig. 5) is too simple, and that it only shows that the theory fits a specific boundary condition. We believe it is relevant to mention that we added this example at the last round of reviews in response to comments from reviewers 1 and 2, both of whom felt it does address the gap between our theory and experiment. (This is stated explicitly reviewer 2’s comments.) To respond to this comment more directly, within the cDFT framework we do not place boundary conditions on the fluid (in contrast to continuum modeling). Rather, the effects of the substrate—in this case an interdigitated electrode—are accounted for by the external potentials. The equilibrium *microscopic* number and charge densities are then obtained by minimizing the grand potential. We then obtain an estimate for the electrostatic free energy per unit area by integrating the product of microscopic charge density and the external electrostatic potential from the interdigitated electrode. We observe that this microscopic picture is consistent with the *macroscopic* trends observed experimentally (See Ref. 18). This is not a trivial result—it suggests that the cDFT is capturing the essence of the Maxwell stress tensor—the lens through which the experimental phenomena are analyzed—as an emergent phenomenon; we have not built this into the theory.

Nonetheless, we appreciate that the connection to experimental geometries can be made stronger, which is also a concern shared by reviewer 4. To this end, we have introduced a new figure (now Fig. 1). This simulation snapshot not only demonstrates how such inhomogeneous fields may arise, but also the influence they may have on the wetting properties of dielectric fluids. Specifically, the simulation snapshot shows water confined between inter-

digitated electrodes (Fig. 1). This geometry introduces strong, spatially varying electric fields, which are conceivably achievable through nanoscale fabrication or van der Waals heterostructure assembly:

- Han et al., Dielectric capacitors with three-dimensional nanoscale interdigital electrodes for energy storage, *Sci. Adv* 1, 9, (2015). <https://doi.org/10.1126/sciadv.1500605>
- Castellanos-Gomez, A., Duan, X., Fei, Z. et al. Van der Waals heterostructures. *Nat Rev Methods Primers* 2, 58 (2022). <https://doi.org/10.1038/s43586-022-00139-1>

To clarify the origin and structure of electric-field gradients in this geometry, we analyze the electrostatic potential and show that it naturally decomposes into sinusoidal modes along x and exponential decay along z . This directly motivates the understanding the capillarity physics in simplified planar field geometries of sinusoidal nature (Figs. 2, 3, 4) and of exponential nature in Fig 5. We also discuss in more detail the physical interpretation of probing the response of the fluid to planar inhomogeneous electric fields.

With regard to changes in the manuscript in response to point (i) raised by the reviewer, we have introduced a new **Figure 1**:

Figure 1: Inhomogeneous electric fields arising from interdigitated electrodes strongly influence water’s wetting behavior. Snapshot of an SPC/E water simulation in a hydrophobic slit with alternating positive and negative electrode patches. Either holding the electrodes at a 10 V potential difference or attributing a fixed charge of $\pm 0.05 e/\text{atom}$ causes the fluid to exhibit enhanced wetting at the walls, accompanied by strong lateral density oscillations. Cross-sections of the electrostatic potential in the constant charge setup are shown parallel to the surface (top right) and normal to the surface (bottom left).

We also make several changes throughout the manuscript:

(Lines 95–131) Although molecular simulations cannot fully capture the influence of EFGs on dielectric fluids, they offer vivid qualitative insights into the underlying physics

and illustrate how EFGs may arise within nanoscale devices, as illustrated in Fig. 1. Here, we present a simulation snapshot of water at 300 K confined between two hydrophobic substrates that have been patterned with alternating stripes of positive and negative charge—a set up that provides a caricature of an interdigitated electrode. As can be clearly seen in Fig. 1, charging this device induces local wetting near the charged stripes, with pronounced density variations along the direction parallel to the surface (x).

These features arise from the complex interplay between the fluid’s charge and number densities, and their collective response to EFGs, arising from the inhomogeneous electrostatic potential $\phi(x, z)$, where z is along the surface normal. It is instructive to consider the form of ϕ far from either surface; here, it will resemble the linear superposition of the asymptotic limit of the two substrates considered independently,

$$\phi_{\text{single}}(x, z) \sim \phi_0 \sin\left(\frac{2\pi x}{L_x}\right) \exp\left(\frac{-2\pi|z - z_s|}{L_x}\right), \quad (1)$$

where z_s indicates the plane of the substrate’s outermost atoms. We therefore observe that inhomogeneity of ϕ is characterized by a sinusoidal oscillation of period L_x along x , and an exponential decay along from the surface along z . This asymptotic analysis captures the essential behavior of $\phi(x, z)$ computed explicitly from the potential energy of a test charge, as seen in Fig. 1.

The pronounced wetting behavior observed with simulation in Fig. 1 strongly suggests that such EFGs will influence capillarity of the polar fluid; that is, the amount of fluid adsorbed at constant chemical potential. Addressing this issue, however, demands an accurate and efficient framework for determining structure, thermodynamics, and phase behavior in an open system—this lies beyond the practical limits of present day molecular simulations. Instead, we turn to classical density functional theory (cDFT), an exact statistical mechanical framework for inhomogeneous fluids.

(Lines 211–226) While EFGs will be most pronounced near the surfaces that generate them, Fig. 1 demonstrates that they may persist relatively far from the interface. In the specific case considered in Fig. 1, midway between the substrates we observe a sinusoidal electrostatic potential along the x direction. This motivates us to understand the direct influence of such sinusoidal potentials on the fluid, without explicitly considering interfaces. In fact, for L_x larger than a few molecular diameters, we can define bulk response if we consider averages over a thin slice of thickness $\Delta z \ll L_x$ (Fig. S16). Although such a clean separation of bulk and interfacial response becomes challenging as L_x approaches molecular length scales, the full potential (as opposed to its asymptotic form) comprises modes of decreasing wavelength—it is therefore instructive to understand bulk-like response across a broad range of wavelengths.

(Lines 396–402) To gain insight into the influence of EFGs, we perform a “computational experiment” in which we apply a sinusoidal electric field across the slit. While such a set up does not correspond to an EFG established by the substrate walls themselves (see, e.g., Fig. 1), it does allow us to assess the effects of a particular mode. The impact is twofold: (1) condensation shifts to more negative $\Delta\mu$...

(Lines 429–442) On the macroscopic scale, these effects manifest in electrowetting¹⁹ and dielectrowetting^{18,20}, where the contact angle of a droplet can be tuned with applied

potentials. Our nanoscale simulations in Fig. 1 show that a similar phenomenology emerges under confinement: wetting is strongly enhanced directly over the electrode patches, while the electrostatic potential generated by the interdigitated electrodes decays into the slit. Following previous experimental work^{18,20}, this motivates us employ a simple planar electrostatic potential, $\phi(z) = \phi_0 \exp(-2\pi z/L_x)$ to probe wetting at the nanoscale. Similar to our arguments above, results obtained from such a potential can be considered to report on average behavior in a slice of thickness $\Delta x \ll L_x$ (Fig. S16).

Regarding point (ii), the reviewer states that the use cDFT combined with machine learning methods, makes the approach seem like a black-box, and that the core assumptions, e.g., of the choice of potentials, and their validity are hard to understand. On the choice of the external potentials that we use, this has been addressed above. Where the cDFT+ML approach is concerned, the theoretical foundations and assumptions are detailed in methodological papers that we have cited:

- Sammüller et al., Neural functional theory for inhomogeneous fluids: Fundamentals and applications, Proc. Natl. Acad. Sci. U.S.A 120, 50, (2023). <https://doi.org/10.1073/pnas.2312484120>
- Bui, Cox, Learning Classical Density Functionals for Ionic Fluids, Phys. Rev. Lett. 134, 148001 (2025). <https://doi.org/10.1103/PhysRevLett.134.148001>
- Sammüller et al., Neural Density Functional Theory of Liquid-Gas Phase Coexistence, Phys. Rev. X 15, 011013 (2025). <https://doi.org/10.1103/PhysRevX.15.011013>
- Bui, Cox, A first-principles approach to electromechanics in liquids, J. Phys.: Condens. Matter 37, 285101 (2025). <https://doi.org/10.1088/1361-648X/ade7e7>

To emphasize the theoretical rigor of the cDFT approach, and to guide the reader more directly to the relevant methodological papers, we now make these previous works more prominent in the manuscript:

(Lines 132–168) Within cDFT, the equilibrium structure and thermodynamics of a fluid can be determined from first principles by its excess intrinsic Helmholtz free energy functional $\mathcal{F}_{\text{intr}}^{\text{ex}}([\rho], T)$, where $\rho(\mathbf{r})$ is the average inhomogeneous density of the fluid and T is the temperature. This central result, established in Ref.³⁰, places cDFT as the liquid-state generalization of its celebrated electronic structure counterpart³¹—as a modern theory for inhomogeneous fluids. cDFT is naturally formulated in the grand canonical ensemble, where the chemical potential acts as the control variable governing particle exchange in confined systems. The chemical potential, μ , maps directly onto the relative humidity or vapor pressure for gases, and chemical activity for liquids.

In practice, the exact form of $\mathcal{F}_{\text{intr}}^{\text{ex}}([\rho], T)$ is generally unknown. Instead of relying on traditional approximations, we leverage state-of-the-art data-driven methodologies to supervise-learn functional mappings directly from quasi-exact reference data from grand canonical Monte Carlo simulations^{32–34}. This machine-learned cDFT framework has already been successfully applied to liquid–gas coexistence³³, liquid–liquid phase separation³⁵, and the electric double layer³⁴. Going beyond established deep-learning approaches to cDFT, we capture electrostriction arising from the coupling between mass and charge density of the fluid by explicitly learning the “hyperfunctional” $\mathcal{F}_{\text{intr}}^{\text{ex}}([\rho, \beta\phi], T)$ where $\phi(\mathbf{r})$ is the inhomogeneous electrostatic potential and $\beta = 1/(k_{\text{B}}T)$ with k_{B}

the Boltzmann constant, as recently introduced in Ref.³⁶. A practical limitation of this cDFT approach is that, at present, only inhomogeneities with planar symmetry can be investigated directly. As a result, the neural-network representation of this functional was trained on simulation data generated under random planar electrostatic potentials $\phi(z)$. Nonetheless, we demonstrate below that calculations in which ϕ only varies along a single cartesian direction provide general insight into the influence of EFGs on wetting and capillarity. Details of the practical implementation of the theory are given in the Methods section.

We again thank the reviewer for their comments. We believe that the resulting discussion concerning the inhomogeneous fields that we apply strengthen the paper. We also believe it has been helpful to provide more context to the cDFT+ML in terms of other work.

Report of Reviewer 4

In this article, the authors use ML-based numerical methods to study a novel phenomenon, dielectrocapillarity, where a spatially-varying electric field is used to modulate the density of a dipolar fluid. More precisely, they use a neuron network to learn thermodynamic functionals from MD simulations, and then use these functionals to (1) estimate the density profile of a bulk liquid under a electric field gradient, notably with consequences on the liquid-gas phase transition, (2) fluid entry into a small pore and (3) model electrowetting.

The effects reported here are definitely novel and promising, both in terms of technique and in relevance with experiments. My main concerns, as detailed below, regard (1) the clarity of certain aspects of the work (notably figures) and (2) the highly idealized settings considered here, which contrast with the bold claims throughout the article. Overall, though, I support publication of this work in Nature Communications.

We thank the reviewer for their careful reading of our manuscript and for the constructive comments. We appreciate the positive assessment of the novelty, relevance, and methodological contributions of the work. Below, we outline our responses to each major point and the changes implemented in the revised manuscript.

1) I find it hard to understand the general procedure behind this work, in particular with what relates to geometry, which is only detailed in SI file (note a typo p8 of SI: $L_x = L_z = 80 \text{ \AA}$ should, I imagine, read $L_x = L_y = 80 \text{ \AA}$). If I understand correctly, the authors generate MD trajectories with a slab geometry ($L_z \ll L_x, L_y$), from which they compute thermodynamic functionals, which are then learnt by a neuron network. How generalizable are the results to other geometries? The electrowetting proof-of-concept shows that changing the shape of the gradient is OK; but could, for example, be the learnt functionals be used to cases where the electric field gradient depends on more than one variable? Or non-slab geometries?

The issue of the geometry is a fair point. As is hopefully clear from our response to reviewer 3, we have made significant efforts to explain how the results from our cDFT study are relevant to realizable geometries. In addition to the general point of geometry, the reviewer raises the specific question of whether the learned functionals can be used where the electric field depends on more than one variable, or non-slab geometries.

To directly address the specific points, we first confirm that the reviewer's understanding is correct: we do indeed train the functionals on slab geometries in which the electrostatic potential only varies along z . The reasons for this are twofold: (i) the typical ML architectures used are limited to planar inhomogeneous geometries; and (ii) by averaging over x and y , we obtain sufficiently converged density profiles that form our training data.

While recent work for truly 2D systems demonstrates that the ML architecture is surmountable:

- Glitsch et al., Neural density functional theory in higher dimensions with convolutional layers, Phys. Rev. E 111, 055305, (2025). <https://doi.org/10.1103/PhysRevE.111.055305>

Obtaining converged density profiles in 3D for a sufficiently large training set remains a daunting task, even if consider “tricks” such as sampling the force (<https://doi.org/10.1063/5.0029113>).

It is important to emphasize, however, that even though we only explicitly sample planar inhomogeneities, we can nonetheless obtain broader insight. The fact that that such functionals

can be used to obtain bulk phase diagrams provides one clear example. (We note that obtaining bulk phase diagrams has been demonstrated previously by others for simpler systems^{33,35}.) Moreover, by analyzing interfacial free energies in planar geometries, information on wetting phenomena in 3D systems can be obtained, as we do in Fig. 5. (Again, analysis in this spirit has previously been performed by others for simpler systems. See, e.g. Ref. 35.)

Our general conclusions about dielectrocapillarity should, therefore, be generalizable to other geometries. This is supported by our new simulation example of water confined between interdigitated electrodes (Fig. 1). As discussed in response to reviewer 3, the electrostatic potential in this case decomposes into sinusoidal modes in x and exponential decay in z . It is clear from the snapshot that we observe enhanced wetting at the wall. It is also clear from Fig. S15 (new), that the density in a thin slice above the electrodes is very similar to that obtained with a planar external electrostatic potential. Furthermore, we also show how simulations with the planar sinusoidal potentials report on the bulk response in the interdigitated electrode simulation (see Figs. S16 and S17).

In light of the reviewer’s comment, we have made several changes throughout the manuscript:

Figure 1: Inhomogeneous electric fields arising from interdigitated electrodes strongly influence water’s wetting behavior. Snapshot of an SPC/E water simulation in a hydrophobic slit with alternating positive and negative electrode patches. Either holding the electrodes at a 10 V potential difference or attributing a fixed charge of $\pm 0.05 e/\text{atom}$ causes the fluid to exhibit enhanced wetting at the walls, accompanied by strong lateral density oscillations. Cross-sections of the electrostatic potential in the constant charge setup are shown parallel to the surface (top right) and normal to the surface (bottom left).

(Lines 95–131) Although molecular simulations cannot fully capture the influence of EFGs on dielectric fluids, they offer vivid qualitative insights into the underlying physics and illustrate how EFGs may arise within nanoscale devices, as illustrated in Fig. 1. Here, we present a simulation snapshot of water at 300 K confined between two hydrophobic substrates that have been patterned with alternating stripes of positive and negative

charge—a set up that provides a caricature of an interdigitated electrode. As can be clearly seen in Fig. 1, charging this device induces local wetting near the charged stripes, with pronounced density variations along the direction parallel to the surface (x).

These features arise from the complex interplay between the fluid’s charge and number densities, and their collective response to EFGs, arising from the inhomogeneous electrostatic potential $\phi(x, z)$, where z is along the surface normal. It is instructive to consider the form of ϕ far from either surface; here, it will resemble the linear superposition of the asymptotic limit of the two substrates considered independently,

$$\phi_{\text{single}}(x, z) \sim \phi_0 \sin\left(\frac{2\pi x}{L_x}\right) \exp\left(\frac{-2\pi|z - z_s|}{L_x}\right), \quad (2)$$

where z_s indicates the plane of the substrate’s outermost atoms. We therefore observe that inhomogeneity of ϕ is characterized by a sinusoidal oscillation of period L_x along x , and an exponential decay along from the surface along z . This asymptotic analysis captures the essential behavior of $\phi(x, z)$ computed explicitly from the potential energy of a test charge, as seen in Fig. 1.

The pronounced wetting behavior observed with simulation in Fig. 1 strongly suggests that such EFGs will influence capillarity of the polar fluid; that is, the amount of fluid adsorbed at constant chemical potential. Addressing this issue, however, demands an accurate and efficient framework for determining structure, thermodynamics, and phase behavior in an open system—this lies beyond the practical limits of present day molecular simulations. Instead, we turn to classical density functional theory (cDFT), an exact statistical mechanical framework for inhomogeneous fluids.

(Lines 211–226) While EFGs will be most pronounced near the surfaces that generate them, Fig. 1 demonstrates that they may persist relatively far from the interface. In the specific case considered in Fig. 1, midway between the substrates we observe a sinusoidal electrostatic potential along the x direction. This motivates us to understand the direct influence of such sinusoidal potentials on the fluid, without explicitly considering interfaces. In fact, for L_x larger than a few molecular diameters, we can define bulk response if we consider averages over a thin slice of thickness $\Delta z \ll L_x$ (Fig. S16). Although such a clean separation of bulk and interfacial response becomes challenging as L_x approaches molecular length scales, the full potential (as opposed to its asymptotic form) comprises modes of decreasing wavelength—it is therefore instructive to understand bulk-like response across a broad range of wavelengths.

(Lines 396–402) To gain insight into the influence of EFGs, we perform a “computational experiment” in which we apply a sinusoidal electric field across the slit. While such a set up does not correspond to an EFG established by the substrate walls themselves (see, e.g., Fig. 1), it does allow us to assess the effects of a particular mode. The impact is twofold: (1) condensation shifts to more negative $\Delta\mu$...

(Lines 429–442) On the macroscopic scale, these effects manifest in electrowetting¹⁹ and dielectrowetting^{18,20}, where the contact angle of a droplet can be tuned with applied potentials. Our nanoscale simulations in Fig. 1 show that a similar phenomenology emerges under confinement: wetting is strongly enhanced directly over the electrode

patches, while the electrostatic potential generated by the interdigitated electrodes decays into the slit. Following previous experimental work^{18,20}, this motivates us employ a simple planar electrostatic potential, $\phi(z) = \phi_0 \exp(-2\pi z/L_x)$ to probe wetting at the nanoscale. Similar to our arguments above, results obtained from such a potential can be considered to report on average behavior in a slice of thickness $\Delta x \ll L_x$ (Fig. S16).

We also clarify in the main text the planar geometry used for MD training:

(**Lines 153–168**) Going beyond established deep-learning approaches to cDFT, we capture electrostriction arising from the coupling between mass and charge density of the fluid by explicitly learning the “hyperfunctional” $\mathcal{F}_{\text{intr}}^{\text{ex}}([\rho, \beta\phi], T)$ where $\phi(\mathbf{r})$ is the inhomogeneous electrostatic potential and $\beta = 1/(k_B T)$ with k_B the Boltzmann constant, as recently introduced in Ref.³⁶. A practical limitation of this cDFT approach is that, at present, only inhomogeneities with planar symmetry can be investigated directly. As a result, the neural-network representation of this functional was trained on simulation data generated under random planar electrostatic potentials $\phi(z)$. Nonetheless, we demonstrate below that calculations in which ϕ only varies along a single cartesian direction provide general insight into the influence of EFGs on wetting and capillarity. Details of the practical implementation of the theory are given in the Methods section.

We corrected the noted typo in the SI. We thank the reviewer for pointing this out.

(**Page S8**) To reduce error due to translating between ensembles, simulation boxes with larger lateral dimensions are used with $L_z = 20 \text{ \AA}$ and $L_x = L_y = 80 \text{ \AA}$.

We also added additional simulation results in subsection **S7.4. Interdigitated electrode simulations** in the SI:

Figure S15: Density profiles of water confined between nanoscale interdigitated electrodes. For both fixed-charge (top) and constant-potential (bottom) simulations, imposing a potential difference of 10 V or a surface charge of $\pm 0.05 e/\text{atom}$ enhances wetting at the walls and induces strong lateral density oscillations.

(Pages S19–S22) Interdigitated electrode simulations

Here we describe the simulation details corresponding to the snapshot in Fig. 1 of the main text. The simulated systems consist of 620 water molecules symmetrically confined between two solid substrates. Each substrate is composed of 2048 atoms arranged on a simple cubic lattice ($32 \times 8 \times 8$, lattice parameter 2.5 \AA). The orthorhombic simulation box has lateral dimensions $80 \times 20 \times 100 \text{ \AA}^3$.

All simulations were performed with the LAMMPS package⁶⁸. Water–water interactions were described using the SPC/E model³⁷, with molecular geometry constrained via the RATTLE algorithm⁷. Substrate atoms were held rigid at their lattice positions. Water oxygen atoms interact with substrate atoms through a 12–6 Lennard–Jones potential with $\epsilon_{\text{wf}} = 0.065 \text{ kJ mol}^{-1}$ and $\sigma_{\text{wf}} = 3.094 \text{ \AA}$, with all Lennard–Jones interactions truncated and shifted at 10 \AA . Electrostatic interactions were evaluated in real space up to 10 \AA , with long-ranged interactions treated using PPPM Ewald summation⁷, such that the RMS force error was 10^5 smaller than the Coulomb force between two unit charges at 10 \AA separation⁷.

Simulations were conducted in the canonical (NVT) ensemble at $T = 300 \text{ K}$, controlled using a Nosé–Hoover chain with 5 thermostats and a damping constant of 0.1 ps . Dynamics were propagated using the velocity Verlet algorithm with time-step of 1 fs . We performed both fixed-charge and constant-potential simulations. Constant-potential simulations employed the fluctuating-charge method^{7, 7}, in which the electrode atom charges are solved self-consistently at each timestep using the ELECTRODE package⁷, using Gaussian charge of width 0.554 \AA . Figure S15 shows the resulting water density distributions before and after charging the electrodes, for both fixed-charge and constant-potential conditions. We observed analogous behavior in simulations comprising 310 molecules of the simple dipolar fluid at a temperature of 500 K .

We can understand the form of the electrostatic potential arising from these substrates by approximating their inhomogeneous charge distribution as being confined to two planes at $z = \pm z_s$, where z is along the surface normal. Denoting the direction parallel to the surfaces as x , the electrostatic potential from one of these substrates is a linear combination of functions of the form $\sin(k_n x) \exp(-|k_n||z - z_s|)$, where $k_n = 2\pi n/L_x$ ($n = \pm 1, \pm 3, \dots$), where L_x is the period along x . Owing to current practical limitations of the neural functional techniques that we employ, we have focused on response to electrostatic potentials with planar symmetry, i.e., either $\sin(k_n x)$ or $\exp(-|k_n|z)$. Far from the surface, the potential will be determined by the asymptotic form,

$$\phi_{\text{single}}(x, z; z_s) \sim \phi_0 \sin\left(\frac{2\pi x}{L_x}\right) \exp\left(-\frac{2\pi|z - z_s|}{L_x}\right) \quad (3)$$

For our simulation set up, the electrostatic potential midway between the substrates will be well-approximated by $\phi(x, z; -z_s) + \phi(x, z; z_s)$.

As discussed in the main text, for L_x larger than a few molecular diameters, response to these planar potentials can be considered to report on averages over a thin slice of either $\Delta z \ll L_x$ or $\Delta x \ll L_x$. To verify the extent to which this holds, we first consider the simple polar fluid. In the central panel of Fig. S16, we compare results from: (i) the average density along z for a slice of thickness $\Delta x = 10 \text{ \AA}$ centered above the negatively charged patch (depicted by the black rectangle in the leftmost panel); and (ii) the average density along z from a simulation with planar inhomogeneous symmetry, where the electrostatic potential is approximated by $\phi(z) = \phi_0[\exp(-2\pi|z + z_s|/L_x) + \exp(-2\pi|z - z_s|/L_x)]$, and where the overall density is chosen to be the same as that

of the thin slice in case (i). As can be seen in the central panel, aside from a slight discrepancy far from the surface, the two sets of simulations agree well with each other. The rightmost panel shows results for a similar procedure, this time taking a thin slice centered at $z = 0$ (i.e., far from the surfaces) and with $\Delta z = 10 \text{ \AA}$, as depicted by the pink rectangle in the leftmost panel. In this case, the planar inhomogeneous simulation was performed with $\phi(x) = \phi_0 \sin(2\pi x/L_x)$. As can be seen in the rightmost panel, the two sets of simulations give near identical results.

In Fig. S17 we show an analogous set of results obtained for water. For the simulations where we compare the exponentially decaying potentials along z (i.e., the central panel in Fig. S17), we again see good agreement between the two sets of simulations. For the sinusoidal potentials (i.e., the rightmost panel in Fig. S17), the response to the planar inhomogeneous potentials have a pronounced asymmetry that is largely suppressed in the simulation with interdigitated electrodes. This difference in geometries indicates a correlation between water's response in the x and z directions that we cannot model directly within the current limitations of the neural functional approach. Nonetheless, the most salient aspects of electromechanical response are captured and, if anything, enhanced in the full simulation with interdigitated electrodes.

Figure S16: Response of a dipolar fluid to electric field gradients. Left: density profile of a supercritical dipolar fluid at $T = 500 \text{ K}$ confined between interdigitated electrodes. Middle/right: density responses in simulations with (middle) an exponentially decaying potential or (right) a sinusoidal potential. The cross-sections highlighted in the left panel map closely onto the corresponding reduced-potential simulations.

Figure S16: Response of a water to electric field gradients. Left: density profile of a supercritical SPC/E water at $T = 700 \text{ K}$ confined between interdigitated electrodes. Middle/right: density responses in simulations with (middle) an exponentially decaying potential or (right) a sinusoidal potential.

2) Related to the previous point, the authors state that their method greatly reduces computational time, by comparing a timescale to evaluate the NN to an MD simulation time. This

is trivial, since what matters here should instead be the total computational time (ie including training and testing), unless the neural functional can be used directly to cases with, for example, completely different geometries. Can the authors comment on that?

The computational advantage of the neural cDFT approach arises from several factors. First, the functional is trained locally, meaning that once learned, it can be applied to systems that are of larger length scales than those used in training. Although we do not fully exploit this property in the present work, it has been demonstrated in Refs.³² and³⁵.

For the purposes of our study, however, the central advantage is that the neural cDFT framework operates naturally in the grand canonical ensemble, giving us direct access to the chemical potential. More importantly, it also enables rapid and accurate evaluation of free energies, including grand potentials across different fluid configurations. This capability is essential for accurately mapping thermodynamics of liquid–vapor coexistence, adsorption, obtaining hysteresis loops by determining metastable phases, and wetting.

To put this in perspective, obtaining the results shown in Figs. 3, 4, and 5 (phase diagrams, adsorption isotherms, and contact-angle behavior) via explicit molecular simulation would require a combination of thermodynamic integration, enhanced sampling, and large-scale simulations. In contrast, the neural cDFT functional is trained using approximately 2000 molecular simulations of modest size (a few hundred molecules) with randomly generated external potentials—corresponding to a total computational cost of order $\sim 10^5$ CPU hours. Once trained, the functional can be reused *without retraining* to evaluate thousands of free energy calculations at near negligible cost.

We have now added this point when discussing the method's advantages

(Lines 169–192) The resulting cDFT framework is not only efficiently computable on standard hardware but also unparalleled in its ability to simultaneously capture microscopic fluid structure and mesoscopic phase behavior under arbitrary non-uniform electric fields. Unlike atomistic simulations, which are computationally prohibitive for such a broad exploration, our method achieves orders-of-magnitude speedup, completing each calculation in less than a minute without sacrificing quantitative accuracy. **Importantly, this computational efficiency is realized after a one-time training stage: once the functional has been learned, it can be used to perform thousands of free-energy calculations at negligible cost. This amortized advantage is what enables us to map complete phase diagrams, adsorption isotherms, and metastable fluid branches—tasks that would require enhanced sampling and thermodynamic integration if carried out by molecular simulation alone.** This efficiency enables an unprecedented, highly accurate mapping of a polar fluid's response to EFGs of varying strengths and wavelengths across different thermodynamic conditions, from supercritical to subcritical regimes, spanning both bulk and confined environments. With this powerful tool in hand, we now uncover emergent electrostrictive phenomena that arise from the complex interplay of thermodynamics, confinement, and response to EFGs.

3) I am relatively unconvinced by the type of electric field gradients that are considered here. While the numerical results seem fine and interesting, I doubt they really correspond to something technically achievable. They use molecular-scale gradients (wavelength comparable to the molecular size σ) while comparing to micron-scale experiments (local or interdigitated

electrodes, for example). That of course does not mean that considering short wavelengths is wrong, but I doubt the effect survives under molecular confinement as a surface provides interactions with liquid other than through a static electric field (adsorption, chemical defects, roughness, dielectric confinement, polarizability, etc.). In addition, under such confinement, I would expect the gradient to be along the surface (because of natural/engineered defects, for example), rather than normal to it; I do not see any way of replicating experimentally the situation of Fig 1 and 2.

We believe that we have addressed this point when answering Point 1 raised by the reviewer. To reiterate, the new simulation shown in Fig. 1 and its associated discussion both motivates and clarifies the form the sinusoidal potentials that we use. Regarding whether this is technically achievable at the nanoscale, we note that nanoscale interdigitated electrodes do exist. In addition, progress in van der Waals heterostructure assembly is rapid, and already being used to probe the dielectric properties of fluids under nanoscale confinement.

- Han et al., Dielectric capacitors with three-dimensional nanoscale interdigital electrodes for energy storage, *Sci. Adv* 1, 9, (2015). <https://doi.org/10.1126/sciadv.1500605>
- Castellanos-Gomez, A., Duan, X., Fei, Z. et al. Van der Waals heterostructures. *Nat Rev Methods Primers* 2, 58 (2022). <https://doi.org/10.1038/s43586-022-00139-1>
- Fumagalli et al., Anomalously low dielectric constant of confined water, *Science* 360, 1339 (2018). <https://doi.org/10.1126/science.aat4191>

For completeness, the changes made to the main text are reproduced below:

(Lines 95–131) Although molecular simulations cannot fully capture the influence of EFGs on dielectric fluids, they offer vivid qualitative insights into the underlying physics and illustrate how EFGs may arise within nanoscale devices, as illustrated in Fig. 1. Here, we present a simulation snapshot of water at 300 K confined between two hydrophobic substrates that have been patterned with alternating stripes of positive and negative charge—a set up that provides a caricature of an interdigitated electrode. As can be clearly seen in Fig. 1, charging this device induces local wetting near the charged stripes, with pronounced density variations along the direction parallel to the surface (x).

These features arise from the complex interplay between the fluid's charge and number densities, and their collective response to EFGs, arising from the inhomogeneous electrostatic potential $\phi(x, z)$, where z is along the surface normal. It is instructive to consider the form of ϕ far from either surface; here, it will resemble the linear superposition of the asymptotic limit of the two substrates considered independently,

$$\phi_{\text{single}}(x, z) \sim \phi_0 \sin\left(\frac{2\pi x}{L_x}\right) \exp\left(\frac{-2\pi|z - z_s|}{L_x}\right), \quad (4)$$

where z_s indicates the plane of the substrate's outermost atoms. We therefore observe that inhomogeneity of ϕ is characterized by a sinusoidal oscillation of period L_x along x , and an exponential decay along from the surface along z . This asymptotic analysis captures the essential behavior of $\phi(x, z)$ computed explicitly from the potential energy of a test charge, as seen in Fig. 1.

The pronounced wetting behavior observed with simulation in Fig. 1 strongly suggests that such EFGs will influence capillarity of the polar fluid; that is, the amount of fluid

adsorbed at constant chemical potential. Addressing this issue, however, demands an accurate and efficient framework for determining structure, thermodynamics, and phase behavior in an open system—this lies beyond the practical limits of present day molecular simulations. Instead, we turn to classical density functional theory (cDFT), an exact statistical mechanical framework for inhomogeneous fluids.

(Lines 211–226) While EFGs will be most pronounced near the surfaces that generate them, Fig. 1 demonstrates that they may persist relatively far from the interface. In the specific case considered in Fig. 1, midway between the substrates we observe a sinusoidal electrostatic potential along the x direction. This motivates us to understand the direct influence of such sinusoidal potentials on the fluid, without explicitly considering interfaces. In fact, for L_x larger than a few molecular diameters, we can define bulk response if we consider averages over a thin slice of thickness $\Delta z \ll L_x$ (Fig. S16). Although such a clean separation of bulk and interfacial response becomes challenging as L_x approaches molecular length scales, the full potential (as opposed to its asymptotic form) comprises modes of decreasing wavelength—it is therefore instructive to understand bulk-like response across a broad range of wavelengths.

(Lines 396–402) To gain insight into the influence of EFGs, we perform a “computational experiment” in which we apply a sinusoidal electric field across the slit. While such a set up does not correspond to an EFG established by the substrate walls themselves (see, e.g., Fig. 1), it does allow us to assess the effects of a particular mode. The impact is twofold: (1) condensation shifts to more negative $\Delta\mu$...

(Lines 429–442) On the macroscopic scale, these effects manifest in electrowetting¹⁹ and dielectrowetting^{18,20}, where the contact angle of a droplet can be tuned with applied potentials. Our nanoscale simulations in Fig. 1 show that a similar phenomenology emerges under confinement: wetting is strongly enhanced directly over the electrode patches, while the electrostatic potential generated by the interdigitated electrodes decays into the slit. Following previous experimental work^{18,20}, this motivates us employ a simple planar electrostatic potential, $\phi(z) = \phi_0 \exp(-2\pi z/L_x)$ to probe wetting at the nanoscale. Similar to our arguments above, results obtained from such a potential can be considered to report on average behavior in a slice of thickness $\Delta x \ll L_x$ (Fig. S16).

4) Similarly, for Fig 3-4 I would expect that chemical effects (eg acid-base reactions with the wall) play a much more important role at the considered lengthscales, and that modelling the walls with a static external electric field is plain wrong. Another thing that comes to my mind is that for this kind of questions (wettability and fluid filling of a small pore), the MD simulations used as training data are typically quite bad - it is for example very hard to correctly estimate a contact angle from simulations of water+LJ walls, and in nanometric slit, slightly changing the LJ parameters of the wall can result in either full wetting or dewetting, at odds with experimental results eg on carbon slits. Here, the authors choose one particular value of the walls LJ coefficients, so I would be very cautious about the results, which could be of second-order magnitude compared to changing interaction parameters.

We agree that chemical specificity and the strength of attractive interactions between the fluid and the wall play an important role in determining the contact angle and, as the reviewer correctly points out, small changes in these parameters can have profound impact on the observed wetting/drying behavior. Establishing precise values for, e.g., the strength of the wall-fluid interaction would be essential if our aim was to reproduce, say, the contact angle of a specific system. But this is not our aim. Rather, within the broader context of capillarity physics, it is well established that wetting and capillary evaporation/condensation are primarily governed by three factors:

1. the wall–fluid interaction strength;
2. the thermodynamic state of the fluid relative to liquid–vapor coexistence;
3. the confinement length scale.

These principles trace back to classical results such as the Kelvin equation, extensive seminal cDFT studies, and many modern simulation studies; a particularly clear presentation on the topic can be found at

- Evans and Marconi, Phase equilibria and solvation forces for fluids confined between parallel walls, *J. Chem. Phys.* 86, 71380–7148 (1987). <https://doi.org/10.1063/1.452363>
- Evans, Fluids adsorbed in narrow pores: phase equilibria and structure, *J. Phys. Condens. Matter* 2, 8989 (1990). <https://doi.org/10.1088/0953-8984/2/46/001>

Our goal is not to model a specific substrate, but to demonstrate that, for polar fluids, there exists an additional and previously underexplored mode of control—namely, dielectrophoretic coupling to electric field gradients—which operates alongside the conventional determinants of capillarity. Framed another way: The influence of wall–fluid interactions on wetting has already been extensively characterized for both simple liquids and water (see Refs.⁷¹ and¹³), and our aim here is to build on this foundation by isolating the dielectrophoretic contribution.

To address this comment, we now provide broader context to the physics of capillarity at the following points in the manuscript:

(Lines 29–40) It is well established—initially at the macroscopic scale through the works of Young, Laplace, and Kelvin^{11,12}, and later at the microscopic scale^{13,14}—that adsorption depends not only on the system’s thermodynamic state, but also on the confinement length and the substrate–fluid interaction. These factors are typically intrinsic material properties. As a result, extensive research into enhancing adsorption in porous materials has focused on optimizing these factors, e.g., by tuning porosity or chemical functionalization¹⁵. However, the potential for manipulation by external means—using applied fields to control confined fluids—remains relatively unexplored.

(Lines 377–383) In the absence of an applied external field, it is well-established that capillarity is controlled by the chemical potential of the reservoir, the length scale of confinement, the substrate–fluid interaction and temperature¹³. With EFGs, we introduce an additional experimental handle by which to control fluid adsorption behavior; we call this new phenomenon “dielectrocapillarity.”

5) Some of the results discussed here are in the large length scale regime, eg Janus colloids. I am afraid this limit is quite a dangerous one to take: colloids under an applied field are basically never at thermal equilibrium (because of electrophoresis, for example), so the current framework would not apply, even if it was extended to include dynamical effects (one would need to add hydrodynamics, ions, etc.). While the thought experiment presented in lines 208-226 is conceptually interesting, it is important to signal such limitations.

We believe that in the previous version of the manuscript likely gave the impression that the sinusoidal inhomogeneous fields were likely to arise under nonequilibrium conditions, such as an AC experiment. We hope that the new discussion surrounding Fig. 1 makes clear that such inhomogeneous fields can in principle be achieved under equilibrium conditions — for the field arising from a substrate with a static charge distribution, or held at constant potential difference, there is no reason to assume *a priori* that the fluid will not reach an equilibrium state. (Any steady state current in this system would violate the underlying symmetry.)

In light of the reviewer's comments, we have changed the wording in the manuscript to emphasize that our comment pertains to an equilibrium effect, and we have also softened the language.

(Lines 302–310) In contrast, the short-ranged colloidal system lacks this screening, leading to a much stronger response (dashed purple lines, Fig. 3b and inset). Such an equilibrium effect could be leveraged for programmable directed self-assembly⁴⁷, where EFGs in combination with tunable Janus particle surfaces⁴⁸ may provide a powerful tool for tailoring the assembly of extended structures with dielectrophoretic forces, with additional tunability arising from the solvent and ionic strength.

6) Line 28: Fundamentally, fluid uptake in these systems is driven by capillary forces Not really. Separation and filtration processes are typically out of equilibrium, with (electro/diffusio)osmotic effects often playing a major role.

Our intention was not to imply that practical separation or filtration operates purely under equilibrium conditions, but rather to highlight the underlying thermodynamic mechanism relevant for our study. We have therefore revised the sentence to read:

(Lines 27–29) The physics of capillarity plays a fundamental role in determining fluid uptake in these systems.

7) Line 40: While a uniform field exerts a force only on free charges such as ions This is misleading; fundamentally this force is cancelled and fully transmitted to the solvent because of viscous dissipation. In almost all nanofluidic experiments, applying a uniform electric field results in a net flow of water (because of electro-osmosis and related phenomena). In typical slit-like geometries like the one considered here, the effect is often dominant compared to other transport phenomena. See for example the 2021 review by Kavokine, Netz and Bocquet.

We have revised this sentence to clarify that we are referring to the direct interaction between the fluid and the field (i.e., $F_{\text{ext}} = qE_{\text{ext}}$):

(Lines 43–46) While a uniform field exerts a direct force only on free charges such as ions, non-uniform fields with electric field gradients (EFGs) generate dielectrophoretic forces on neutral polar molecules.

8) I find Fig 2 difficult to follow due to the high number of curves, colors, line styles, etc. In particular, I do not fully understand Fig 2d. What is represented as inset? What are the two blue and red rectangles on the right?

The inset in Fig. 2d was originally intended to illustrate the density profiles of the corresponding gas and liquid phases under no field and under a field gradient. Since this information is already presented in Fig. 2c, we have removed the insets from Fig. 2d. The blue and red boxes indicate the supercritical and subcritical temperatures for which the responses in Figs. 2a-c are shown. We have revised the figure to clarify this distinction. This is now Figure 3 in the revised manuscript.

Figure 3: Controlling liquid–vapor equilibrium with EFGs. (a) At $T^* > T_{c,0}^*$ where $T_{c,0}^* \approx 1.97$, increasing EFGs by independently varying λ and E_{\max}^* amplifies dielectrophoretic rise. (b) The local electrostrictive response of the dipolar fluid, as measured by the rise in the maximum density peak relative to zero field $\Delta\rho_{\max}^*$, is highly non-linear. Solid and dashed lines show the response of systems with dipolar interactions that are long-ranged (LR), e.g., polar molecules, and screened short-ranged (SR), e.g., colloids, respectively. The effect of LR interactions becomes pronounced for $\lambda \gg \sigma$. The SR fluid is a nearly identical dipolar fluid, but whose Coulomb potential is replaced by $\text{erfc}(\kappa r)/r$ where $\kappa^{-1} = 1.5\sigma$. (c) At an isotherm where $T^* < T_{c,0}^*$, stable solutions for the density $\rho^*(z)$ under a sinusoidal electric field with $\lambda/\sigma = 1.7$, are shown for different values of the chemical potential. These results are used to investigate liquid–vapor coexistence. (d) Results in light pink show the binodal of the dipolar fluid in the absence of an electric field. At $E_{\max}^* = 4$, T_c^* shifts to a lower temperature, as seen in the binodal in dark purple. Solid symbols show results obtained from the multiscale cDFT approach, while crosses indicate estimates of T_c^* using the law of rectilinear diameters and critical exponents¹⁴. Solid lines serve as a guide to the eye.

9) For all figures, some of the pastel colors are difficult to see after printing.

We updated the figures to use darker colors for better visibility in print.

10) Line 322: That is, EFGs impact criticality of polar fluids in both the three-dimensional and confined two-dimensional Ising universality classes I do not understand this sentence, I would suggest explaining more. I guess that the authors are referring to the fact that the liquid-gas phase transition corresponds to the 3D Ising class; but I am less familiar with the capillary transition and I would imagine that it does not belong to a single universality class but rather would depend on eg. the pores geometry. In practice, you would likely need to have a wedge/1D/0D geometry to have a localized electric field in the confined system, so I do not think it is realistic to have the exact 2D regime.

I also do not understand what this discussion is bringing here; if the observed effect modifies the system's chemical potential, it is obvious that it will affect both phase transitions, which are defined from it. It is not particularly surprising that modifying the system's parameters shifts the critical points; what would be more relevant would be to discuss orders of magnitude, like the 50K shift in the case of the liquid-gas transition of water. In addition to this, the authors could discuss what orders of magnitude can be reached in terms of electric field (gradient) experimentally. They list a value of $E = 0.4 \text{ V/\AA}$.

To get this kind of field from say a point charge of charge Zq - say surface defects - in water over 1nm (which corresponds roughly to the setting considered here), you would need $Z = 10$, which is a bit wild.

In the interests of a general readership, we have removed the statement about the 2D/3D universality. It is important to emphasize, however, that we do not modify the system's chemical potential — we are working in the grand canonical ensemble in which the chemical potential is held constant. The EFGs introduce an additional electrostatic contribution to the free energy, but they are conjugate to the charge density, unlike the chemical potential, which is conjugate to the one-body density.

Regarding orders of magnitude, the field gradients used in our study correspond to modest charge densities in the underlying MD simulations: on the order of $0.05 e/\text{atom}$, or potential difference of 10 V across nanometric distances. These values are within the range generated near charged surfaces and patterned electrodes at the nanoscale. (Refs. ^{17,18,24})

(Caption to figure 1) Inhomogeneous electric fields arising from interdigitated electrodes strongly influence water's wetting behavior. Snapshot of an SPC/E water simulation in a hydrophobic slit with alternating positive and negative electrode patches. Either holding the electrodes at a 10 V potential difference or attributing a fixed charge of $\pm 0.05 e/\text{atom}$ causes the fluid to exhibit enhanced wetting at the walls, accompanied by strong lateral density oscillations. Cross-sections of the electrostatic potential in the constant charge setup are shown parallel to the surface (top right) and normal to the surface (bottom left).

11) Related to the previous point, it seems to me that the only two relevant regimes for how to create an electric field gradient within a confined pore are (1) electric field gradient occurring on a molecular lengthscale close to the walls or (2) a gradient with a lengthscale equal to the system size. The authors do not really discuss the impact of the system size in the settings

corresponding to Fig 3.

The reviewer is correct to point out that the confinement length can play an important role. As discussed in our response to Point 4, the confinement length is indeed one of the key determinants of capillarity, and its influence on wetting and condensation has been extensively characterized in prior studies of simple and polar fluids. In the present work, our aim is to isolate the role of dielectrophoretic forces, and for this reason we focus on a moderately wide solvophobic slit, $H = 6.6\sigma$, where the two walls are sufficiently separated that confinement effects do not overwhelm the electric field-induced response.

It would be interesting to investigate any interplay between the length and the field; it lies beyond the scope of the current paper, however.

For completeness, we list the relevant changes to the manuscript. (They are the same as those in Point 4.)

(Lines 29–40) It is well established—initially at the macroscopic scale through the works of Young, Laplace, and Kelvin^{11,12}, and later at the microscopic scale^{13,14}—that adsorption depends not only on the system’s thermodynamic state, but also on the confinement length and the substrate–fluid interaction. These factors are typically intrinsic material properties. As a result, extensive research into enhancing adsorption in porous materials has focused on optimizing these factors, e.g., by tuning porosity or chemical functionalization¹⁵. However, the potential for manipulation by external means—using applied fields to control confined fluids—remains relatively unexplored.

(Lines 377–383) In the absence of an applied external field, it is well-established that capillarity is controlled by the chemical potential of the reservoir, the length scale of confinement, the substrate–fluid interaction and temperature¹³. With EFGs, we introduce an additional experimental handle by which to control fluid adsorption behavior; we call this new phenomenon “dielectrocapillarity.”

We also clarified the electrostatic potential lengthscale in relation to the interdigitated electrode set-up.

(Lines 430–443) On the macroscopic scale, these effects manifest in electrowetting¹⁹ and dielectrowetting^{18,20}, where the contact angle of a droplet can be tuned with applied potentials. Our nanoscale simulations in Fig. 1 show that a similar phenomenology emerges under confinement: wetting is strongly enhanced directly over the electrode patches, while the electrostatic potential generated by the interdigitated electrodes decays into the slit. Following previous experimental work^{18,20}, this motivates us employ a simple planar electrostatic potential, $\phi(z) = \phi_0 \exp(-2\pi z/L_x)$ to probe wetting at the nanoscale. Similar to our arguments above, results obtained from such a potential can be considered to report on average behavior in a slice of thickness $\Delta x \ll L_x$ (Fig. S16).

(Lines 456–462) Our multiscale framework provides a microscopic perspective on this phenomenology, and allows us to test whether, at the nanoscale, EFGs indeed enhance the wetting of dielectric liquids and the extent to which the scaling prescribed by Eq. ?? holds. To this end, we applied $\phi(z)$ with $L_x \approx 7\sigma$ to the confined dipolar model, symmetrically from both walls of a solvophobic slit.

We again thank the reviewer for their constructive comments. In response, we have added new molecular simulations that explicitly connect our model electric field gradients to experimentally relevant geometries, notably interdigitated electrodes. We have also clarified the role of geometry and length scales throughout the manuscript, emphasizing both the scope and limitations of the present approach. Finally, we have expanded the discussion to place dielectrocapillarity within the established historical framework of capillary physics, highlighting it as an additional control mechanism alongside confinement, thermodynamic state, and wall–fluid interactions.

REFERENCES

- ¹Z. Chen, P. Li, R. Anderson, X. Wang, X. Zhang, L. Robison, L. R. Redfern, S. Moribe, T. Islamoglu, D. A. Gómez-Gualdrón, T. Yildirim, J. F. Stoddart, and O. K. Farha, Balancing volumetric and gravimetric uptake in highly porous materials for clean energy, *Science* **368**, 297 (2020).
- ²J. Chmiola, G. Yushin, Y. Gogotsi, C. Portet, P. Simon, and P. L. Taberna, Anomalous increase in carbon capacitance at pore sizes less than 1 nanometer, *Science* **313**, 1760 (2006).
- ³X. Liu, D. Lyu, C. Merlet, M. J. A. Leesmith, X. Hua, Z. Xu, C. P. Grey, and A. C. Forse, Structural disorder determines capacitance in nanoporous carbons, *Science* **384**, 321 (2024).
- ⁴Q. Yang, P. Z. Sun, L. Fumagalli, Y. V. Stebunov, S. J. Haigh, Z. W. Zhou, I. V. Grigorieva, F. C. Wang, and A. K. Geim, Capillary condensation under atomic-scale confinement, *Nature* **588**, 250 (2020).
- ⁵L. Fumagalli, A. Esfandiari, R. Fabregas, S. Hu, P. Ares, A. Janardanan, Q. Yang, B. Radha, T. Taniguchi, K. Watanabe, G. Gomila, K. S. Novoselov, and A. K. Geim, Anomalously low dielectric constant of confined water, *Science* **360**, 1339 (2018).
- ⁶L. Bocquet, Nanofluidics coming of age, *Nat. Mater.* **19**, 254 (2020).
- ⁷A. Noy and S. B. Darling, Nanofluidic computing makes a splash, *Science* **379**, 143 (2023).
- ⁸M. Salanne, B. Rotenberg, K. Naoi, K. Kaneko, P.-L. Taberna, C. P. Grey, B. Dunn, and P. Simon, Efficient storage mechanisms for building better supercapacitors, *Nat. Energy* **1**, 16070 (2016).
- ⁹D. S. Sholl and R. P. Lively, Seven chemical separations to change the world, *Nature* **532**, 435 (2016).
- ¹⁰D. L. Gin and R. D. Noble, Designing the next generation of chemical separation membranes, *Science* **332**, 674 (2011).
- ¹¹P. de Gennes, F. Brochard-Wyart, and D. Quere, *Capillarity and Wetting Phenomena: Drops, Bubbles, Pearls, Waves* (Springer New York, 2003).
- ¹²L. R. Fisher, R. A. Gamble, and J. Middlehurst, The kelvin equation and the capillary condensation of water, *Nature* **290**, 575 (1981).
- ¹³R. Evans, Fluids adsorbed in narrow pores: phase equilibria and structure, *J. Phys. Condens. Matter* **2**, 8989 (1990).
- ¹⁴J. Rowlinson and B. Widom, *Molecular Theory of Capillarity*, Dover books on chemistry (Dover Publications, 2002).
- ¹⁵C. Gu, N. Hosono, J.-J. Zheng, Y. Sato, S. Kusaka, S. Sakaki, and S. Kitagawa, Design and control of gas diffusion process in a nanoporous soft crystal, *Science* **363**, 387 (2019).
- ¹⁶Y. Tsoi, F. Tournilhac, and L. Leibler, Demixing in simple fluids induced by electric field gradients, *Nature* **430**, 544 (2004).
- ¹⁷R. A. Hayes and B. J. Feenstra, Video-speed electronic paper based on electrowetting, *Nature* **425**, 383 (2003).
- ¹⁸G. McHale, C. V. Brown, M. I. Newton, G. G. Wells, and N. Sampara, Dielectrowetting driven spreading of droplets, *Phys. Rev. Lett.* **107**, 186101 (2011).
- ¹⁹F. Mugele and J.-C. Baret, Electrowetting: from basics to applications, *J. Phys.: Condens. Matter* **17**, R705 (2005).
- ²⁰G. McHale, C. V. Brown, and N. Sampara, Voltage-induced spreading and superspreading of liquids, *Nat.*

- Commun **4**, 1605 (2013).
- ²¹A. M. J. Edwards, C. V. Brown, M. I. Newton, and G. McHale, Dielectrowetting: The past, present and future, Curr. Opin. Colloid Interface Sci **36**, 28 (2018).
- ²²K. Y. C. Lee, J. F. Klingler, and H. M. McConnell, Electric field-induced concentration gradients in lipid monolayers, Science **263**, 655 (1994).
- ²³R. Dupuis, P.-L. Valdenaire, R. J.-M. Pellenq, and K. Ioannidou, How chemical defects influence the charging of nanoporous carbon supercapacitors, Proc. Natl. Acad. Sci. U.S.A **119**, e2121945119 (2022).
- ²⁴F. Han, G. Meng, F. Zhou, L. Song, X. Li, X. Hu, X. Zhu, B. Wu, and B. Wei, Dielectric capacitors with three-dimensional nanoscale interdigital electrodes for energy storage, Sci. Adv **1**, e1500605 (2015).
- ²⁵A. Castellanos-Gomez, X. Duan, Z. Fei, H. R. Gutierrez, Y. Huang, X. Huang, J. Quereda, Q. Qian, E. Sutter, and P. Sutter, Van der waals heterostructures, Nat. Rev. Methods Primers **2**, 58 (2022).
- ²⁶C. Merlet, B. Rotenberg, P. A. Madden, P.-L. Taberna, P. Simon, Y. Gogotsi, and M. Salanne, On the molecular origin of supercapacitance in nanoporous carbon electrodes, Nat. Mater. **11**, 306 (2012).
- ²⁷V. Kapil, C. Schran, A. Zen, J. Chen, C. J. Pickard, and A. Michaelides, The first-principles phase diagram of monolayer nanoconfined water, Nature **609**, 512 (2022).
- ²⁸T. Xiong, C. Li, X. He, B. Xie, J. Zong, Y. Jiang, W. Ma, F. Wu, J. Fei, P. Yu, and L. Mao, Neuromorphic functions with a polyelectrolyte-confined fluidic memristor, Science **379**, 156 (2023).
- ²⁹P. Robin, N. Kavokine, and L. Bocquet, Modeling of emergent memory and voltage spiking in ionic transport through angstrom-scale slits, Science **373**, 687 (2021).
- ³⁰R. Evans, The nature of the liquid-vapour interface and other topics in the statistical mechanics of non-uniform, classical fluids, Adv. Phys **28**, 143 (1979).
- ³¹W. Kohn, Nobel lecture: Electronic structure of matter—wave functions and density functionals, Rev. Mod. Phys. **71**, 1253 (1999).
- ³²F. Sammüller, S. Hermann, D. de las Heras, and M. Schmidt, Neural functional theory for inhomogeneous fluids: Fundamentals and applications, Proc. Natl. Acad. Sci. U.S.A **120**, e2312484120 (2023).
- ³³F. Sammüller, M. Schmidt, and R. Evans, Neural density functional theory of liquid-gas phase coexistence, Phys. Rev. X **15**, 011013 (2025).
- ³⁴A. T. Bui and S. J. Cox, Learning classical density functionals for ionic fluids, Phys. Rev. Lett. **134**, 148001 (2025).
- ³⁵S. Robitschko, F. Sammüller, M. Schmidt, and R. Evans, Learning the bulk and interfacial physics of liquid-liquid phase separation with neural density functionals, J. Chem. Phys **163**, 161101 (2025).
- ³⁶A. T. Bui and S. J. Cox, A first-principles approach to electromechanics in liquids, J. Phys.: Condens. Matter **37**, 285101 (2025).
- ³⁷H. J. C. Berendsen, J. R. Grigera, and T. P. Straatsma, The missing term in effective pair potentials, J. Phys. Chem. **91**, 6269 (1987).
- ³⁸P. Debye and K. Kleboth, Electrical field effect on the critical opalescence, J. Chem. Phys **42**, 3155 (1965).
- ³⁹M. J. Stevens and G. S. Grest, Phase coexistence of a Stockmayer fluid in an applied field, Phys. Rev. E **51**, 5976 (1995).
- ⁴⁰C. Zhang and M. Sprik, Electromechanics of the liquid water vapour interface, Phys. Chem. Chem. Phys. **22**, 10676 (2020).
- ⁴¹G. Cassone and F. Martelli, Electrofreezing of liquid water at ambient conditions, Nat. Commun **15**, 1856 (2024).
- ⁴²H. A. Pohl, *Dielectrophoresis : the behavior of neutral matter in nonuniform electric fields*, Cambridge monographs on physics (Cambridge University Press, Cambridge, 1978) includes bibliographical references and index.
- ⁴³L. Landau and E. Lifshitz, *Electrodynamics of Continuous Media* (Pergamon Press, 1984).
- ⁴⁴H. A. Pohl and I. Hawk, Separation of living and dead cells by dielectrophoresis, Science **152**, 647 (1966).
- ⁴⁵S. Gangwal, O. J. Cayre, and O. D. Velev, Dielectrophoretic assembly of metallodielectric Janus particles in AC electric fields, Langmuir **24**, 13312 (2008).

- ⁴⁶L. Hong, A. Cacciuto, E. Luijten, and S. Granick, Clusters of charged Janus spheres, *Nano Lett* **6**, 2510 (2006).
- ⁴⁷A. McMullen, M. Muñoz Basagoiti, Z. Zeravcic, and J. Brujic, Self-assembly of emulsion droplets through programmable folding, *Nature* **610**, 502 (2022).
- ⁴⁸A. Walther and A. H. E. Müller, Janus particles: Synthesis, self-assembly, physical properties, and applications, *Chem. Rev* **113**, 5194 (2013).
- ⁴⁹N. Lu, P. Zhang, Q. Zhang, R. Qiao, Q. He, H.-B. Li, Y. Wang, J. Guo, D. Zhang, Z. Duan, Z. Li, M. Wang, S. Yang, M. Yan, E. Arenholz, S. Zhou, W. Yang, L. Gu, C.-W. Nan, J. Wu, Y. Tokura, and P. Yu, Electric-field control of tri-state phase transformation with a selective dual-ion switch, *Nature* **546**, 124 (2017).
- ⁵⁰F. Zhang, H. Zhang, S. Krylyuk, C. A. Milligan, Y. Zhu, D. Y. Zemlyanov, L. A. Bendersky, B. P. Burton, A. V. Davydov, and J. Appenzeller, Electric-field induced structural transition in vertical MoTe₂- and Mo_{1-x}W_xTe₂-based resistive memories, *Nat. Mater.* **18**, 55 (2019).
- ⁵¹J. Hegseth and K. Amara, Critical temperature shift in pure fluid SF₆ caused by an electric field, *Phys. Rev. Lett.* **93**, 057402 (2004).
- ⁵²S. G. Moore, M. J. Stevens, and G. S. Grest, Liquid-vapor interface of the Stockmayer fluid in a uniform external field, *Phys. Rev. E* **91**, 022309 (2015).
- ⁵³K. A. Maerzke and J. I. Siepmann, Effects of an applied electric field on the vapor-liquid equilibria of water, methanol, and dimethyl ether, *J. Phys. Chem. B* **114**, 4261 (2010).
- ⁵⁴A. Z. Panagiotopoulos, Adsorption and capillary condensation of fluids in cylindrical pores by Monte Carlo simulation in the Gibbs ensemble, *Mol. Phys.* **62**, 701 (1987).
- ⁵⁵R. Valiullin, S. Naumov, P. Galvosas, J. Kärger, H.-J. Woo, F. Porcheron, and P. A. Monson, Exploration of molecular dynamics during transient sorption of fluids in mesoporous materials, *Nature* **443**, 965 (2006).
- ⁵⁶T. Horikawa, D. D. Do, and D. Nicholson, Capillary condensation of adsorbates in porous materials, *Adv. Colloid Interface Sci* **169**, 40 (2011).
- ⁵⁷D. C. Grahame, The electrical double layer and the theory of electrocapillarity., *Chem. Rev* **41**, 441 (1947).
- ⁵⁸C. Picard, V. Gérard, L. Michel, X. Cattoën, and E. Charlaix, Dynamics of heterogeneous wetting in periodic hybrid nanopores, *J. Chem. Phys* **154**, 164710 (2021).
- ⁵⁹R. E. Migacz, M. Castleberry, and J. T. Ault, Enhanced diffusiophoresis in dead-end pores with time-dependent boundary solute concentration, *Phys. Rev. Fluids* **9**, 044203 (2024).
- ⁶⁰A. Giacomello, M. Chinappi, S. Meloni, and C. M. Casciola, Metastable wetting on superhydrophobic surfaces: Continuum and atomistic views of the cassie-baxter-wenzel transition, *Phys. Rev. Lett* **109**, 226102 (2012).
- ⁶¹S. Shin, E. Um, B. Sabass, J. T. Ault, M. Rahimi, P. B. Warren, and H. A. Stone, Size-dependent control of colloid transport via solute gradients in dead-end channels, *Proc. Natl. Acad. Sci. U.S.A* **113**, 257 (2016).
- ⁶²S. Shin, J. T. Ault, P. B. Warren, and H. A. Stone, Accumulation of colloidal particles in flow junctions induced by fluid flow and diffusiophoresis, *Phys. Rev. X* **7**, 041038 (2017).
- ⁶³S. Marbach and L. Bocquet, Osmosis, from molecular insights to large-scale applications, *Chem. Soc. Rev.* **48**, 3102 (2019).
- ⁶⁴T. Zimmermann, F. Sammüller, S. Hermann, M. Schmidt, and D. de las Heras, Neural force functional for non-equilibrium many-body colloidal systems, *Mach. Learn.: Sci. Technol.* **5**, 035062 (2024).
- ⁶⁵Y. Tsoni and L. Leibler, Phase-separation in ion-containing mixtures in electric fields, *Proc. Natl. Acad. Sci. U.S.A* **104**, 7348 (2007).
- ⁶⁶A. T. Bui and S. J. Cox, Research data supporting “Dielectrocapillarity for exquisite control of fluid”. Zenodo., <https://doi.org/XX.XXXX/zenodo.XXXXX> (2025).
- ⁶⁷A. T. Bui, GCMC with Gaussian truncated potentials, <https://github.com/annatbui/GCMC> (2024).
- ⁶⁸A. P. Thompson, H. M. Aktulga, R. Berger, D. S. Bolintineanu, W. M. Brown, P. S. Crozier, P. J. in 't Veld, A. Kohlmeyer, S. G. Moore, T. D. Nguyen, R. Shan, M. J. Stevens, J. Tranchida, C. Trott, and S. J. Plimpton, LAMMPS - a flexible simulation tool for particle-based materials modeling at the atomic, meso, and continuum scales, *Comput. Phys. Commun* **271**, 108171 (2022).
- ⁶⁹F. Sammüller, S. Robitschko, S. Hermann, and M. Schmidt, Hyperdensity functional theory of soft matter,

Phys. Rev. Lett. **133**, 098201 (2024).

⁷⁰F. Chollet, *Deep Learning with Python* (Manning Publications, 2017).

⁷¹R. Evans and N. B. Wilding, Quantifying density fluctuations in water at a hydrophobic surface: Evidence for critical drying, Phys. Rev. Lett. **115**, 016103 (2015).